Resource

# Trimodal single-cell profiling reveals a novel pediatric CD8αα+ T cell subset and broad age-related molecular reprogramming across the T cell compartment

Zachary Thomson[1], Ziyuan He [1], Elliott Swanson [1,8], Katherine Henderson[1], Cole Phalen [1], Samir Rachid Zaim[1], Mark-Phillip Pebworth[1], Lauren Y. Okada [1], Alexander T. Heubeck [1], Charles R. Roll[1,9], Veronica Hernandez[1], Morgan Weiss[1], Palak C. Genge[1], Julian Reading [1], Josephine R. Giles[2], Sasikanth Manne[2], Jeanette Dougherty[2], C. J. Jasen[2], Allison R. Greenplate [2,3], Lynne A. Becker [1], Lucas T. Graybuck [1], Suhas V. Vasaikar[1,10], Gregory L. Szeto[1,10], Adam K. Savage [1], Cate Speake [4], Jane H. Buckner [5], Xiao-jun Li [1], Thomas F. Bumol[1], E. John Wherry [2,6], Troy R. Torgerson [1], Laura A. Vella[6,7], Sarah E. Henrickson[6,7], Peter J. Skene [1,11] ✉ & Claire E. Gustafson [1,11] ✉

Age-associated changes in the T cell compartment are well described. However, limitations of current single-modal or bimodal single-cell assays, including flow cytometry, RNA-seq (RNA sequencing) and CITE-seq (cellular indexing of transcriptomes and epitopes by sequencing), have restricted our ability to deconvolve more complex cellular and molecular changes. Here, we profile >300,000 single T cells from healthy children (aged 11–13 years) and older adults (aged 55–65 years) by using the trimodal assay TEA-seq (single-cell analysis of mRNA transcripts, surface protein epitopes and chromatin accessibility), which revealed that molecular programming of T cell subsets shifts toward a more activated basal state with age. Naive CD4+ T cells, considered relatively resistant to aging, exhibited pronounced transcriptional and epigenetic reprogramming. Moreover, we discovered a novel CD8αα+ T cell subset lost with age that is epigenetically poised for rapid effector responses and has distinct inhibitory, costimulatory and tissue-homing properties. Together, these data reveal new insights into age-associated changes in the T cell compartment that may contribute to differential immune responses.

Increased susceptibility to infectious agents such as influenza A virus and *Streptococcus pneumoniae* is known to occur at the extremes of age. However, immune responses in children and older adults are not identical, as demonstrated by the markedly higher rates of hospitalization and death from severe acute respiratory syndrome coronavirus 2 (SARS-CoV-2) infection in older adults[1]. Naive T cell responses are critical for defense against emerging viral infections and long-lasting, effective vaccine responses; however, differential immunity due to T cell variability between healthy children and adults is not well understood.

---

A hallmark of immune aging in adults is the loss of naive CD8+ T cells. Studies have demonstrated that the naive CD8+ T cell compartment is also affected by naive-like memory cell infiltration[2–4] and pseudodifferentiation toward memory-like epigenetic programming that biases naive CD8+ T cell development into effector phenotypes[5,6]. In adult mice, naive CD8+ T cells show altered epigenetic programming that favors the formation of memory T cells, whereas naive CD8+ T cells in newborn mice exhibit more innate-like effector responses to infection[7,8]. Although these mouse studies excluded the naive CD4+ T cell compartment, human naive CD4+ T cells seem less affected by age, with less decline in numbers and fewer molecular changes[9]. Naive CD4+ T cells exhibit age-related functional differences in antigen-specific responses, preferentially polarizing toward programming of T helper type 2 cells in children[10,11]. Moreover, naive CD4+ T cells in older adults are epigenetically biased toward effector-like polarization compared to those in younger adults[12]. This suggests distinct molecular programming directly linked with age in naive CD4+ T cells. A detailed analysis of cellular and molecular heterogeneity within the human CD8+ and CD4+ T cell compartments across age groups is needed to understand differential immune responsiveness.

Most single-cell studies on cellular heterogeneity in humans and mice have been restricted to protein, RNA or chromatin accessibility analysis in a single modality[7,8,13,14], limiting the deconvolution of complex cellular alterations that may occur across age. The novel trimodal assay TEA-seq (single-cell analysis of mRNA transcripts, surface protein epitopes and chromatin accessibility) permits simultaneous single-cell analysis in the proteome, transcriptome and epigenome[15]. This trimodal approach is particularly important for T cells because certain canonical markers can be assessed in only one type of modality, such as protein isoforms, cytokine expression and transcription factor (TF) activity. The ability to differentiate T cell subsets through a combination of three modalities also allows for direct study of the interplay between canonical surface protein phenotypes and transcriptional and epigenetic programs and provides unprecedented, detailed resolution of the complex heterogeneity among T cells.

In this study, we used TEA-seq to dissect the compositional and molecular alterations within the T cell compartment across the spectrum of healthy age. The results showed broad differential transcriptional and epigenetic alterations within the T cell compartment of older adults compared to children. Adult naive CD4+ T cells exhibited a distinct molecular program indicative of low-grade activation despite retaining a surface proteome essentially identical to that in children. The molecular landscape of naive CD8+ T cells was more resilient to aging, but the composition of infiltrating naive-like memory cells differed considerably across age, leading to the discovery of a novel CD8αα+ T cell subset poised for rapid effector responses lost with age (from ~1.5% of T cells in children to <0.05% of T cells in adults). Collectively, these data highlight the complex heterogeneity within the T cell compartment across age. This data resource is also provided at https://explore.allenimmunology.org/explore as an interactive visualization tool for further exploration of human T cells.

## Results

### Age-related transcriptional and epigenetic changes in T cell subsets

To study T cell heterogeneity across human age, we used TEA-seq to perform deep multi-omic analysis of T cells isolated from the peripheral blood of pediatric (aged 11–13 years, $n = 8$) and older adult (aged 55–65 years, $n = 8$) female donors (Fig. 1a). We analyzed a total of 324,255 T cells, including 204,586 CD4+ T cells and 95,832 CD8+ T cells (Fig. 1b). Single-cell RNA sequencing (scRNA-seq) was additionally performed on 541,803 T cells from a cohort of 16 pediatric, 16 young adult (aged 25–35 years) and 16 older adult donors with equal sex distribution (Fig. 1a,b). Antibody-derived tags (ADTs) were used to detect protein abundance and perform cell gating analogous to flow cytometry (Fig. 1c,d and

Extended Data Fig. 1a). Nine T cell subsets were defined according to markers described in Supplementary Table 1. ADT-defined T cell subsets were highly correlated with those detected by spectral flow cytometry across all donors (Extended Data Fig. 1b) but differed from those identified by Seurat RNA-based or assay for transposase-accessible chromatin (ATAC)-based label transfer methods, with an average deviation of 29.3% (Extended Data Fig. 2a–c). Combined data from all three modalities indicated that subsets clustered as expected by differentiation states (Fig. 1e).

The frequencies of ADT-defined T cell subsets in children and older adults were consistent with immune aging, including a reduced frequency of naive CD8+ T cells (Fig. 2a). Transcriptional and epigenetic profiles indicated that age corresponded with differences within subsets (for example, within the naive T cell compartment) more than frequency shifts across subsets (for example, from naive to memory) (Fig. 2b). Conversely, cytomegalovirus (CMV) infection, a common confounder in age-related studies, corresponded with frequency shifts across T cell subsets independent of age (Fig. 2c). Age also had a greater impact on the number of differentially expressed genes (DEGs) and differentially accessible ATAC peaks (DAPs) than CMV infection status (Fig. 2d,e). With age, increased numbers of DEGs and DAPs were found across multiple subsets, including both naive CD8+ and CD4+ T cells. CMV infection had little impact on the transcriptional profile and chromatin landscape of naive CD4+ and CD8+ T cells, consistent with previous reports of CMV infection driving the expansion of effector memory T cells but not naive or central memory T cells[16]. Further pathway analysis of DEGs revealed that older age was associated with downregulation of RNA splicing and oxidative phosphorylation pathways across multiple T cell subsets, whereas CMV infection was associated with downregulation of tumor necrosis factor (TNF) signaling and upregulation of the natural killer (NK) cell cytotoxicity pathway in effector populations (Fig. 2f). Epigenetically, the binding motifs for the TFs FOS and JUN were more accessible, whereas those for nuclear factor-κB (NF-κB) subunit 1 (NFKB1) and the proto-oncoprotein REL were less accessible, in adult T cells (Fig. 2g). No TF binding motif enrichment was associated with CMV infection, in line with CMV driving few epigenetic changes across T cell subsets. Thus, age-specific, global molecular alterations exist in the T cell compartment of children and adults.

### Dynamic molecular reprogramming of naive CD4+ T cells across age

Naive CD4+ T cells in adults are believed to be relatively resistant to aging[9]; however, we observed the most age-related epigenetic changes in this subset compared to all other T cell subsets. This led to the question of whether naive CD4+ T cells may be composed of different subsets and/or demonstrate a distinct molecular program in children compared to adults. To investigate these hypotheses, we performed unsupervised clustering of the ADT-defined naive (CD45RA+C–C motif chemokine receptor 7 (CCR7)+CD27+) CD4+ T cells (99,501 total cells) based on a three-way weighted nearest-neighbor (3WNN) method using a combination of ADT, RNA and ATAC data (Fig. 3a). Subsets identified within the naive CD4+ T cell compartment included true naive T cells (CD49d[ADT]−*FAS*[RNA]−interferon-γ (IFNγ)[ATAC]−), stem cell memory (SCM) cells (CD49d[ADT]+*FAS*[RNA]+IFNγ[ATAC]+) and CD25− regulatory T (T_reg) cells (*FOXP3*[RNA]+CD25[ADT]−*IL2RA*[RNA]+) (Fig. 3b,c). An increased frequency of CD4+ SCM cells (4.2% in children, 9.2% in adults; adjusted *P* value ($P_{adj}$) = 0.03) and a decreased frequency of CD25− T_reg cells (3.4% in children, 1.9% in adults; $P_{adj}$ = 0.03) were observed in adults compared to children. These shifts accounted for a 3.5% increase within the overall naive CD4+ T cell compartment in adults. True naive CD4+ T cells had no significant change in frequency across age (92.3% in children, 88.2% in adults; $P_{adj}$ = 0.23) (Fig. 3d).

We next assessed age-related differences in the surface proteome, transcriptome and epigenome within naive CD4+ T cell subsets. Clustering

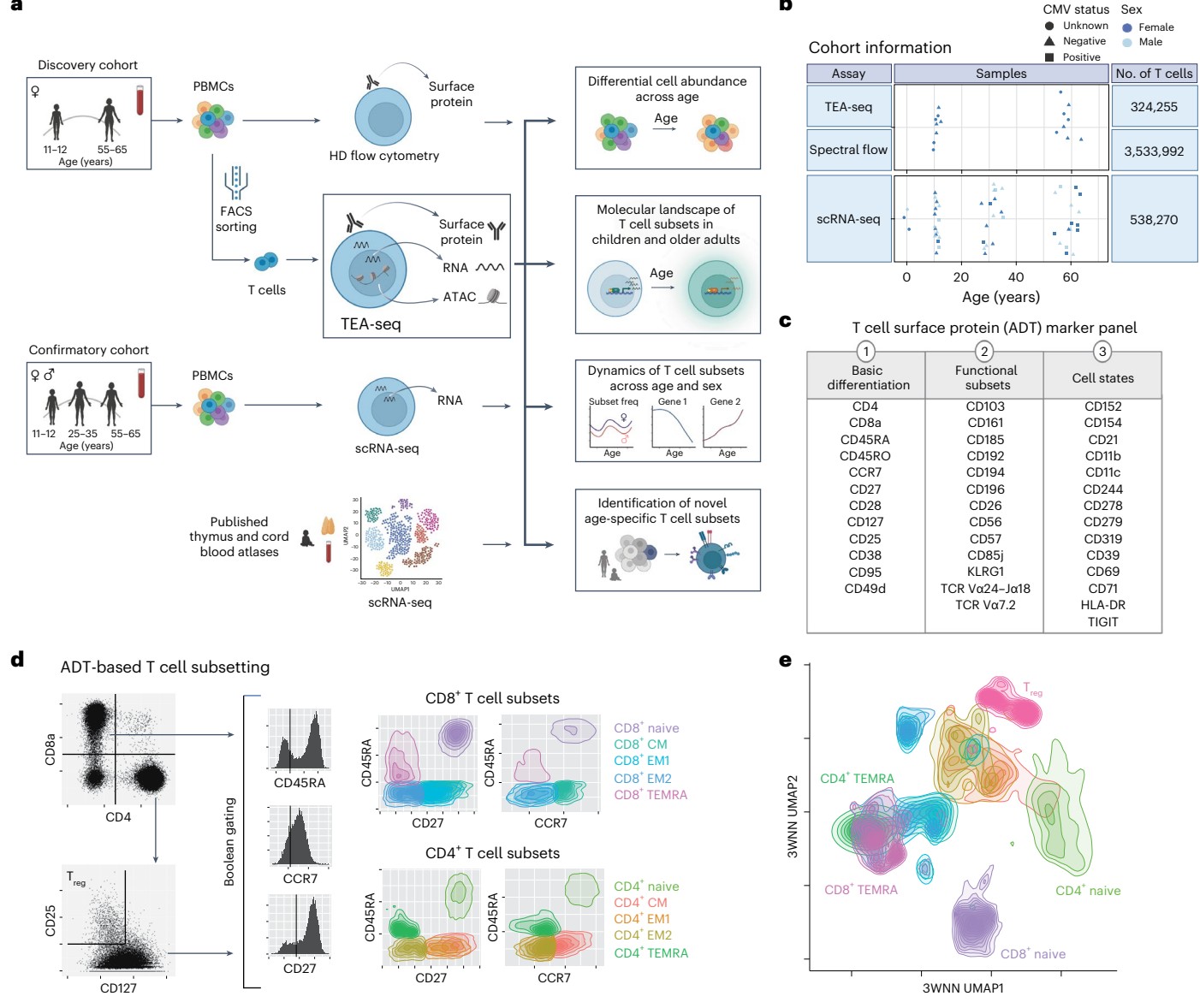

**Fig. 1 | Approach for investigating T cell subsets across age using the trimodal TEA-seq assay. a**, Overview of the discovery (n = 8 donors per age group) and confirmatory (n = 16 donors per age group) cohorts and associated assays. HD, high-dimensional; FACS, fluorescence-activated cell sorting; UMAP 1/2, Uniform Manifold Approximation and Projection 1/2; Subset freq, subset frequency. **b**, Cohort demographics and number of T cells per assay. **c**, T cell-targeted ADT surface marker panel (40 antibodies) used in TEA-seq analysis. HLA-DR, human leukocyte antigen D related; TIGIT, T cell immunoglobulin and immunoreceptor tyrosine-based inhibitory motif domain. **d**, T cell subset gating strategy for TEA-seq data using the expression of seven ADT markers: CD8, CD4, CD25, CD127, CD45RA, CCR7 and CD27. CM, central memory; EM1, effector memory type 1; EM2, effector memory type 2; TEMRA, terminally differentiated effector memory. **e**, 3WNN (ADT + RNA + ATAC) UMAP plot of ADT-defined T cell subsets from all donors, based on cellular density and colored according to T cell subset.

of cells based on surface proteome alone revealed little difference with age (Fig. 3e). However, children showed distinct clustering based on RNA and ATAC profiles (Fig. 3e and Extended Data Fig. 3a,b). True naive CD4$^+$ T cells also had multiple age-related DEGs, with similar numbers within the SCM and CD25$^-$ T$_{reg}$ subsets (Fig. 3f), and showed differences in chromatin accessibility across age (Fig. 3g). Analysis of genes enriched in children identified multiple differentially expressed TFs (for example, *SOX4*, *TOX* and *DACH1*) (Fig. 4a), whereas genes enriched in adults shared expression with CD4$^+$ SCM cells, including the peptidase *CPQ*, the TF *STAT4* and the phosphatidylinositol signaling transducer *INPP4B* (Fig. 4a,b).

We determined whether differential TF expression influences chromatin accessibility. TF motif enrichment across DAPs indicated altered TF usage with age. True naive CD4$^+$ T cells in adults were preferentially biased toward accessibility in regions with TFs related to activation (for example, Krüppel-like factors (KLFs), specific protein 1 (SP1)) and cytokine signaling (for example, IFN regulatory factors (IRFs)) (Fig. 4c,d). Conversely, true naive CD4$^+$ T cells in children had TF motif accessibility associated with NF-κB signaling (for example, RELB, cAMP-responsive element binding protein 1 (CREB1)) and transforming growth factor-β signaling (for example, SOX4). These data indicate that true naive CD4$^+$ T cells are transcriptionally and epigenetically distinct in children and older adults.

To better understand the dynamics of true naive CD4$^+$ T cell reprogramming across age, we performed scRNA-seq on peripheral blood mononuclear cells (PBMCs) from children (n = 16), young adults (n = 16, aged 25–35 years) and older adults (n = 16). We integrated data from an available cord blood scRNA-seq dataset (Fig. 4e).

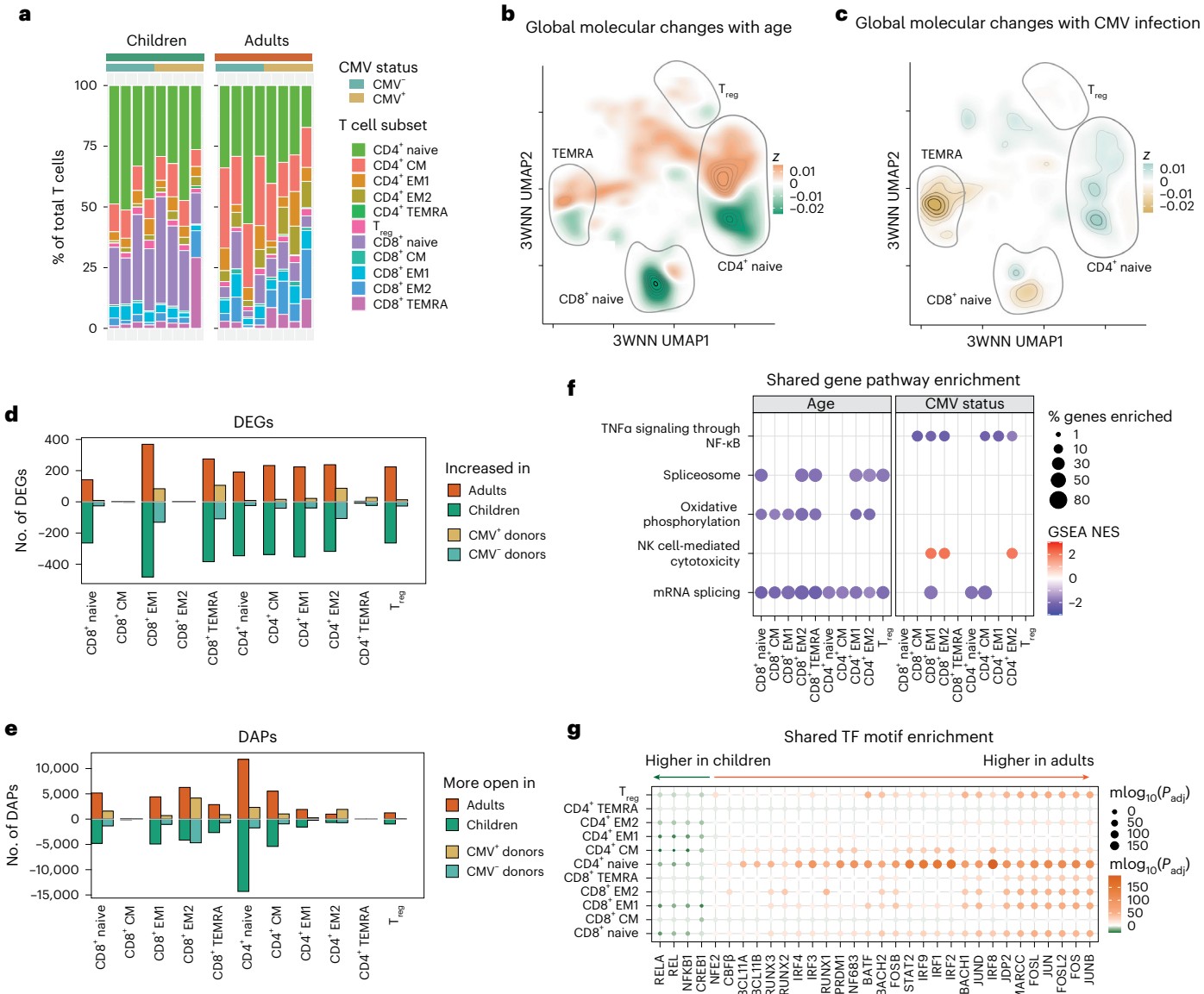

**Fig. 2 | Impact of age on the transcriptional and epigenetic landscape of T cell subsets. a**, Mean frequency of each T cell subset within the T cell compartment in children and older adults, grouped by CMV infection status. **b,c**, 3WNN UMAP plots colored according to cell density in each age category (**b**; green, greater in children; orange, greater in older adults) or in each CMV infection status group (**c**; blue, greater in CMV-negative donors; yellow, greater in CMV-positive donors). **d,e**, Number of DEGs (**d**) and DAPs (**e**) within each T cell subset by age (green, higher in children; orange, higher in older adults) or CMV infection status (blue, higher in CMV-negative donors; yellow, higher in CMV-positive donors). **f**, Gene set enrichment analysis (GSEA) of each T cell subset, comparing age- or CMV infection status-related differences. A false discovery rate (FDR) of <0.05 was considered significant. Dot size corresponds to the percentage of leading edge genes enriched in the indicated pathway. Dot color corresponds to the normalized enrichment score (NES). **g**, Shared TF motif enrichment based on

DAPs between age groups or CMV infection status within each T cell subset. No significant motifs were detected for CMV comparisons. Both the size and color of each point correspond to the $P_{adj}$ of enrichment determined by hypergeometric testing, with green indicating higher accessibility in pediatric donors and orange indicating higher accessibility in adult donors. NFE2, nuclear factor, erythroid 2; CBFβ, core-binding factor subunit β; BCL11A/BCL11B, B-cell lymphoma/leukemia 11A/B; RUNX1/RUNX2/RUNX3, Runt-related TF 1/2/3; IRF1/IRF2/IRF3/IRF4/IRF8/IRF9, IFN regulatory factor 1/2/3/4/8/9; PRDM1, PR domain zinc finger protein 1; ZNF683, zinc finger protein 683; BATF, basic leucine zipper TF, ATF-like; BACH1/BACH2, broad complex-tramtrack-bric a brac and cap'n'collar homology 1/2; STAT2, signal transducer and activator of transcription 2; JDP2, JUN dimerization protein 2; SMARCC, SWI/SNF-related, matrix-associated, actin-dependent regulator of chromatin subfamily C.

A total of 124,564 naive CD4+ T cells were identified using Seurat's reference-based RNA label transfer method, which had a 90% agreement with our TEA-seq 'true' naive CD4+ T cell designations (Extended Data Fig. 2d), in contrast to 76% agreement with the ADT-only naive CD4+ T cell designations (Extended Data Fig. 2b). Naive CD4+ T cells from children clustered separately from those from both young and older adults (Fig. 4f). Naive CD4+ T cells from young and older adults also exhibited more similar gene expression patterns compared to

naive CD4+ T cells from children (Extended Data Fig. 3c). Consistent with this, two pediatric-signature genes, *TOX* and *SOX4*, were highly expressed in naive CD4+ T cells from cord blood but showed decreased expression with age, whereas older adult-signature genes (for example, *CPQ*, *STAT4*) demonstrated a stepwise increase with age (Fig. 4g). These changes were also confirmed by bulk reverse transcription followed by qPCR (Extended Data Fig. 3d). Together, these data demonstrate that the pediatric-specific molecular programming of

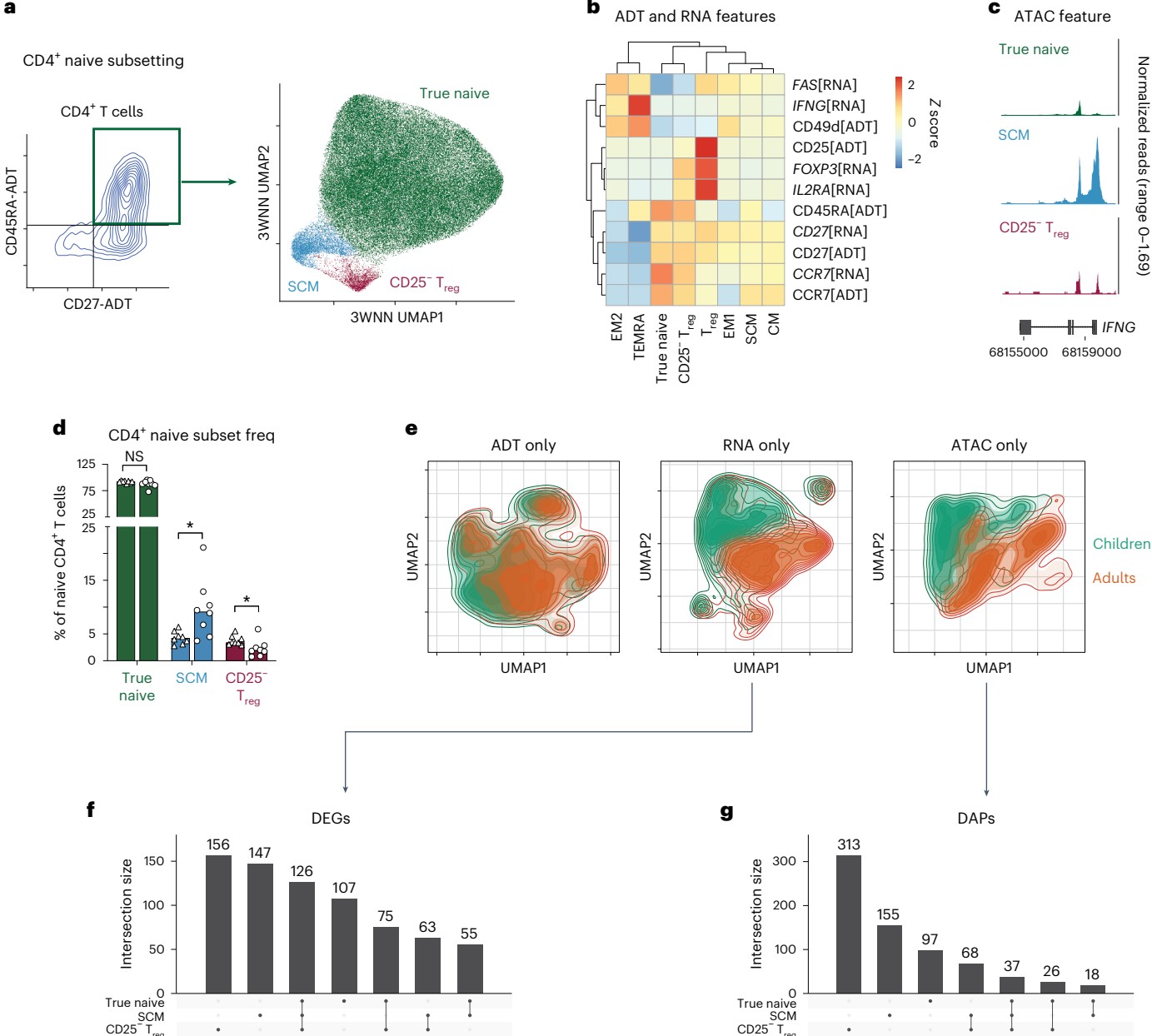

**Fig. 3 | Age-specific alterations in the naive CD4⁺ T cell compartment.**
**a**, Identification of subsets within CD4⁺CD27⁺CD197⁺CD45RA⁺ T cells through a trimodal analysis, shown in a 3WNN UMAP plot with the true naive, SCM and CD25[ADT]⁻ T_reg subsets colored. **b**, ADT and RNA markers delineating naive CD4⁺ T cell subsets. The modality of detection is indicated in square brackets. **c**, Chromatin accessibility tracks of the *IFNG* gene region in naive CD4⁺ T cell subsets, showing normalized read coverage. **d**, Bar plot (median value shown) of the frequencies of naive CD4⁺ T cell subsets within the overall naive CD4⁺ compartment by age group (*n* = 8 per group). Triangles are children and circles are adults. *P* values were determined by a two-tailed Mann–Whitney test with the Holm–Sidak multiple-comparison method. *\*P* < 0.05 (*P* = 0.03); NS, not significant. **e**, Single-modality (ADT, RNA or ATAC) cell density UMAP plots colored by age group (green, children; orange, adults). **f,g**, UpSet plots showing the number of DEGs (**f**) or DAPs (**g**) between age groups for each combination of naive CD4⁺ T cell subsets.

true naive CD4⁺ T cells arises early in life and gradually shifts toward a transcriptionally and epigenetically distinct state in adults.

**Age-specific reorganization of naive-like memory CD8⁺ T cells**
Compositional heterogeneity in 'naive' CD8⁺ T cells is known to change during adult aging with an expansion of CD8⁺ SCM and memory-like naive precursor (MNP) populations[3,4,17]. However, whether these compositional changes extend to the naive CD8⁺ T cell compartment of children is unclear. Using unsupervised reclustering of ADT-defined naive CD8⁺ T cells (46,122 total cells), we identified five

cell subsets within the naive CD8⁺ T cell compartment: true naive T cells (CD49d[ADT]⁻*FAS*[RNA]⁻IFNγ[ATAC]⁻), SCM cells (CD49d[ADT]⁺*FAS*[RNA]⁺IFNγ[ATAC]⁺), two MNP populations (MNP-1 and MNP-2, CD49d[ADT]ʰⁱ*FAS*[RNA]ˡᵒʷIFNγ[ATAC]ˡᵒʷ/⁺) and mucosal-associated invariant T cells (MAIT; T cell receptor (TCR) Vα7.2[ADT]⁺ and CD161[ADT]⁺) (Fig. 5a–c). The frequencies of the CD8⁺ SCM (2.1% in children, 8.8% in adults; *P*_adj = 0.02) and CD8⁺ MNP-1 (3% in children, 7.3% in adults; *P*_adj = 0.003) T cell subsets increased with age (Fig. 5d). However, the frequency of the CD8⁺ MNP-2 subset significantly decreased with age (3.3% in children, 0.8% in adults; *P*_adj = 0.0008). No difference in

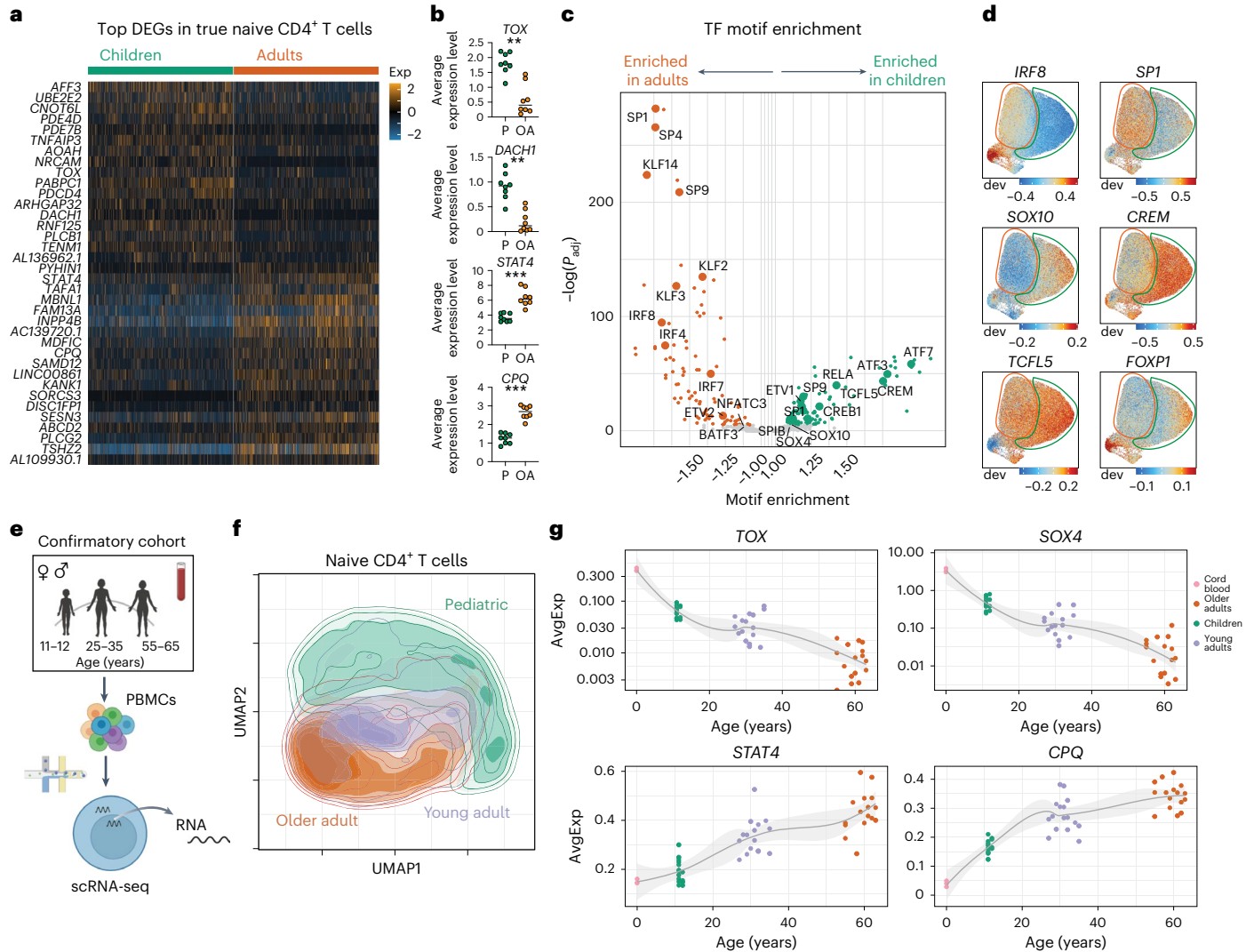

**Fig. 4 | Molecular reprogramming of naive CD4⁺ T cells across age. a**, Heat map of the top 20 DEGs for each age group in individual true naive CD4⁺ T cells. For visualization, values are scaled (*z* score) per gene. Exp, scaled expression. **b**, Dot plots of average pseudobulk gene expression for select transcripts in true naive CD4⁺ T cells separated by age (*n* = 8 per group; P, pediatric; OA, older adult). The line indicates the median value. *P* values were determined by a two-tailed Mann–Whitney test. \*\**P* = 0.0006, \*\*\**P* = 0.0002. **c**, TF binding motif enrichment comparison between age groups in true naive CD4⁺ T cells. The *P*$_{adj}$ of enrichment was determined by hypergeometric testing. ETV1/ETV2, ETS translocation variant 1/2; NFATC3, nuclear factor of activated T cells, cytoplasmic

3; ATF3/ATF7, activating TF 3/7; TCFL5, TF-like 5 protein; CREM, cAMP-responsive element modulator; SPIB, Spi-B TF; SOX4/SOX10, SRY-box TF 4/10. **d**, ChromVar motif enrichment UMAP plots. Areas enriched for true naive CD4⁺ T cells in older adults (orange) and children (green) are outlined. dev, deviation. **e**, Overview of the scRNA-seq confirmatory cohort (*n* = 16 per age group). **f**, RNA-based UMAP plot of naive CD4⁺ T cells from the confirmatory cohort. **g**, Average pseudobulk expression of select signature genes in the naive CD4⁺ T cell subset for each donor across all age groups, including an external cord blood (*n* = 3) dataset. Best-fit lines with 95% confidence intervals are shown. AvgExp, average expression.

true naive CD8⁺ T cells or naive MAIT CD8⁺ T cells was observed. Overall, the frequency of naive-like memory CD8⁺ T cells increased from ~9% in children to ~19% in adults (Fig. 5e), contrary to small shifts in the CD4⁺ compartment.

The unexpected age-related heterogeneity in naive-like memory T cells included a novel pediatric-specific population that we termed 'MNP-2'. We next analyzed the molecular relationship of this unique subset to the entire T cell compartment. Other naive-like memory populations (SCM and MNP-1) clustered with memory subsets, whereas MNP-2 cells grouped with a distinct, unknown cluster of T cells (Fig. 5f). The SCM and MNP-1 subsets also showed high similarity to memory CD8⁺ T cells in individual ATAC and RNA analyses (Fig. 5g and Extended Data Fig. 4a). scRNA-seq revealed that all naive-like memory CD8⁺ subsets expressed naive-like transcription and quiescence factors such as *LEF1*, *BACH2* and *FOXP1*; however, each subset also expressed

a unique profile of integrins, NK surface receptors, TFs and effector molecules (Fig. 5h and Supplementary Table 2). All naive-like memory subsets exhibited enriched TF motif accessibility related to increased effector function, such as the eomesodermin (EOMES) and T-box 21 (TBX21; also known as T-bet) motifs, compared to true naive CD8⁺ T cells (Extended Data Fig. 4b–d). However, the MNP-2 subset was distinctly enriched for the KLF and SP motifs, whereas the SCM and MNP-1 subsets were more significantly enriched for the JUN/FOS motifs (Extended Data Fig. 4d), suggesting that MNP-2 cells are distinct from the classic memory CD8⁺ T cell subsets.

To confirm the age-related dynamics of MNP-2 cells, we used the gene expression signature (*KLRC3⁺LEF1⁺CD8A⁺*) of these cells to identify them in our scRNA-seq dataset (Fig. 5i). Consistent with our TEA-seq analysis, the median MNP-2 cell frequencies showed a ~10-fold reduction with age, decreasing from 1.6% in children to 0.04% in older adults

(Fig. 5j). Thus, we found an age-specific restructuring of naive-like memory T cell subsets within the 'naive' CD8[+] T cell compartment, highlighted by the loss of a unique, previously undescribed naive-like memory T cell subset in adults.

## A novel subset of CD8αα[+] T cells in children lost with age

The identification of a pediatric-specific, naive-like memory subset was unexpected. Given the uniquely high expression of *KLR* transcripts in MNP-2 cells, we hypothesized that these cells are innate-like T cells. To assess this, we reanalyzed our TEA-seq dataset in situ to identify all MAIT cells (TCR Vα7.2[ADT][+] and CD161[ADT][+], 9,948 cells) as well as Vδ1[+] and Vδ2[+] γδ T cells (*TRGC1*[RNA][+] or *TRGC2*[RNA][+] and *TRGDC*[RNA][+], 12,630 cells with 4,451 *TRDV1*[RNA][+] (Vδ1[+]) and 8,179 *TRDV2*[RNA][+] (Vδ2[+]) cells) among CD3[+] T cells (Fig. 6a). 3WNN clustering of these populations revealed that the MNP-2 subset is similar to a subpopulation of lymphoid enhancer-binding factor 1 (LEF1)[hi]Vδ1[+] γδ T cells, also specifically enriched in children (Fig. 6b and Extended Data Fig. 5a,c). However, MNP-2 cells did not express γδ TCR genes (*TRGC2*[RNA][+], *TRDC*[RNA][+], *TRDV1*[RNA][+], *TRDV2*[RNA][+]) or MAIT TCR (TCR Vα7.2[ADT][+]). Meanwhile, they expressed *TRGC1* and αβ TCR genes (*TRAC*[RNA][+]*TRBC2*[RNA][+]), in line with the gene signatures of unconventional CD8[+] T cells (Fig. 6c). MNP-2 cells did not express protein or RNA for the classic NK T cell marker CD56 (*NCAM1* gene; Fig. 6c) or show CMV-specific enrichment (Extended Data Fig. 5b) but expressed similar RNA levels of *NKG7* as the MAIT and γδ T cell subsets (Fig. 6d). Notably, RNA expression of *CD8A* in the absence of *CD8B* expression (Fig. 6d) suggests that MNP-2 cells may be a CD8αα[+] population distinct from classic innate-like subsets.

We next integrated our MNP-2 dataset with a pediatric thymus scRNA-seq dataset that identified a new subset of thymic CD8αα[+] T cells[18]. Unlike the majority of pediatric MAIT and γδ T cells, MNP-2 cells clustered closely with thymic-derived T cells (Fig. 6e). Notably, MNP-2 cells were most similar to the thymic *ZNF683*-expressing CD8αα[+] subtype but retained much higher levels of the interleukin-21 (IL-21) receptor (*IL21R*) (Fig. 6e–g). In silico reanalysis of key surface protein markers of the MNP-2 population revealed high IL-21R, CD244 and CD11b coexpression (Extended Data Fig. 6a–c). Transcriptional analysis of CD244[+]CD11b[+]CD8[+] T cells from cord blood confirmed a CD8αα[+] T cell gene signature (Extended Data Fig. 6d). Moreover, the surface protein profile of MNP-2 cells, which showed a CD8α[hi]CD8β[low] phenotype (Extended Data Fig. 6e), was distinct from that expressed by activated naive CD8[+] T cells over time (Extended Data Fig. 7), suggesting that MNP-2 cells are a unique population of CD8αα[+] T cells in children.

As the variable range of MNP-2 cell frequencies implied compositional diversity, we further examined MNP-2 heterogeneity. Integrated reanalysis of the MNP-2 cluster (2,804 total cells) in our TEA-seq dataset revealed multiple CD8αα[+] T cell clusters in children that were globally lost with age (Fig. 6h). Chromatin accessibility analysis showed that the three main transcriptionally distinct clusters (that is, 1, 2 and 3) were epigenetically similar (Fig. 6i). Moreover, these clusters exhibited

key RNA features of the original MNP-2 population, including high expression of *KLRC2*, *IL21R* and *LEF1* (Fig. 6j and Extended Data Fig. 8a). Remaining clusters were identified as *MME*[RNA][+]PD-1[ADT][hi], *CR1*[RNA][+] and two subsets of CD4[+] T cells (Extended Data Fig. 8a–c). IL-21R[hi] MNP-2 cells were present in three different states, highlighted by the RNA expression of different functional markers, including granzyme K (*GZMK*), granulysin (*GNLY*) and the integrin *ITGB1* (Fig. 6j and Extended Data Fig. 8b). However, these populations maintained many similarities, including high expression of TFs related to naivety (for example, *FOXP1*, *LEF1*) and effector function (for example, *TBX21*) (Fig. 6k and Extended Data Fig. 8b). MNP-2 heterogeneity was similar among pediatric donors; however, children with CMV infection trended toward having a greater reduction in the frequency of 'resting' MNP-2 cells (Extended Data Fig. 8d,e). Collectively, these data demonstrate the presence of multiple types of CD8αα[+] T cells in children, with a dominant CD244[+]CD11b[+] 'MNP-2' population.

## MNP-2 cells are poised for memory-like effector responses

Given the high basal expression of IL-21R in MNP-2 cells (Fig. 5h and Extended Data Fig. 6), we investigated the functional capacity of this population to respond to IL-21 stimulation through CITE-seq (cellular indexing of transcriptomes and epitopes by sequencing) analysis of pediatric CD8[+] T cells (*n* = 4 donors) (Fig. 7a), allowing simultaneous interrogation of naive, MNP-2 and memory CD8[+] T cells, as well as MAIT and γδ T cells, before and 4 h after stimulation (Fig. 7b). All subsets demonstrated a transcriptional response to IL-21 stimulation, including upregulation of the cytokine signaling-related genes *JAK3*, *STAT3* and *SOCS1* (Fig. 7c and Extended Data Fig. 9a). Gene expression patterns in the MNP-2 and memory subsets were distinct from those in naive CD8[+] T cells, including the highest expression of *BCL6* in MNP-2 cells (Fig. 7d,e). The phenotypic profile of MNP-2 cells was also distinct from that of virtual memory cells (Extended Data Fig. 9b)[19,20]. Like other memory T cell populations, MNP-2 cells upregulated the cytolytic molecule *PRF1* in response to IL-21 stimulation (Fig. 7e), suggesting a cytotoxic role in specific IL-21-rich tissue contexts.

We next compared early functional responses to direct TCR stimulation (anti-CD3/anti-CD28 beads) (that is, what a T cell does do in response to an antigen) and phorbol 12-myristate 13-acetate plus ionomycin (PMA/iono) activation (that is, what a T cell could do in response to an antigen) (Fig. 8a,b). Indicative of global activation, all T cell subsets from the four donors had upregulated expression of CD69 (Extended Data Fig. 10a,b). MNP-2 cells exhibited transcriptional changes reflective of memory CD8[+] T cells, with a small set of unique TCR-induced genes compared to other subsets (Fig. 8c). MNP-2 cells lacked upregulation of genes involved in RNA metabolism, unlike both naive and memory cells (Extended Data Fig. 10c). After TCR stimulation, pediatric memory CD8[+] T cells had increased *IFNG* expression, whereas MNP-2 cells had significantly lower expression of *IFNG* (Fig. 8d and Extended Data Fig. 10d). The limited *IFNG* expression was not due to these cells exhibiting exhausted (that is, T cell immunoglobulin and mucin domain-containing protein 3 (TIM3), lymphocyte

**Fig. 5 | Reorganization of the naive-like memory CD8[+] T cell compartment across age. a**, Identification of subsets within CD8[+]CD27[+]CD197[+]CD45RA[+] T cells through a trimodal analysis, shown in a 3WNN UMAP plot with the true naive, SCM, MNP-1, MNP-2 and MAIT subsets colored. **b**, Expression of select RNA and ADT cell type markers, shown in 3WNN UMAP plots. The modality of detection is indicated in square brackets. Density, gene-weighted 2D kernel density. **c**, Chromatin accessibility tracks of the *IFNG* gene region in naive CD8[+] T cell subsets, showing normalized read coverage. **d**, Bar plot (median value shown) of the frequencies of naive CD8[+] T cell subsets within the overall naive CD8[+] compartment by age group (*n* = 8 per group). *P* values were determined by a two-tailed Mann–Whitney test with the Holm–Sidak multiple-comparison method. \**P* < 0.05 (*P* = 0.02), \*\**P* < 0.01 (*P* = 0.003), \*\*\**P* < 0.001 (*P* = 0.0008). **e**, Age-specific composition of the non-naive compartment found within naive CD8[+]

T cells. **f**, 3WNN UMAP plot of all T cells overlaid with naive CD8[+] T cell subsets and separated by age. Only cells from the naive CD8[+] T cell compartment of children (left) or adults (right) are colored; all other cells are gray. **g**, Comparison of differential chromatin accessibility across all CD8[+] T cell subsets (24,874 features). For visualization, all values are scaled (*z* score) per differential region. **h**, Dot plot of select DEGs across naive CD8[+] T cell subsets. The size of points corresponds to the fraction of cells expressing each gene; color corresponds to average expression. AvgExp, scaled average expression. **i**, Identification of the MNP-2 subset through gene expression profiling in the scRNA-seq confirmatory cohort. Density, gene-weighted 2D kernel density. **j**, MNP-2 subset frequencies within the total T cells across all age groups including an external cord blood (*n* = 3) dataset.

activation gene 3 (LAG-3), EOMES, cytotoxic T lymphocyte-associated protein 4 (CTLA-4)) or senescent (CD57, killer cell lectin-like receptor G1 (KLRG1), CD85j) protein or RNA signatures (Extended Data Fig. 10e)[21]. However, MNP-2 cells lacked surface expression of the CD28 costimulatory receptor (Extended Data Fig. 10f) and thus cannot respond to costimulatory signals provided by anti-CD3/anti-CD28 beads, indicating differential TCR signaling in MNP-2 cells.

To bypass any potential altered regulation of the TCR complex, we next performed stimulation with PMA/iono. We found an ~84-fold increase in *IFNG* expression with stimulation (Fig. 8d,e and Extended Data Fig. 10d) and similar increases in other effector-related genes such as *CCL3*, *CCL4*, *CCL5* and *CSF2* (Extended Data Fig. 10g). Although their responses were more similar to those of memory rather than naive cells (Fig. 8c,d), MNP-2 cells were not polyfunctional, as demonstrated

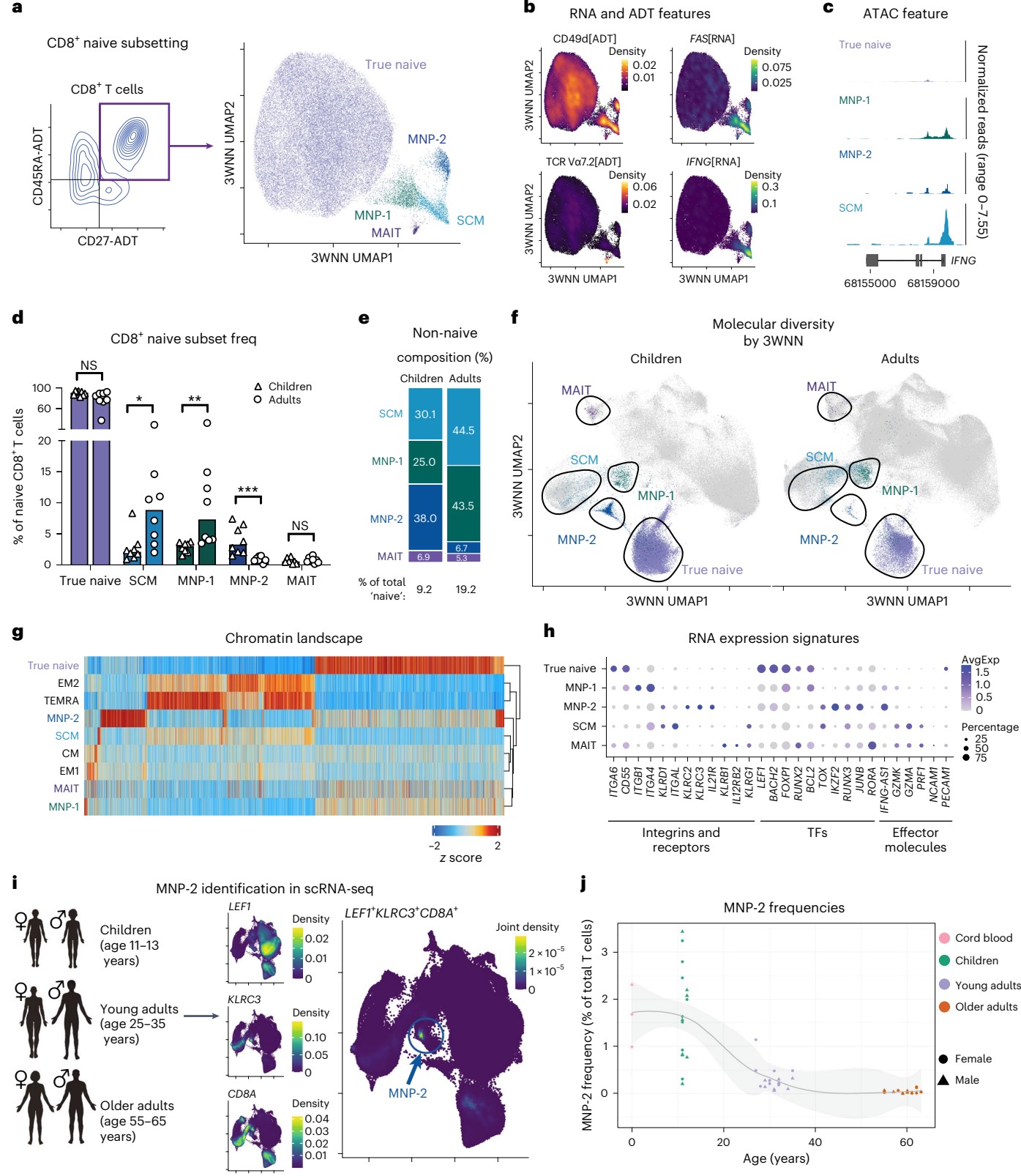

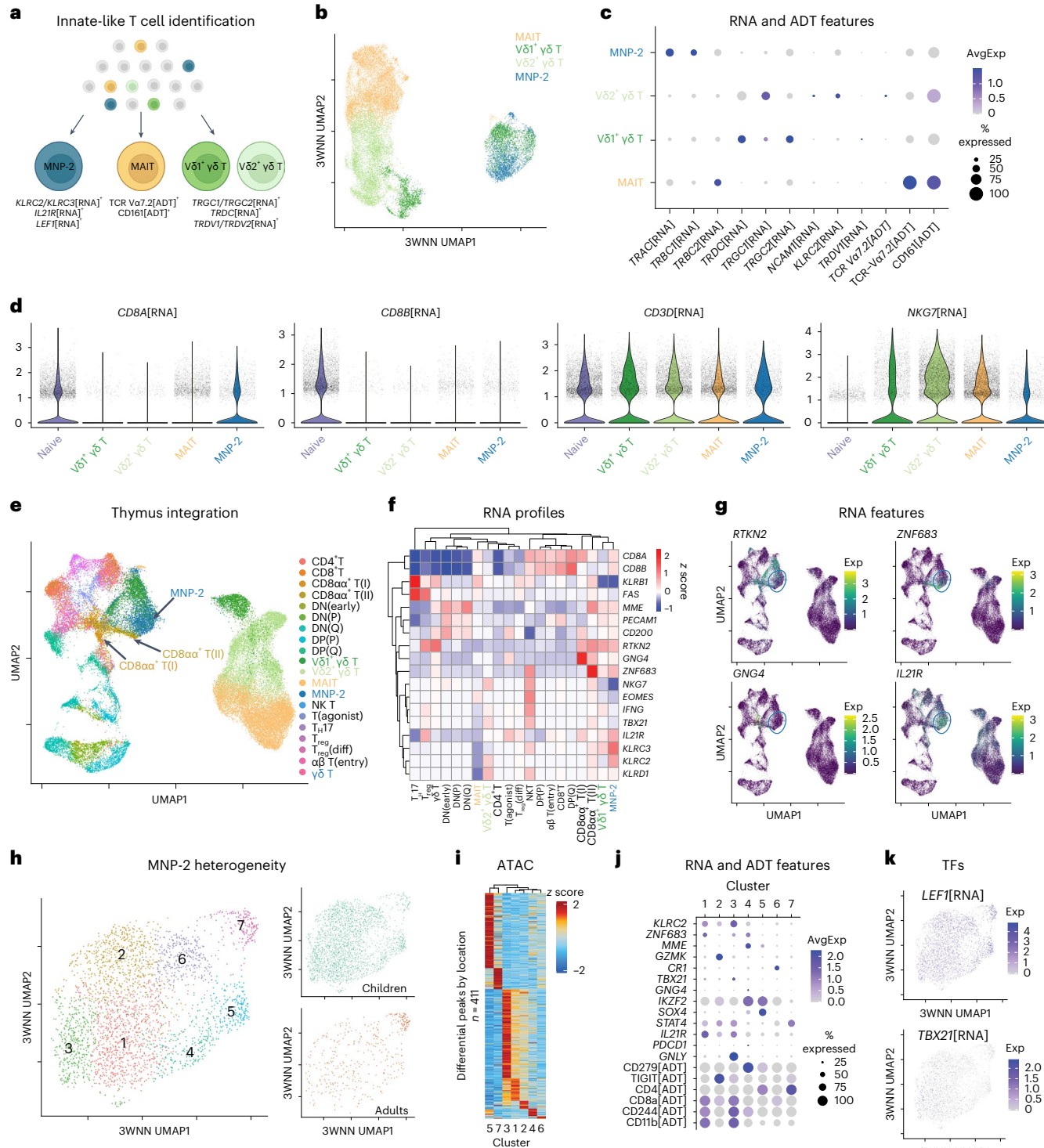

**Fig. 6 | A pediatric-specific naive-like memory CD8⁺ T cell subset (MNP-2) is a unique IL-21RʰⁱCD8αα⁺ population. a**, In situ reanalysis of the TEA-seq dataset for multimodal identification of MNP-2, MAIT and γδ T cell populations. **b**, 3WNN UMAP plot of the MNP-2, MAIT, Vδ1⁺ γδ and Vδ2⁺ γδ T cell populations. **c**, Dot plot showing the expression of γδ T (for example, *TRDC*[RNA], *TRGC1*[RNA], *TRDV1*[RNA]), MAIT (for example, TCR Vα7.2[ADT], CD161[ADT]) and NK T (for example, *NCAM1*[RNA]) cell-type-specific markers on each defined T cell subset. **d**, Violin plots of the single-cell expression of select genes for all T cells (for example, *CD3D*[RNA]), T cell coreceptors (for example, *CD8A*[RNA], *CD8B*[RNA]) and innate-like T cells (for example, *NKG7*[RNA]). **e**, UMAP integration of RNA expression for MNP-2, MAIT and γδ T cells from the TEA-seq dataset with an external pediatric thymic T cell dataset[35]. DN, double negative; DP, double positive; P, proliferating; Q, quiescent; T_H17, T helper type 17 cell;

diff, differentiating. **f**, Heat map of select genes related to T cell subsets and functionality compared across T cell types. For visualization, values are scaled (z score) for each gene. Hierarchical clustering of rows (genes) and columns (cell types) was constructed using pheatmap. **g**, CD8αα⁺ subset-specific gene expression shown in integrated RNA UMAP plots with the MNP-2 population circled in blue. **h**, Subclustering of MNP-2 cells shown in a 3WNN UMAP plot (clusters are numbered); right plots show cells divided by age (green, children; orange, adults). **i**, Comparison of differential chromatin accessibility across MNP-2 subclusters (411 features). For visualization, all values are scaled (z score) per differential region. **j**, Dot plot of select protein and RNA expression of cluster-defining markers. **k**, Single-cell RNA expression of the TFs *TBX21* and *LEF1* in MNP-2 subsets, shown in 3WNN UMAP plots.

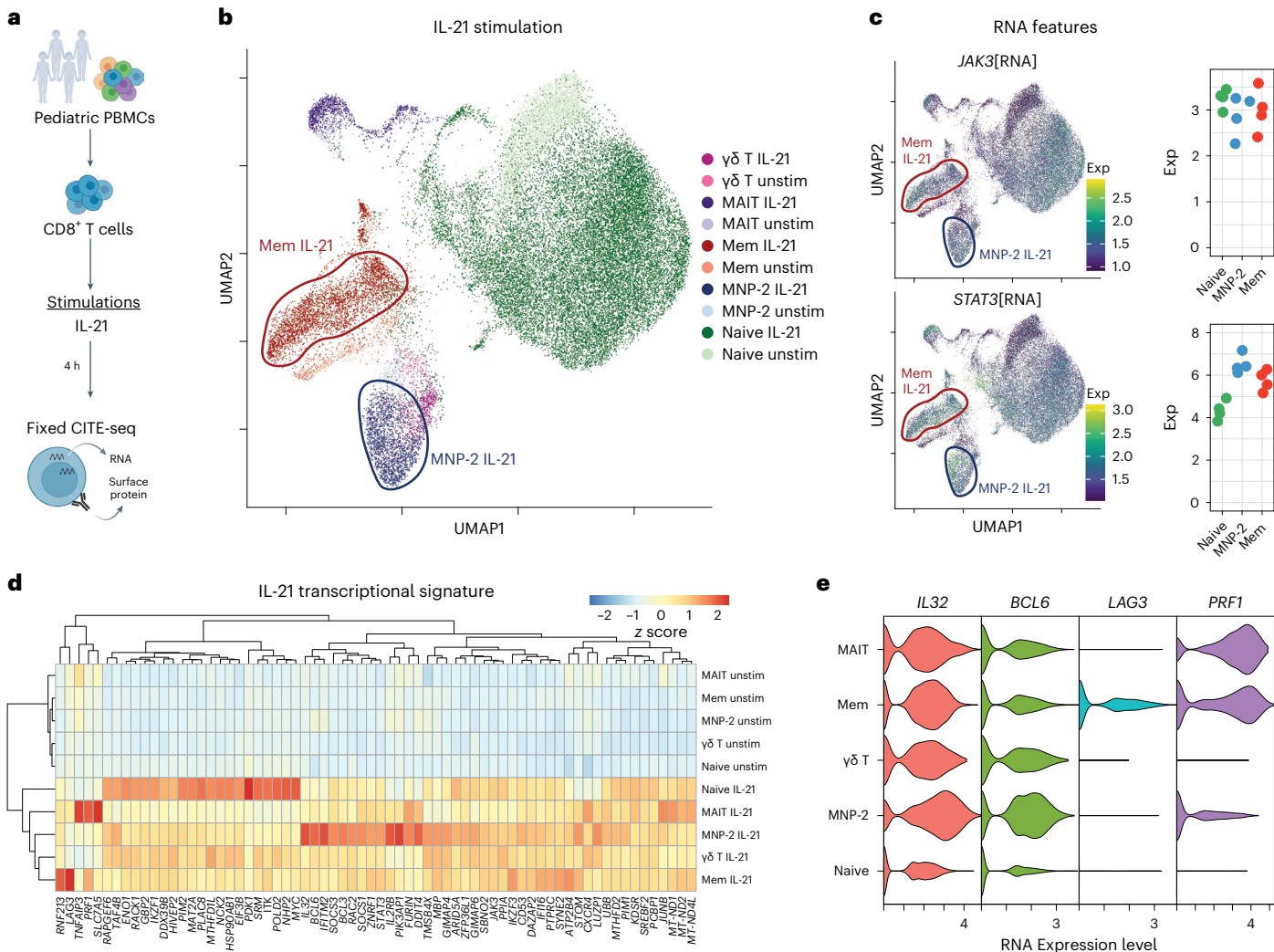

**Fig. 7 | IL-21-induced responses in pediatric CD8⁺ T cell subsets. a**, Overview of a fixed CITE-seq experiment for IL-21 stimulation. **b**, RNA-based UMAP plot of unstimulated and IL-21 (50 ng ml⁻¹, 4 h)-stimulated pediatric CD8⁺ T cells (*n* = 4 pediatric donors). Select stimulated subsets are indicated. Mem, memory; unstim, unstimulated. **c**, Select gene expression indicative of IL-21R signaling in IL-21 stimulation, shown in RNA-based UMAP plots. Pseudobulk RNA expression in naive, MNP-2 and memory CD8⁺ T cells after IL-21 stimulation is shown to the right of each UMAP plot. exp, average expression level. **d**, Comparison of DEGs across each subset of IL-21-stimulated and unstimulated CD8⁺ T cells. **e**, Violin plots of select DEGs from CD8⁺ T cell subsets after stimulation with IL-21.

by the absence of other effector molecules such as *TNF*, *IL2* and *GZMB* after stimulation (Fig. 8d,e) and consistent with upregulated expression of *SPRY2*, a known suppressor of polyfunctionality (Extended Data Fig. 10d)[22]. MNP-2 cells also exhibited the strongest upregulation of the costimulatory receptor 4-1BB (that is, *TNFRSF9* RNA and CD137 protein) and the mucosal tissue-homing molecule *CRTAM* (Fig. 8e and Extended Data Fig. 10h). Thus, MNP-2 cells are poised to rapidly express *IFNG* in response to antigens but not intrinsically polyfunctional like the classic memory CD8⁺ T cells in children.

The poised effector state of MNP-2 cells in conjunction with features of tissue homing leads to the question of whether this population may have a role in immunity against infection and/or in inflammation. Although scRNA-seq studies on children are limited, we were able to detect MNP-2 cells using our TEA-seq-defined signature in children with SARS-CoV-2-associated multisystem inflammatory syndrome (MIS-C)[23] (Fig. 8f). Children with active MIS-C had a markedly decreased frequency of MNP-2 cells compared to healthy controls (Fig. 8g). Moreover, children with more severe disease had even lower MNP-2 cell frequencies than those with moderate disease, with levels rebounding after recovery (Extended Data Fig. 10i). Analysis of TCR gene usage also

revealed a broad repertoire in MNP-2 cells in children (Extended Data Fig. 10j), indicating that MNP-2 cells are a diverse population of T cells that are recruited to sites of active inflammation and may contribute to immune resolution within tissues in children.

## Discussion

Aging has a profound impact on T cells; however, our understanding of the complexity of this impact across the age spectrum is limited. Here, we used TEA-seq to simultaneously interrogate the cellular and molecular heterogeneity of the T cell compartment in children and adults. We established that age considerably affects the composition, transcriptome and epigenome across T cell subsets in contrast to CMV infection, which preferentially affects composition due to expansion of effector populations. Detailed interrogation of naive T cell subsets revealed substantial molecular reprogramming in the CD4⁺ compartment, whereas the CD8⁺ compartment exhibited compositional changes driving age-related differences, including the loss of a unique effector CD8αα⁺ T cell subset in adults.

Immune aging is marked by the numerical loss of naive CD8⁺ T cells; however, more recent studies have indicated that memory

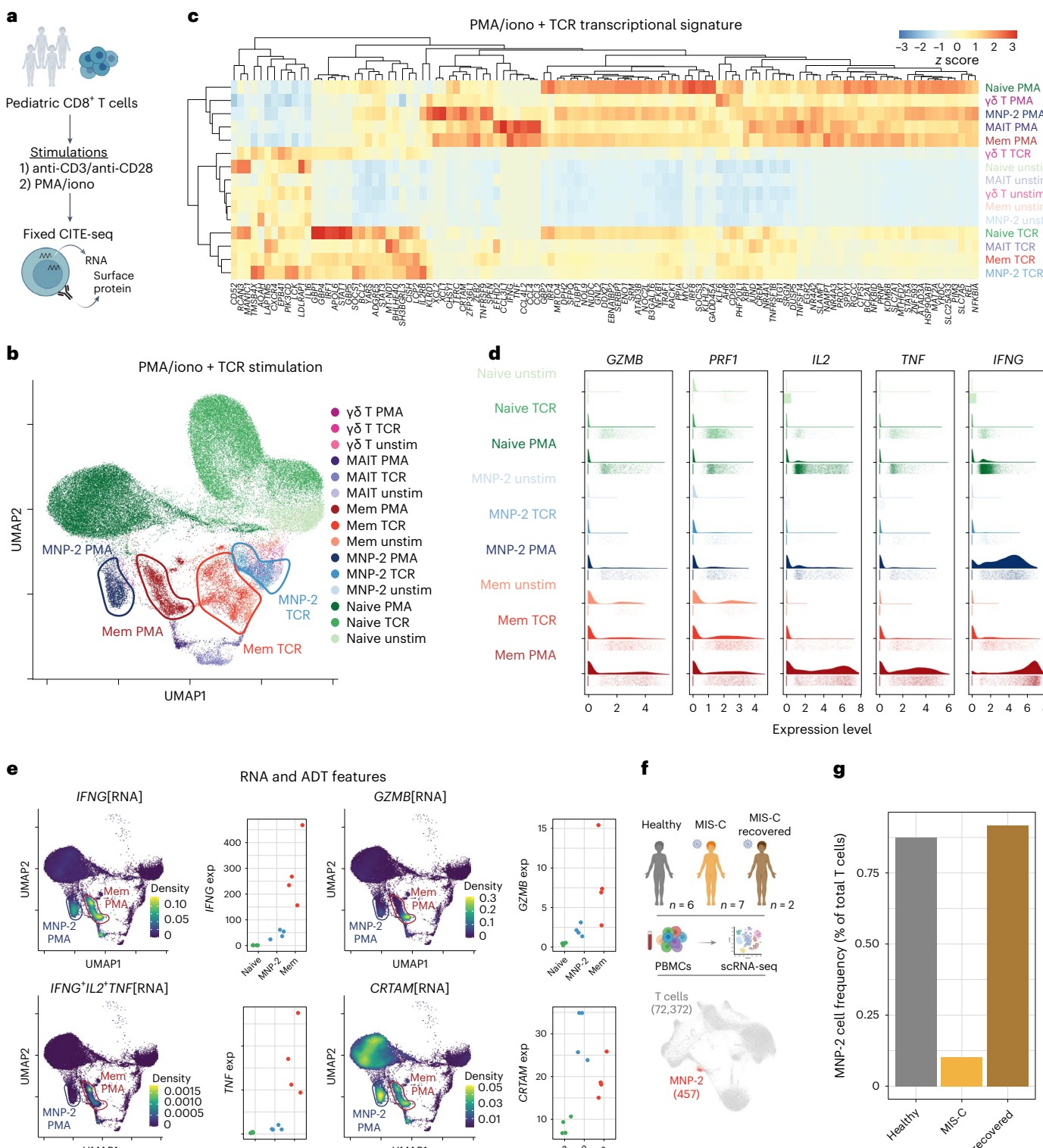

**Fig. 8 | Distinct effector responses in MNP-2 cells from children. a**, Overview of a fixed CITE-seq experiment (*n* = 4 pediatric donors) for TCR stimulation. **b**, RNA-based UMAP plot of unstimulated, anti-CD3/anti-CD28 (TCR; 0.5:1 beads per cell)-stimulated and PMA/iono (PMA 50 ng ml[−1], iono 1 μg ml[−1])-stimulated pediatric CD8[+] T cells. Stimulated subsets are indicated with the stimulation condition. **c**, Comparison of DEGs across each subset of unstimulated, TCR-stimulated and PMA/iono-stimulated CD8[+] T cells. **d**, Violin plots of the single-cell expression of select effector genes for naive, MNP-2 and memory CD8[+] T cells before and after stimulation with TCR and PMA/iono. **e**, Expression density of select RNA and ADT cell type markers, shown in UMAP plots of PMA/iono-stimulated and unstimulated cells. The modality of detection is indicated in square brackets. Density, gene-weighted 2D kernel density; exp, average expression level. **f**, Overview of an external pediatric MIS-C scRNA-seq dataset used for MNP-2 cell identification and frequency comparison. **g**, Frequency of MNP-2 cells in the total peripheral T cells of healthy children (*n* = 6), children with active MIS-C (*n* = 7) and children who had recovered from MIS-C (*n* = 2).

cell infiltration, pseudodifferentiation and clonal expansion occur[24]. Our multimodal analysis allows simultaneous analysis of composition, memory infiltration and pseudodifferentiation in the naive CD8[+] and CD4[+] T cell compartments of children compared to adults. We found that true naive CD4[+] T cells are the most affected by age, exhibiting distinct transcriptional and epigenetic programming in children and adults. Pediatric naive CD4[+] T cells are primarily present in a cellular state indicative of quiescence, whereas adult naive CD4[+] T cells are biased toward an activated state. The subtlety of this change in cell state, in the absence of major alterations in their 'naive' program, is similar to findings of recent studies in the field of stem cell aging, in which quiescent stem cells were found to shift into a more readily activated state upon bystander exposure to the aging microenvironment[25,26]. This cellular priming leads to reduced pluripotency in stem cells, suggesting that reprogramming of naive CD4[+] T cells across age may also affect their differentiation potential and be related to dysfunction noted in advanced aging[12].

This omics dataset also demonstrates that the differentiation-related transcription and epigenetic signatures found in previous bulk genomic studies of naive CD8[+] T cell aging[5] are consistent with the molecular profiles of age-expanded naive-like memory CD8[+] populations and in line with minimal evidence of pseudodifferentiation in highly purified naive CD8[+] T cells from young adults compared to those from older adults[27]. However, our data also reveal that memory T cell infiltration is not the sole driver of naive CD8[+] T cell aging but that a specific reorganization within the 'naive' CD8[+] T cell compartment occurs between childhood and adulthood. This reorganization is characterized by the 'loss' of a previously undescribed IL-21R[hi]CD244[hi]CD11b[hi] population of CD8αα[+] T cells in adults. Indeed, this unique MNP-2 subset composed <0.05% of the adult T cell compartment but was heterogeneous and exhibited a broad TCR repertoire in children—all factors that likely contributed to the lack of previous identification. MNP-2 cells also exhibit more stem-like features[28] with enrichment of naive TFs (for example, LEF1), distinguishing them from other types of unconventional CD8[+] T cells described in adults that expand during chronic viral infection, acute infection and/or autoimmunity and exhibit distinct phenotypes (for example, terminally differentiated, regulatory)[29–31].

The marked loss of MNP-2 cells in the periphery of children with active MIS-C suggests that these cells home to tissue sites during an active inflammatory response. Although they exhibit limited polyfunctionality, MNP-2 cells are poised to produce both IFNγ and perforin under specific stimulatory conditions; thus, they may contribute directly to local immune response within tissue sites[32]. Their tissue-homing properties may also explain their loss in the periphery with age, as thymic production wanes and low-grade tissue inflammation increases[33]. In advanced aging, the development of memory T cells is impaired, favoring effector cell generation[6,12]. Conversely, MNP-2 cells appear biased toward memory generation at the cost of superior effector functions, based on their high expression of BCL6 after stimulation[34]. Further studies into the antigen specificity and responses of this unconventional CD8αα[+] T cell population and its importance in tissue-specific immunity and resolution of inflammation across diverse pediatric populations are warranted.

Collectively, these experiments demonstrate a heterogeneous naive T cell compartment in humans, with the CD8[+] and CD4[+] T cell subsets differentially influenced by age. These variations may have translational implications in the context of infection, vaccination and therapeutic intervention, as overall T cell responses may differ between children and adults. We also demonstrated the potential of TEA-seq as a powerful discovery platform to further enhance our understanding of T cell subsets in many autoimmune and/or inflammatory disease states, such as rheumatoid arthritis, human immunodeficiency virus infection and obesity, to facilitate the identification of molecular drivers of T cell dysfunction for therapeutic targets.

## Online content

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

[1]Allen Institute for Immunology, Seattle, WA, USA. [2]Department of Systems Pharmacology and Translational Therapeutics, University of Pennsylvania School of Medicine, Philadelphia, PA, USA. [3]Immune Health, University of Pennsylvania Perelman School of Medicine, Philadelphia, PA, USA. [4]Center for Interventional Immunology, Benaroya Research Institute at Virginia Mason, Seattle, WA, USA. [5]Center for Translational Immunology, Benaroya Research Institute at Virginia Mason, Seattle, WA, USA. [6]Institute for Immunology, University of Pennsylvania Perelman School of Medicine, Philadelphia, PA, USA. [7]Department of Pediatrics, Children's Hospital of Philadelphia and the University of Pennsylvania Perelman School of Medicine, Philadelphia, PA, USA. [8]Present address: Department of Genome Sciences, University of Washington School of Medicine, Seattle, WA, USA. [9]Present address: Microbiology, Immunology and Cancer Biology (MICaB) Program, University of Minnesota, Minneapolis, Minneapolis, MN, USA. [10]Present address: Seagen, Bothell, WA, USA. [11]These authors contributed equally: Peter J. Skene, Claire E. Gustafson. ✉e-mail: peter.skene@alleninstitute.org; claire.gustafson@alleninstitute.org

## Methods

### Adult and pediatric cohorts

Cohort demographics are provided in Supplementary Table 3. Healthy 25- to 35-year-old and 55- to 65-year-old adult donors were recruited from the greater Seattle area as part of the Sound Life Project, a protocol approved by the institutional review board (IRB) of the Benaroya Research Institute. Donors were excluded from enrollment if they had a history of chronic disease, autoimmune disease, severe allergy or chronic infection. Meanwhile, healthy 11- to 13-year-old pediatric donors were recruited from the greater Philadelphia area under a protocol approved by the IRB of the Children's Hospital of Philadelphia. Donors were excluded from enrollment if they had a history of immune deficiency, fever or antibiotic use within the month before sample collection, chronic medication use, or a body mass index >2 s.d. above or below the mean for their age. All adult participants provided informed consent before participation. Informed consent for the participation of minors was obtained from a legally authorized representative of the child. If capable, the participating child also provided assent to participate in the study. All samples were collected, processed to PBMCs through a Ficoll-based approach and frozen in FBS with 10% dimethyl sulfoxide (DMSO) within 4 h of blood draw. Cord and peripheral blood samples for follow-up studies were purchased from Bloodworks Northwest and BioIVT, with written informed consent and approval by the Allen Institute IRB.

### TEA-seq

For TEA-seq experiments, eight pediatric and eight older adult female donors were selected (Fig. 1b). Half the pediatric and adult donors were CMV-positive based on testing in a Clinical Laboratory Improvement Amendments-approved laboratory. TEA-seq library preparation was performed as described previously[15], with the addition of Cell Hashing[36] to allow for sample multiplexing and limit well-to-well batch effects. In brief, samples were thawed and processed across three batches, with each batch containing a common PBMC control. Antibody staining for Cell Hashing and cell sorting was performed simultaneously on $2 \times 10^6$ cells from each sample. Each sample was incubated with a sample-specific barcoded TotalSeq-A antibody, anti-CD45 antibody and anti-CD3 antibody. The samples were then pooled by T cell proportions previously determined by flow cytometry, targeting 800,000 T cells for each donor sample and 200,000 T cells for the control, and sorted on a BD FACSAria Fusion flow cytometer (BD Biosciences). T cells were sorted as live single CD45$^+$CD3$^+$ cells; $2 \times 10^6$ sorted T cells were then used for library preparation. A panel of 55 target-specific barcoded oligonucleotide-conjugated antibodies (BioLegend TotalSeq-A) was used for these studies (Supplementary Table 3). Individual ATAC, RNA, hashtag oligonucleotide (HTO) and ADT libraries were prepared, sequenced and processed as described previously[15].

### TEA-seq data preprocessing

ADT and HTO count matrices were generated using BarCounter (v1.0) (refs. 37). The RNA and ADT count matrices were then combined into a single Seurat object. Cells were selected based on the following cutoffs: >250 genes per cell, >500 RNA unique molecular identifiers (UMIs) per cell, <10,000 ADT UMIs per cell, <35% mitochondrial reads and <20,000 RNA UMIs per cell. Normalization, feature selection and scaling were performed on the RNA matrix (Seurat SCTransform function, default settings), followed by principal component analysis (PCA; Seurat RunPCA function, default settings). A UMAP projection was generated (Seurat RunUMAP, dims = 1:30), and clustering was performed (Seurat FindNeighbors (dims = 1:30), followed by Seurat FindClusters (resolution = 0.5)). We used the Seurat Multimodal Reference Dataset for PBMCs (available from the Satija laboratory, New York Genome Center[38]) to perform label transfer on the dataset by using the functions described in the Seurat (v4) vignettes (Seurat FindTransferAnchors, followed by Seurat TransferData). Two clusters were identified to be non-T cells and excluded from downstream analysis. Sample-specific

transcripts, AC105402.3 and MTRNR2L8, were identified and removed before further downstream RNA analysis.

### ADT-based cell type identification

We used CD4, CD8, CD197, CD27 and CD45RA ADT markers to identify T cell subsets. For subset identification, each of the three batches was separated into its own Seurat object before analysis to account for differences in sequencing depth and average ADT UMIs per cell. ADTs were normalized and cells were identified based on the markers outlined in Fig. 1 and Supplementary Table 1 using Boolean gating.

### ADT, RNA and ATAC label transfers

RNA-based label transfer was performed using single-positive T cell subsets from the Seurat reference described above and using the Seurat functions FindTransferAnchors and TransferData. Label transfer from ATAC data was performed using the same reference, based on ArchR (v1.0.2) documentation (https://archrproject.com)[39]. A first round of unconstrained integration was performed, and cells were labeled based on the Seurat L1 cell types. The second round of labeling then used the constrained approach to transfer the L2 cell types within the groups identified in the L1 integration. To directly compare the results from both RNA and ATAC label transfers with our ADT-defined populations, select cell types were merged manually.

### TEA-seq T cell subset analyses

**3WNN clustering.** We performed PCA on both RNA and ADT count matrices and corrected for potential batch effects using Harmony (https://github.com/immunogenomics/harmony)[40]. For ATAC, a latent semantic indexing (LSI) embedding was calculated in ArchR (ArchR addIterativeLSI function, varFeatures = 75,000), and batch correction was performed (ArchR addHarmony function, groupBy = 'batch_id'). The corrected LSI embedding was transferred to the Seurat object for 3WNN integration and clustering on all Harmony-corrected principal components and LSI dimensions (Seurat FindMultiModalNeighbors function, dims.list = list(1:25, 1:20, 1:29) for RNA, ADT and ATAC, respectively).

**RNA modality analysis.** DEG analysis was performed with the hurdle model implemented in the MAST package[41]. $P$ values were adjusted for multiple comparisons using the Benjamini–Hochberg method[42]. $P_{adj} < 0.05$ and log(fold change) > 0.1 were considered significant.

**ATAC modality analysis.** LSI, clustering, group coverage computation, reproducible peak set annotation (MACS2), motif enrichment and ChromVar deviations enrichment were performed according to the ArchR documentation. The peak matrix was used to identify DAPs between groups. DAPs were used in motif enrichment analysis (ArchR peakAnnoEnrichment function, with cutoffs FDR ≤ 0.1 and log(fold change) ≥ 0.5).

### DEG pathway enrichment analysis

Pathway enrichment analysis was performed with GSEA[43] implemented in the fgsea package[44] to compare adult and pediatric donors and by CMV infection status. A custom collection of gene sets that included the Hallmark (v7.2) gene sets, Kyoto Encyclopedia of Genes and Genomes (v7.2) and Reactome (v7.2) from the Molecular Signatures Database (v4.0) was used as the pathway database, as previously described[45]. The pathway enrichment $P$ values were adjusted using the Benjamini–Hochberg method, and pathways with $P_{adj} < 0.05$ were considered significantly enriched.

### TF motif analysis

For each ADT-labeled cell type, age group (that is, children versus adults) and CMV infection status were compared to identify DAPs (ArchR getMarkerFeatures function). Motif enrichment

(ArchR peakAnnoEnrichment function) was then performed using the resulting DAPs with an FDR cutoff of ≤0.1 and a $\log_2$(fold change) cutoff of ≥0.5. Motifs for each cell type were then further filtered by an $\text{mlog}_{10}(P_{adj}) > 5$ cutoff and found to be differentially expressed in at least six of the cell types. As no enriched motifs were detected based on CMV infection status, no plots were generated for visualization.

### Naive CD4$^+$ and CD8$^+$ T cell subanalysis

We performed 3WNN clustering, as described above, for ADT-identified CD4$^+$ and CD8$^+$ naive T cells separately. Leiden clusters were then identified at multiple resolutions by varying the resolution parameter of the Seurat FindClusters function from 0.1 to 0.8 and were visualized using the Clustree package[46] (https://github.com/lazappi/clustree) to identify the optimal resolution. Marker genes for each cluster were then identified using Seurat's FindAllMarkers function. ATAC analysis was performed on the same separated populations, using the same approach described above, in ArchR.

### Flow cytometry

To assess T cell subset frequencies, PBMCs were analyzed using a 25-color T cell phenotyping flow cytometry panel (Supplementary Table 3), using a standardized method previously published[47]. Cells were analyzed on a five-laser Cytek Aurora spectral flow cytometer. Spectral unmixing was calculated with prerecorded reference controls using Cytek SpectroFlo software (v2.0.2). Cell types were quantified by traditional bivariate gating analysis performed with FlowJo cytometry software (v10.8).

### Power analysis for the confirmatory cohort

The appropriate sample size for the confirmatory cohort was determined according to the minimum sample size required to identify a 1% difference while controlling for type I and type II error rates of 0.05 or 0.02 with an estimated frequency s.d. of 0.45. This resulted in $n = 5$ per group for a two-sample $t$-test. Sample size correction based on the asymptotic relative efficiency of the Mann–Whitney $U$ test (that is, 15.7%) resulted in a minimum required sample size of $n = 6$ per group to identify a 1% difference at 80% power and control for type I and II error rates of 0.05 and 0.2, respectively. Sample size and power calculations do not cover hypotheses beyond the pediatric–older adult cohort comparison.

### Confirmatory cohort scRNA-seq

scRNA-seq was performed on PBMCs from 16 pediatric, 16 young adult and 16 older adult donors (Fig. 1b and Supplementary Table 3), as previously described[47]. In brief, scRNA-seq libraries were generated using a modified 10x Genomics Chromium 3′ single-cell gene expression assay with Cell Hashing. Eight donors were pooled per library, with the addition of a common batch control sample in each library. Libraries were sequenced on an Illumina NovaSeq platform. Hashed 10x Genomics scRNA-seq data processing was carried out using CellRanger (10x Genomics) and BarWare[37] to generate sample-specific output files. For scRNA-seq analysis, count matrices from each sample were merged into age-specific Seurat objects, followed by normalization, feature selection, scaling, PCA, UMAP embedding and clustering, as described above. Label transfer from the T cell fraction of the PBMC Seurat reference was performed for each age-specific dataset, as described above. Following label transfer, all objects were merged into a single dataset. Cells identified as naive CD4$^+$ T cells with a prediction score of >0.7 were retained for downstream analysis. We then averaged the expression from each cell in each age group (Seurat AverageExpression function, group.by = 'age') for DEGs identified by TEA-seq analysis for use in visualization.

### T cell subset sorting

T cells were directly isolated from peripheral or cord blood using the RosetteSep human T cell enrichment cocktail according to the manufacturer's protocol (Stem Cell Technologies). T cells were cryopreserved in 90% FBS plus 10% DMSO and stored in vapor-phase liquid nitrogen following isolation. Cryopreserved T cells were rapidly thawed and stained with the sorting antibody panel described in Supplementary Table 4. Naive CD4$^+$ T cells were sorted using the FACS-Melody cell sorter with FACSChorus (v2.0) software (BD Biosciences), according to the following phenotype: live, single, CD3$^+$CD8$^-$CD4$^+$CCR 7$^+$CD45RA$^+$CD27$^+$CD95$^-$ cells. A total of 500,000 cells per sample were then pelleted and snap-frozen in dry ice and ethanol for RNA isolation. For MNP-2 subset analysis, 5,000 cells each of MNP-2 and naive CD8$^+$ T cells were sorted, based on the CD244$^+$CD11b$^+$CD8$^+$CD4$^-$CD3$^+$TCRαβ$^+$ and CD244$^-$CD11b$^-$CD8$^+$CD4$^-$CD3$^+$TCRαβ$^+$ phenotypes, respectively, for RNA isolation.

### RNA extraction and qPCR

Total RNA was isolated using the RNeasy Plus mini or micro kit (Qiagen) according to the manufacturer's protocol. cDNA was generated using the SuperScript IV VILO Master Mix (Invitrogen). TaqMan probe sets (Supplementary Table 5) were used for qPCR using the TaqMan Fast Advanced Master Mix on the Bio-Rad CFX96 real-time instrument. All genes were normalized to the housekeeping gene *RPLP0*, and gene expression levels were compared using the $2^{(-\Delta Ct)}$ method.

### MNP-2 functional studies

PBMCs and cord blood mononuclear cells (CBMCs) were isolated from peripheral blood samples using standard Ficoll-Paque separation, cryopreserved in 90% FBS plus 10% DMSO and stored in vapor-phase liquid nitrogen. T cells were enriched from cord blood using the RosetteSep human T cell enrichment kit (Stem Cell Technologies).

### Naive CD8$^+$ T cell activation

CBMCs or enriched cord blood T cells were enriched for naive CD8$^+$ T cells using the Naive CD8$^+$ T Cell Isolation kit (Stem Cell Technologies) according to the manufacturer's protocol. Naive CD8$^+$ T cells were plated at 50,000 cells per well in 96-well round-bottom tissue culture plates (untreated) and stimulated with Dynabeads Human T Activator CD3/CD28 beads (0.5 beads per cell) for 1, 2, 3 and 7 days before collection and staining for flow cytometry with a T cell activation panel (Supplementary Table 4).

### CD8$^+$ T cell responses through CITE-seq

CD8$^+$ T cells were enriched from cryopreserved pediatric PBMCs (four female donors) (Supplementary Table 3) using the EasySep Human CD8$^+$ T Cell Enrichment Cocktail (Stem Cell Technologies) according to the manufacturer's protocol. Enriched CD8$^+$ T cells were plated at 200,000 cells per well in 96-well round-bottom tissue culture plates (untreated) and incubated for 4 h at 37 °C and 5% $CO_2$ in RPMI 1640 plus 10% FBS with medium alone, IL-21 (50 ng ml$^{-1}$), PMA/iono (PMA 50 ng ml$^{-1}$, iono 1 μg ml$^{-1}$) or Dynabeads Human T Activator CD3/CD28 (0.5 beads per cell, Thermo Fisher Scientific). After 4 h, cells were collected and stained using TotalSeq-B Human Universal Cocktail (BioLegend) following the manufacturer's protocol. After antibody staining, cells were fixed and quenched according to the 10x Genomics Fixation of Cells and Nuclei for Chromium Fixed RNA Profiling user guide. Cells were fixed for 16 h and 26 min at 4 °C. RNA was barcoded using the Fixed RNA Feature Barcode kit (10x Genomics). Quality control of prepared libraries for sequencing was performed by TapeStation (Agilent) analysis of 1:50 dilutions of each final library in Buffer EB (Qiagen). Libraries were quantified using the Quant-iT PicoGreen dsDNA Assay (Thermo Fisher Scientific). ADT and scRNA-seq gene expression libraries were sequenced using the NovaSeq S2 platform (Illumina) at read depths of 7,500 and 12,500 reads per cell, respectively. A PhiX control library was spiked in at 10%.

### Statistical analysis

Statistical analysis was done using GraphPad Prism 9 for macOS (v9.5.0) software. Two-tailed Mann–Whitney tests were used to compare two

groups. Two-tailed paired *t*-tests were used for within-donor comparisons of two populations. A two-tailed Mann–Whitney test with the Holm–Sidak multiple-comparison method was used to compare three or more groups. *P* values <0.05 were considered statistically significant. No data were excluded from analyses.

### Reporting summary
Further information on research design is available in the Nature Portfolio Reporting Summary linked to this article.

### Data availability
Raw data will be deposited in the National Center for Biotechnology Information (NCBI) Database of Genotypes and Phenotypes (dbGaP, study identifier phs003400.v1) for controlled access upon peer-reviewed publication. Processed data are deposited in the NCBI Gene Expression Omnibus (GEO) database (series accession no. GSE214546). The external cord blood (accession no. GSE157007) and pediatric MIS-C (accession no. GSE166489) datasets are from the GEO database. The thymus dataset is from ArrayExpress (accession no. E-MTAB-8581). A custom collection of gene sets that included the Hallmark (v7.2) gene sets, Kyoto Encyclopedia of Genes and Genomes (v7.2) and Reactome (v7.2) from the Molecular Signatures Database (v4.0) was used as the pathway database in GSEA analyses. Source data are provided with this paper.

### Code availability
Code used for analysis and figure generation in this paper is available on GitHub (https://github.com/aifimmunology/Aging_Tcell_TEA-seq). The TEA-seq visualization tool is directly accessible at https://explore.allen-immunology.org/explore/e53df468-4a8e-49a4-8b6d-525b0f9914ab.

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

## Acknowledgements
We thank the study participants and the clinical research teams at the University of Pennsylvania, Children's Hospital of Philadelphia and Benaroya Research Institute for their dedication to this project. We thank the Allen Institute founder, P.G. Allen, for his vision, encouragement and support. We also thank all members of the Allen Institute for Immunology, in particular the facilities and operations teams who helped maintain the productive research environment, as well as the Human Immune System Explorer (HISE) software development team for their constant support. Research reported in this publication was supported by National Institute on Aging award K01AG068373 (C.E.G.) and by the Allen Institute for Immunology. This research was conducted while C.E.G. was a Diamond/American Federation for Aging Research award recipient. Overview figures were created with BioRender.com. The content is solely the responsibility of the authors and does not necessarily represent the official views of the National Institutes of Health.

## Author contributions
Conceptualization: C.E.G., Z.T., E.S., S.R.Z., J.R., L.T.G., G.L.S., A.K.S., C.S., J.H.B., X.L., T.R.T., E.J.W., T.F.B., L.A.V., S.E.H., P.J.S. Methodology: C.E.G., Z.T., E.S., K.H., C.P., S.R.Z., M.-P.P., L.Y.O., A.T.H., C.R.R., V.H., M.W., P.C.G., J.R., J.R.G., S.M., L.T.G., S.V.V., G.L.S., A.K.S., T.R.T., T.F.B., L.A.V., S.E.H., P.J.S. Investigation: C.E.G., Z.T., Z.H., K.H., C.P., M.-P.P., L.Y.O., A.T.H., C.R.R., V.H., M.W., P.C.G., J.R., J.R.G., S.M., J.D., C.J.J., A.R.G., X.L., T.R.T., E.J.W., L.A.V., S.E.H., P.J.S. Visualization: C.E.G., Z.T., Z.H., K.H., M.-P.P., L.Y.O., T.R.T. Funding acquisition: C.E.G., L.A.B., T.R.T., T.F.B., P.J.S. Project administration: J.D., C.J.J., A.R.G., L.A.B., C.S., J.H.B., E.J.W., P.J.S. Supervision: C.E.G., Z.T., L.T.G., S.V.V., G.L.S., T.R.T., P.J.S. Writing—original draft: C.E.G., Z.T., Z.H., K.H., C.P., S.R.Z., P.J.S. Writing—review and editing: C.E.G., Z.T., Z.H., J.R., L.T.G., S.V.V., G.L.S., A.K.S., C.S., J.H.B., T.R.T., E.J.W., T.F.B., L.A.V., S.E.H., P.J.S.

## Competing interests
G.L.S. and S.V.V. are current employees of Seagen. All other authors declare that they have no competing interests.

## Additional information
**Extended data** is available for this paper at https://doi.org/10.1038/s41590-023-01641-8.

**Correspondence and requests for materials** should be addressed to Peter J. Skene or Claire E. Gustafson.

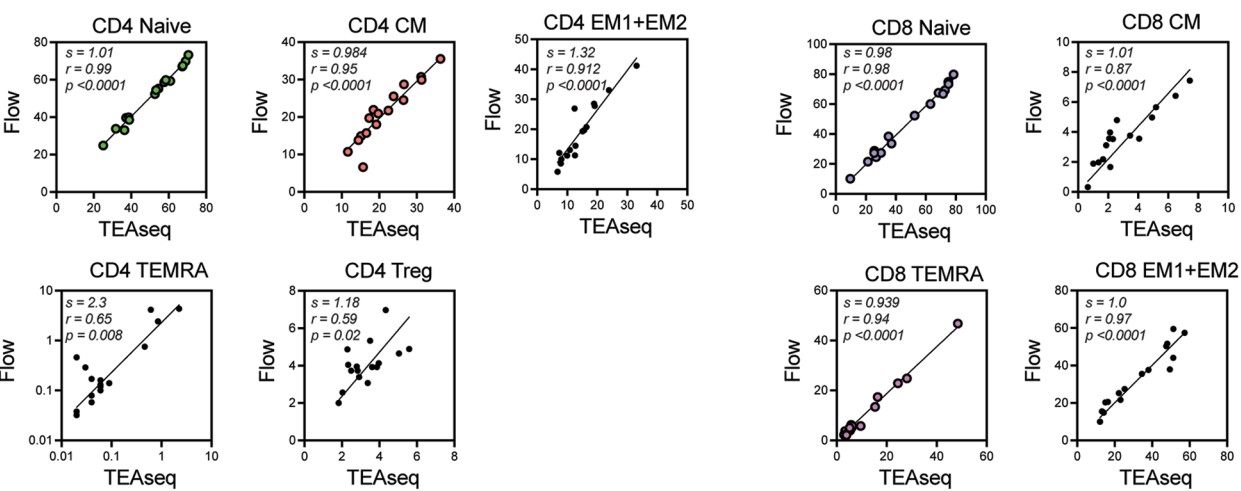

**Extended Data Fig. 1 | Defining T cell subsets based on surface proteins across different assays. (a)** Flow cytometry gating strategy for identifying T cell subsets. **(b)** Two-sided Spearman correlations for frequencies of T cell subsets within total T cells determined by either ADT-based gating in TEA-seq or by flow cytometry in the same donor samples. s, slope.

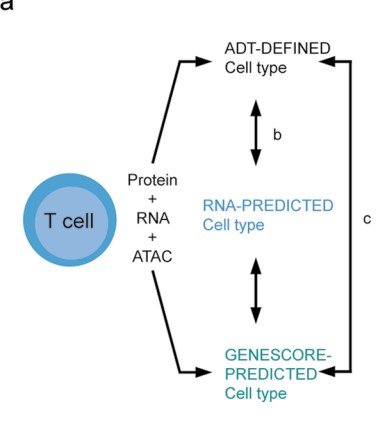

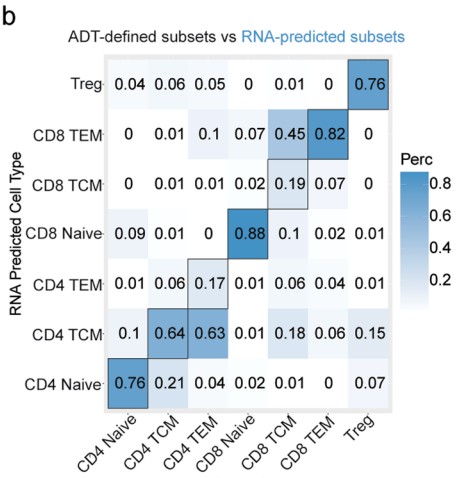

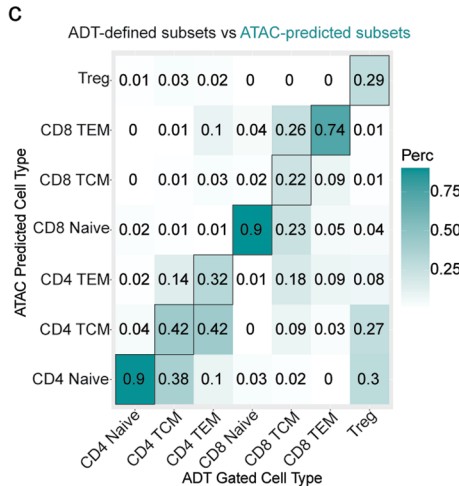

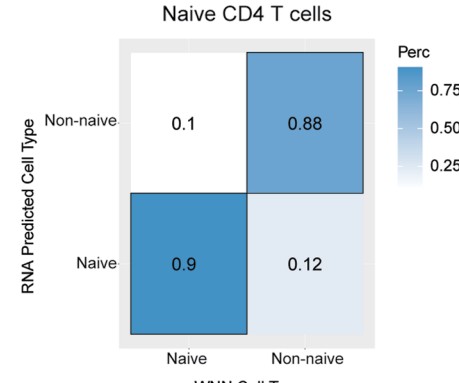

**Extended Data Fig. 2 | Comparative analysis of T cell subset definitions across individual modalities. (a)** Overview of single-cell labeling methods used for each TEA-seq modality (ADT, RNA, or ATAC). **(b)** Confusion plot comparison of T cell subset labels of single T cells between ADT-defined and Seurat RNA-prediction methods. **(c)** Confusion plot comparison of T cell subset labels of single T cells between ADT-defined and ATAC-prediction (ArchR) methods. **(d)** Confusion plot comparison of WNN labels and RNA-based label transfer of ADT-defined (CD45RA+CD197+CD27+) naive CD4 T cells.

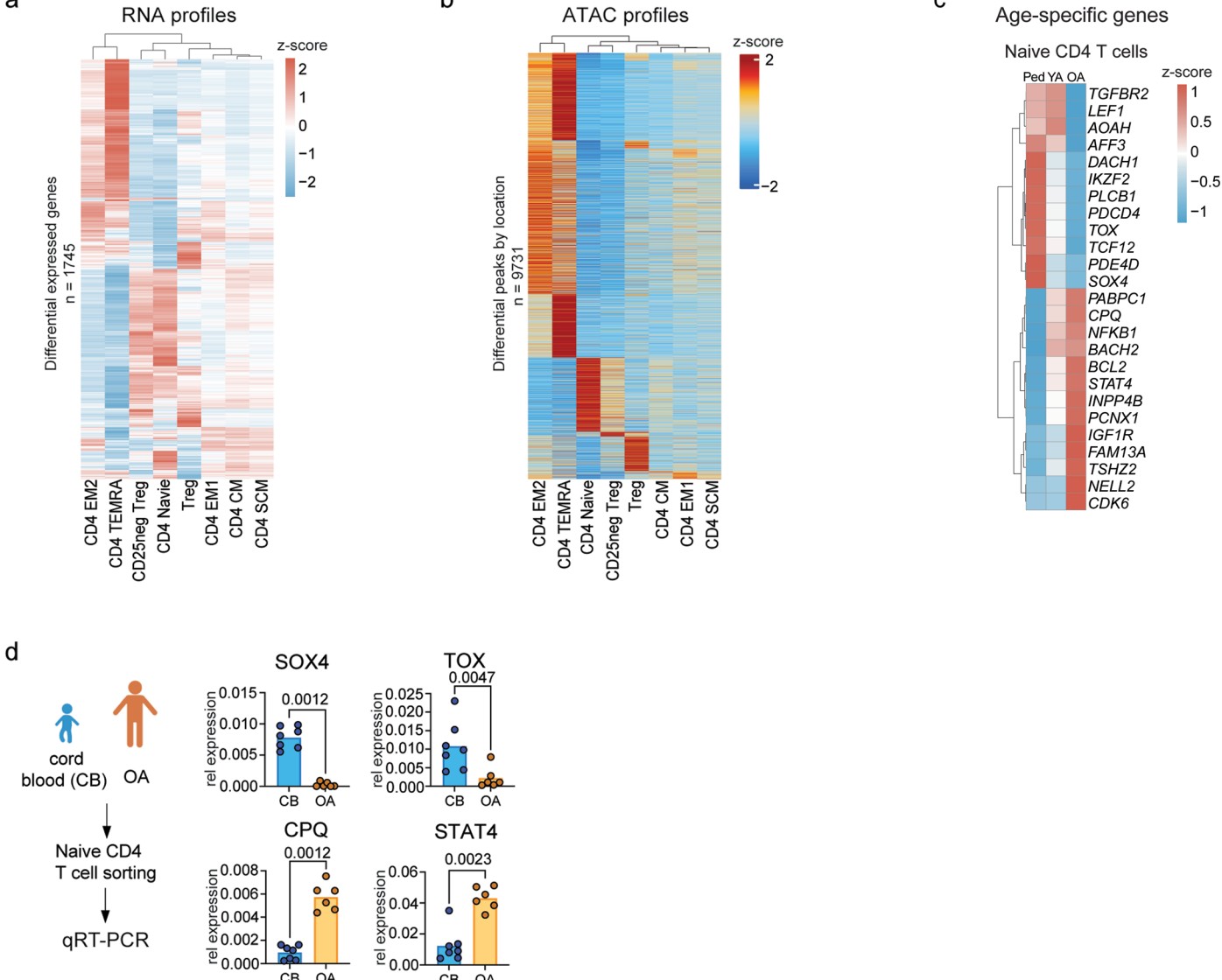

**Extended Data Fig. 3 | Comparison of transcriptional and epigenetic profiles of true naive CD4 T cells with CD4 T cell subsets and across age. (a)** Heatmap of all differentially expression genes (DEGs) using Seurat's FindAllMarkers function (parameters: logfc.threshold = 0.25, $P < 0.05$ determined by two-tailed Wilcoxon's rank sum test) between CD4 T cell subsets in TEA-seq dataset. **(b)** Heatmap of all differentially accessible peaks using ArchR's getMarkerFeatures (parameters: FDR <= 0.1, Log2FC ≥ 0.5) between CD4 T cell subsets in TEA-seq dataset. **(c)** Pseudo-bulk expression values from our confirmatory scRNA-seq dataset from pediatric (Ped), young adult (YA), and older adult naive CD4 T cells of select age-specific genes identified in TEA-seq. **(d)** Gene expression of SOX4, TOX, CPQ, and STAT4 in bulk-sorted naive CD4 T cells (CD3⁺CD4⁺CD8^neg CD45RA⁺CCR7⁺CD27⁺CD95^neg) from newborn cord blood (n = 7 donors) and older adult peripheral blood (n = 6 donors) using qRT-PCR. Two-tailed Mann-Whitney test. * $P < 0.05$, **** $P < 0.0001$. In panels a–c, values have been scaled (z-score) per gene or peak.

a

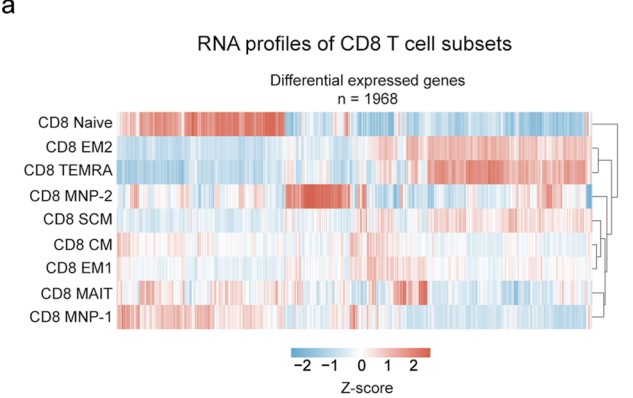

b

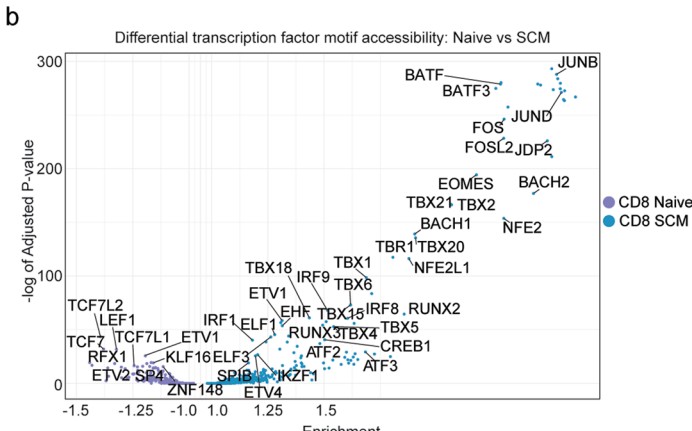

c

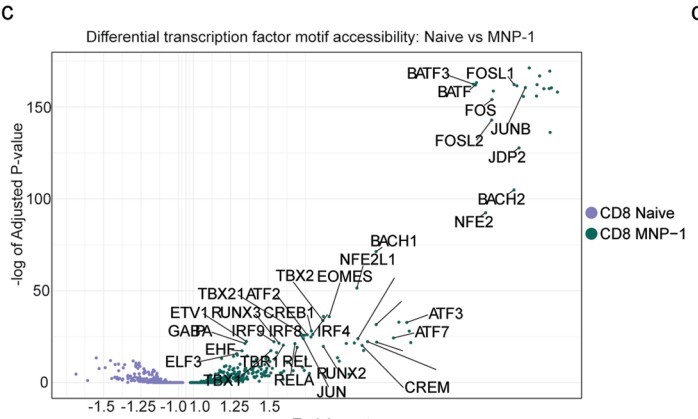

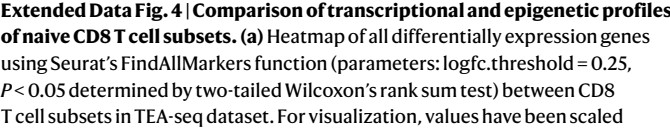

d

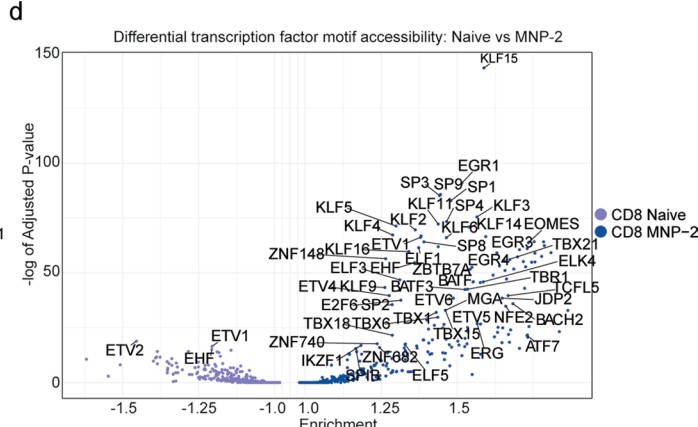

**Extended Data Fig. 4 | Comparison of transcriptional and epigenetic profiles of naive CD8 T cell subsets. (a)** Heatmap of all differentially expression genes using Seurat's FindAllMarkers function (parameters: logfc.threshold = 0.25, *P* < 0.05 determined by two-tailed Wilcoxon's rank sum test) between CD8 T cell subsets in TEA-seq dataset. For visualization, values have been scaled

(z-score) for each marker. **(b–d)** Transcription factor motif enrichment based on differentially accessible peaks between **(b)** true naive versus SCM, **(c)** true naive vs MNP-1, and **(d)** true naive versus MNP-2 CD8 T cell subsets. The adjusted pval of enrichment determined by hypergeometric testing.

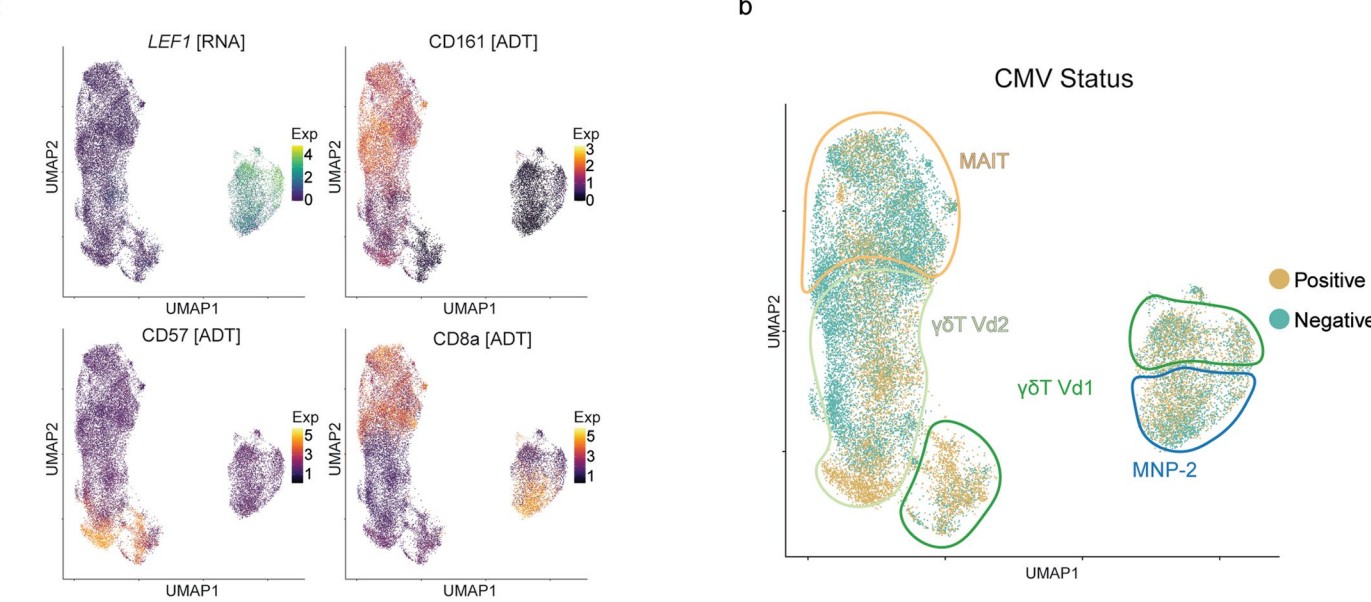

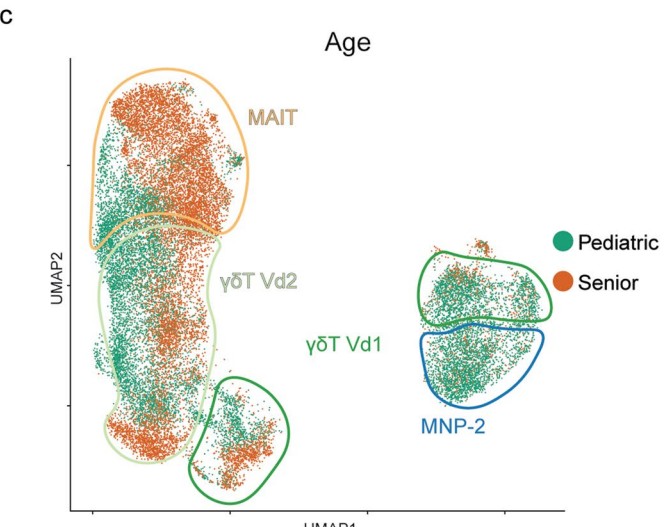

**Extended Data Fig. 5 | Comparison of MNP-2, MAIT and gdT cell subsets. (a)** Protein and RNA markers of interest on integrated 3WNN UMAP of innate-like T cell subsets and MNP-2 cells. **(b-c)** Distribution of innate-like T cell subsets and MNP-2 cells by **(b)** CMV infection status and **(c)** age. Each specific cell subset is labeled.

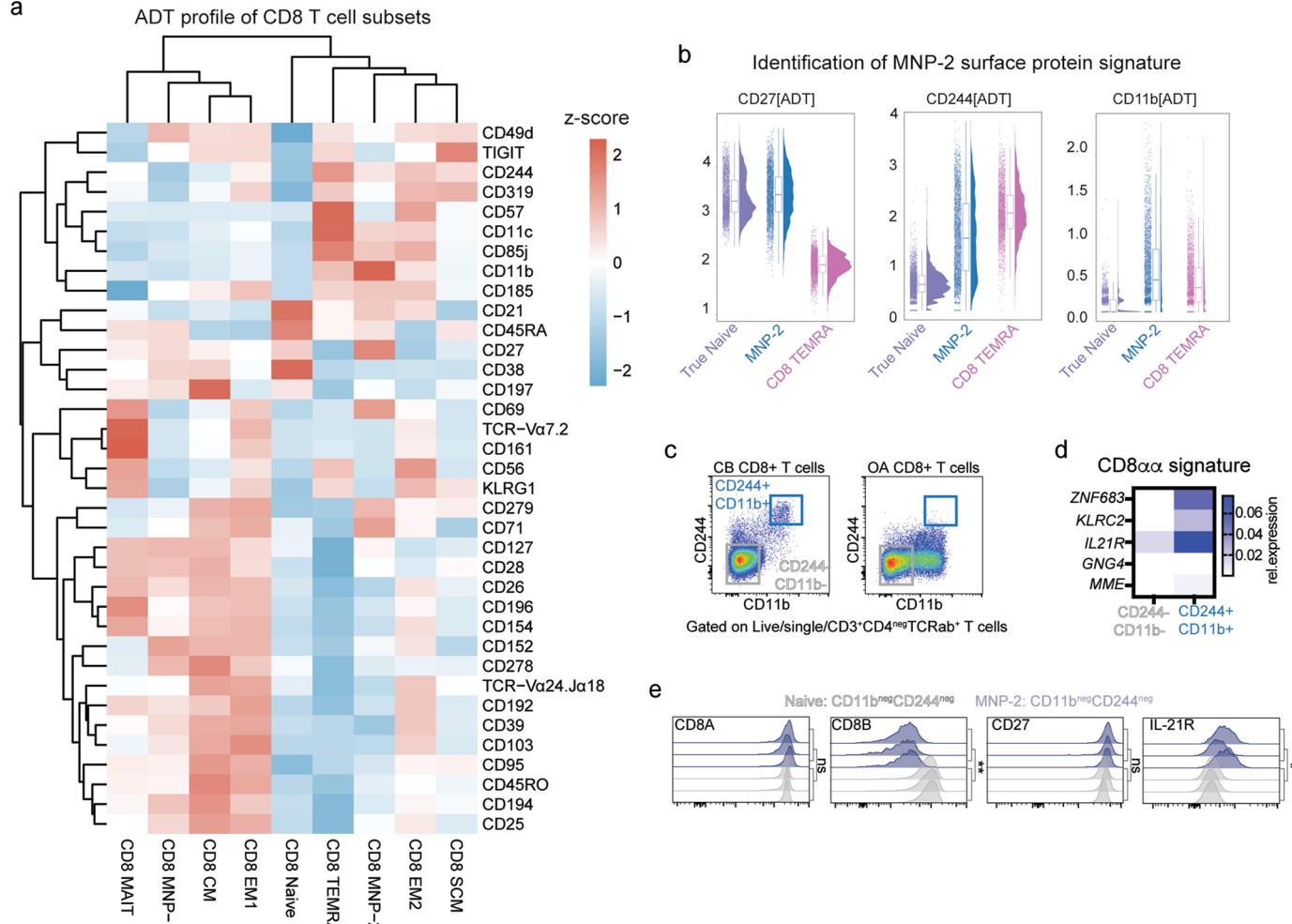

**Extended Data Fig. 6 | ADT expression profiles of MNP-2 CD8 T cells.**
(**a**) Expression heatmap of 36 ADTs across all CD8 T cell subsets in our TEA-seq dataset. Lineage markers CD3, CD4, CD8a, and CD16 were excluded from analysis. For visualization, values have been scaled (z-score) for each marker. (**b**) Single cell expression of surface protein ADTs CD27, CD244, and CD11b on MNP-2, true naive and TEMRA CD8 T cell subsets pooled from all donors (n = 16) in our TEA-seq dataset. Box plots are 25% and 75% quartiles with median shown. **c**) Protein expression of CD244 and CD11b on TCRab⁺ CD8⁺CD4ⁿᵉᵍ T cells

in cord blood (CB) and older adult (OA) PBMCs determined by flow cytometry. (**d**) Average RNA expression of CD8aa-specific genes in CD244⁺CD11b⁺ and CD244ⁿᵉᵍCD11bⁿᵉᵍ populations of TCRab⁺CD8⁺CD4ⁿᵉᵍ T cells sorted from cord blood (n = 4 donors) determined by qRT-PCR. (**e**) Surface protein expression of CD8A, CD8B, CD27, and IL-21R on CD244+CD11b+ and CD244ⁿᵉᵍCD11bⁿᵉᵍ populations of TCRab⁺ CD8⁺CD4ⁿᵉᵍ T cells from cord blood (n = 3 donors). *P*-values were determined by two-tailed paired *t*-test. *P < 0.05 (p = 0.028), **P < 0.01 (P = 0.0016), ns = not significant.

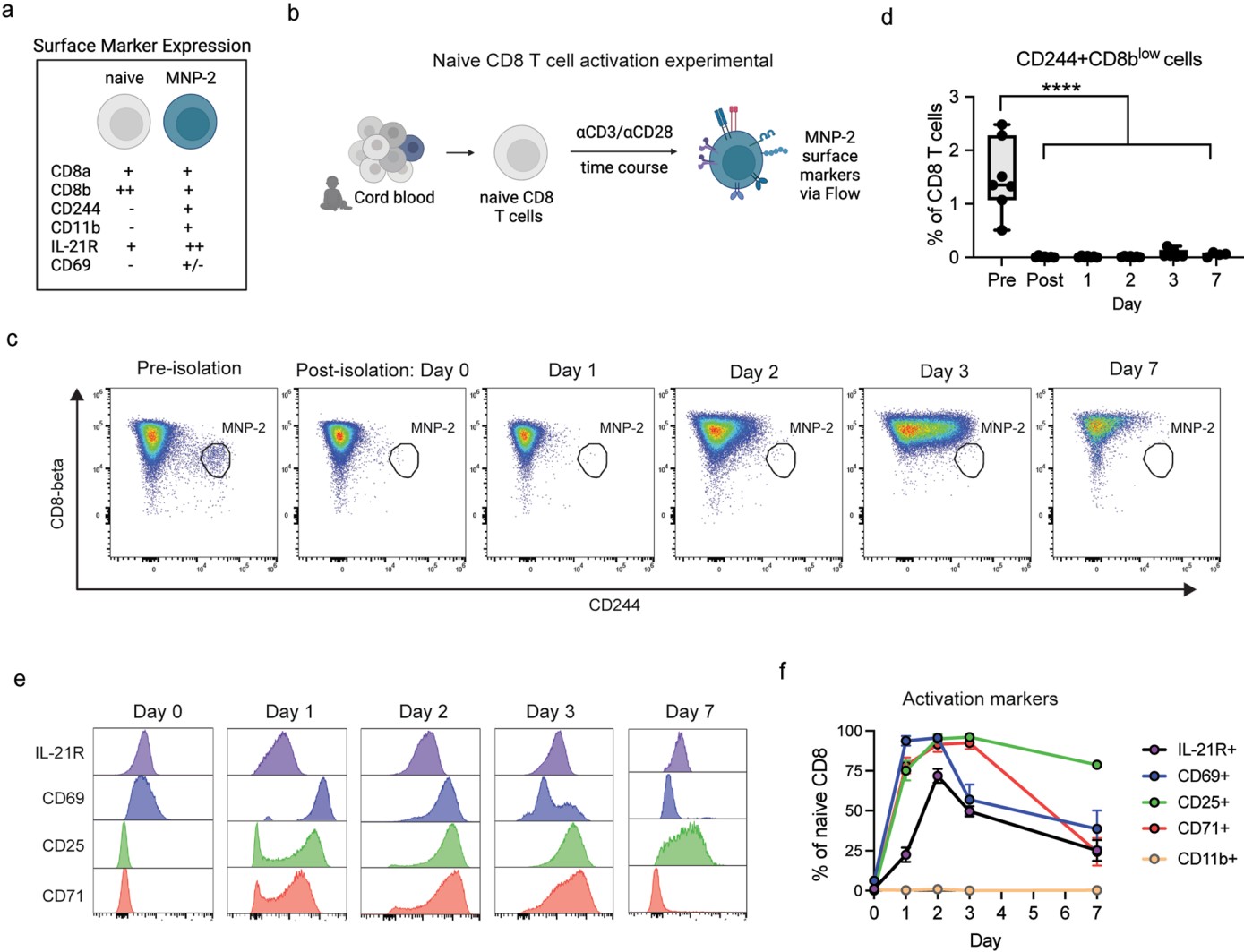

**Extended Data Fig. 7 | No expression of MNP-2 surface phenotype with naive CD8 T cell activation. (a)** Surface protein marker profiles of naive and MNP-2 CD8 T cells. **(b)** Naive CD8 T cell activation experiment with TCR stimulation (anti-CD3/anti-CD28 beads (0.5 beads per cell)). Cells were assessed over a 7-day time course for MNP-2 surface markers (CD8-beta, CD244, CD11b, IL21R) and activation markers (CD69, CD25, CD71) by flow cytometry. **(c)** Representative plot of MNP-2 cells delineated by CD8-beta$^{Low}$ and CD244$^{high}$ co-expression pre- and post-isolation, as well as over a 7-day time course post-TCR stimulation.

**(d)** Frequencies of CD8-beta$^{Low}$CD244$^{high}$ MNP-2 cells over the TCR stimulation time course (n = 4–7 donors). Results are from three independent experiments. $P$-values were determined by one-way ANOVA with Holm-Sidak's multiple comparisons test. ****$P < 0.0001$. Tukey's box plots with median with 1$^{st}$ and 3$^{rd}$ quartiles shown. **(e)** Representative histograms of activation markers and **(f)** frequencies of positive cells for each activation marker over the TCR stimulation time course (n = 4 donors). Mean +/− sem.

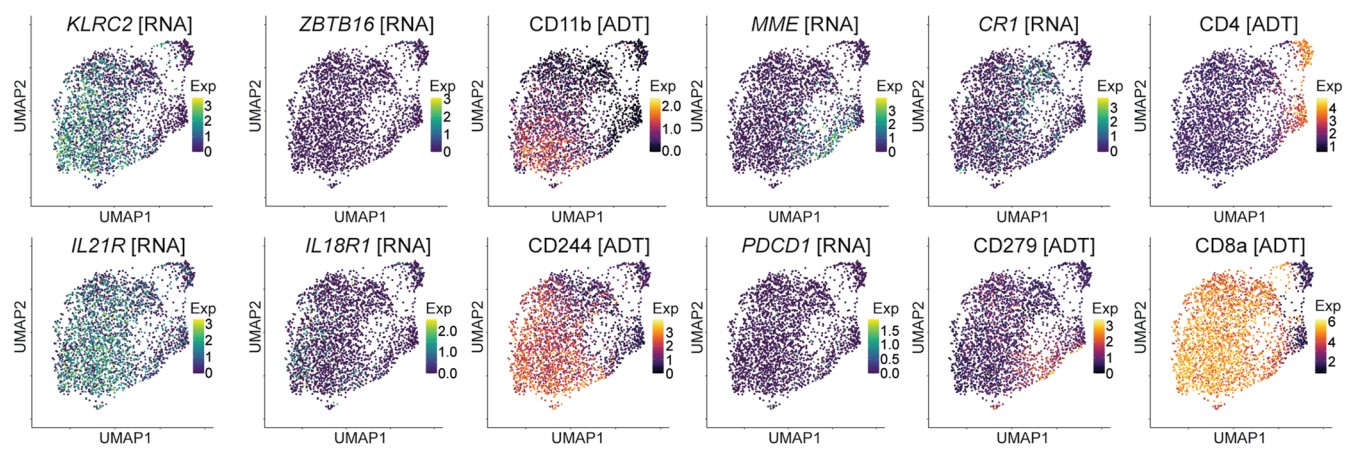

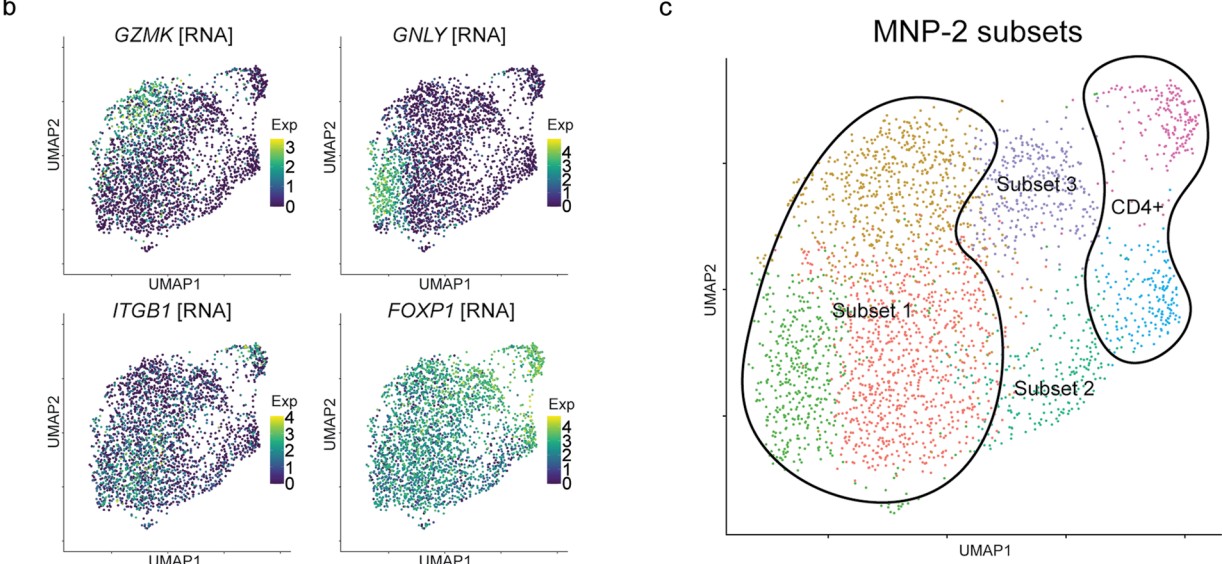

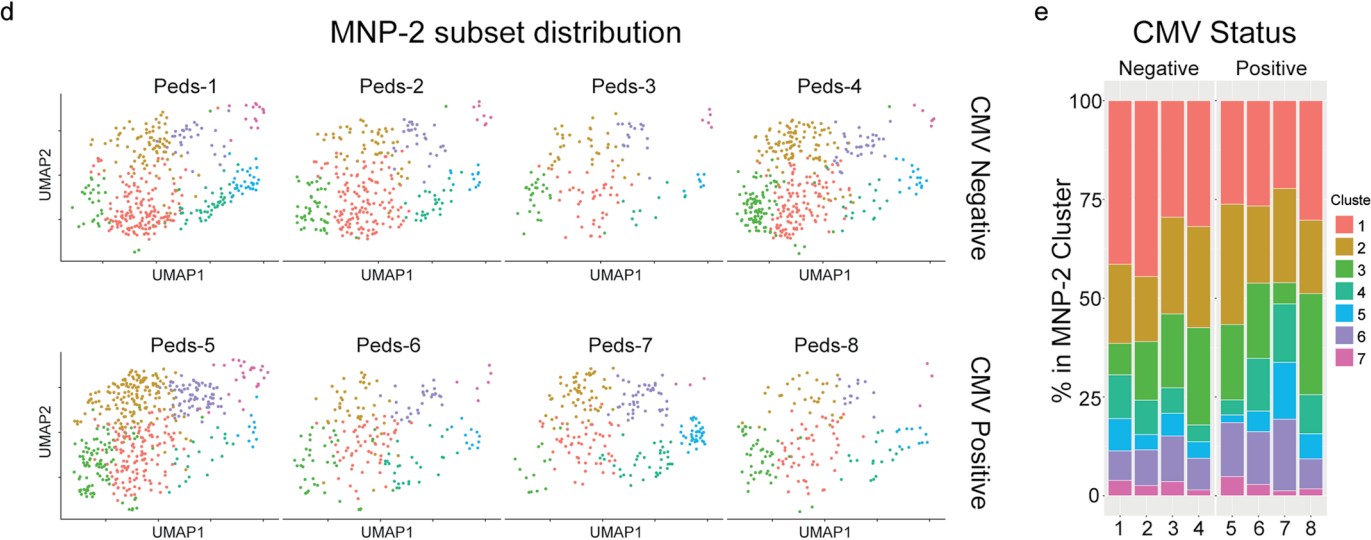

**Extended Data Fig. 8 | Heterogeneity within MNP-2 cells in children.**
(**a–b**) Single-cell RNA and ADT expression of (**a**) MNP-2 subset and (**b**) state-specific markers in total MNP-2 cells on 3WNN UMAP. (**c**) Different MNP-2 subsets identified in 3WNN clustering. (**d**) Distribution of MNP-2 cells with 3WNN UMAP for each individual pediatric donor. Top row is CMV-negative donors, bottom row is CMV-positive donors. (**e**) Proportion of each identified cluster within MNP-2 cells in children, separated by CMV infection status (n = 8 total donors).

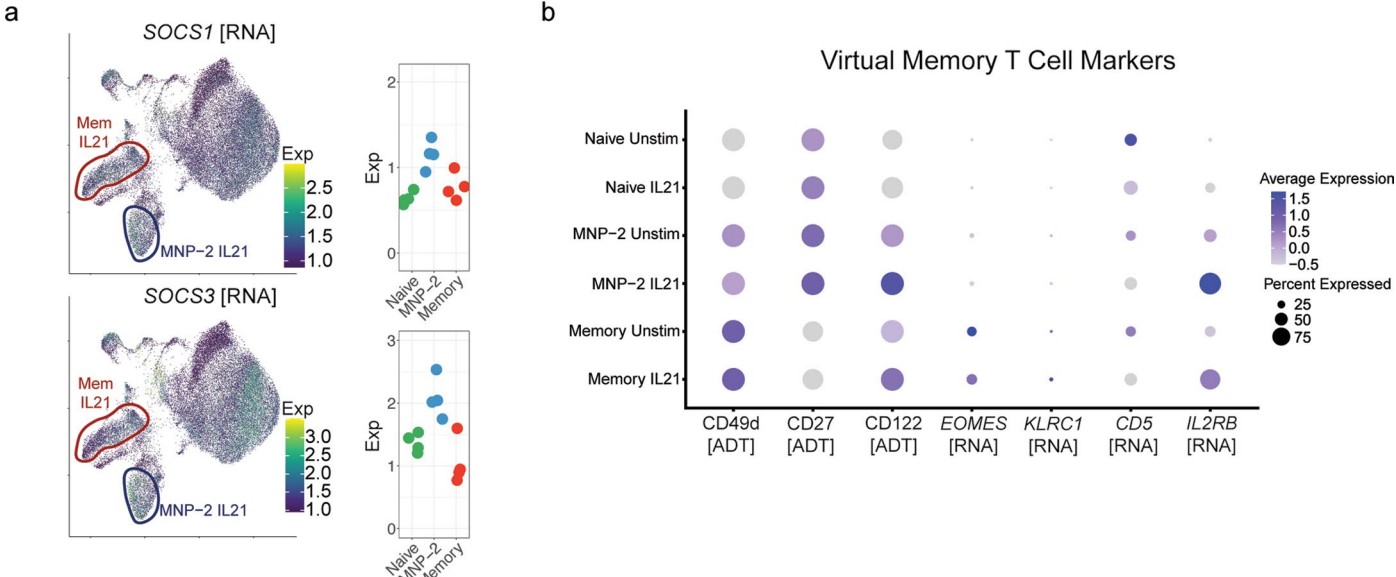

**Extended Data Fig. 9 | Pediatric CD8 T cell responses to IL-21 stimulation. (a)** Select gene expression from IL-21R signaling pathway in IL-21 stimulation UMAP. **(b)** Violin plots of RNA and ADT expression of virtual memory CD8 T cell markers on naïve, MNP-2 and memory CD8 T cells pre- and post-IL-21 stimulation.

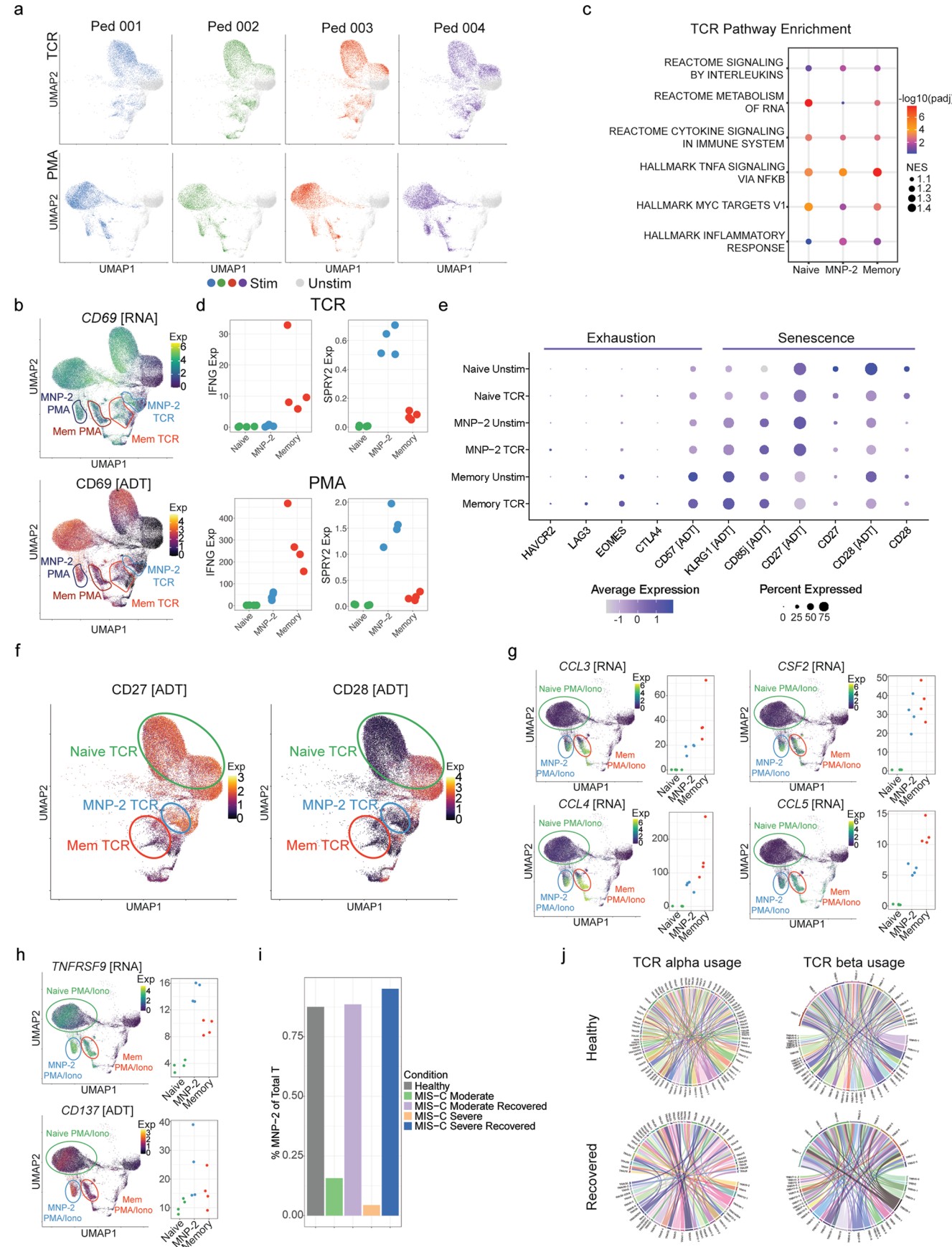

**Extended Data Fig. 10 | See next page for caption.**

**Extended Data Fig. 10 | Profiling MNP-2 responses to TCR and PMA/iono stimulation. (a)** RNA-based UMAP of TCR (upper row) and PMA/ionomycin (lower row) stimulations for each of the 4 pediatric donors. **(b)** RNA-based UMAP with CD69 gene and ADT expression in unstimulated and 4h TCR (aCD3/aCD28 beads; 0.5:1 beads per cell) and 4h PMA/iono stimulated pediatric CD8 T cells (n = 4 donors). Subsets with stimulation condition are circled. **(c)** GSEA analysis in naive, MNP-2, and memory CD8 T cell subsets comparing TCR stimulated versus unstimulated conditions. FDR < 0.05 was considered significant in the fgsea analysis. Dot size corresponds to the percent of genes that showed enrichment for the indicated pathway and cell type. Dot color corresponds to the normalized enrichment score (NES). **(d)** Pseudobulk RNA expression in naive, MNP-2, and memory CD8 T cells of *IFNG* and *SPRY2* post-TCR or post-PMA/iono stimulation. **(e)** Dot plot of exhaustion and senescence-related gene and protein (ADT)

expression profiles in naive, MNP-2, and memory CD8 T cells pre- and post-TCR stimulation. **(f)** CD27 and CD28 surface protein expression in unstimulated and TCR stimulated pediatric CD8 T cells on RNA-based UMAP. **(g)** Effector gene profile (*CCL3, CCL4, CCL5, CSF2*) and **(h)** co-stimulatory receptor (CD137[ADT], *TNFRSF9*[RNA]) expression in unstimulated and PMA (50 ng/ml) plus ionomycin (1 μg/ml) (PMA/iono) stimulated pediatric CD8 T cells on RNA-based UMAP. Pseudobulk RNA expression in naive, MNP-2, and memory CD8 T cells is shown to the right of each UMAP. **(i)** Frequency of MNP-2 cells in total peripheral T cells in healthy (n = 6), moderate MIS-C (n = 2), recovered moderate MIS-C (n = 1), severe MIS-C (n = 5), recovered severe MIS-C (n = 1) children from an external dataset (GSE166489)). **(j)** TCR alpha and beta V-J gene usage in MNP-2 cells in healthy (*TRA*: 153 cells, *TRB*: 208 cells) and recovered MIS-C (*TRA*: 67 cells, *TRB*: 84 cells) children.

# Reporting Summary

## Statistics

For all statistical analyses, confirm that the following items are present in the figure legend, table legend, main text, or Methods section.

| n/a | Confirmed | |
|---|---|---|
| ☐ | ☒ | The exact sample size (*n*) for each experimental group/condition, given as a discrete number and unit of measurement |
| ☐ | ☒ | A statement on whether measurements were taken from distinct samples or whether the same sample was measured repeatedly |
| ☐ | ☒ | The statistical test(s) used AND whether they are one- or two-sided<br>*Only common tests should be described solely by name; describe more complex techniques in the Methods section.* |
| ☐ | ☒ | A description of all covariates tested |
| ☐ | ☒ | A description of any assumptions or corrections, such as tests of normality and adjustment for multiple comparisons |
| ☐ | ☒ | A full description of the statistical parameters including central tendency (e.g. means) or other basic estimates (e.g. regression coefficient) AND variation (e.g. standard deviation) or associated estimates of uncertainty (e.g. confidence intervals) |
| ☐ | ☒ | For null hypothesis testing, the test statistic (e.g. *F*, *t*, *r*) with confidence intervals, effect sizes, degrees of freedom and *P* value noted<br>*Give P values as exact values whenever suitable.* |
| ☒ | ☐ | For Bayesian analysis, information on the choice of priors and Markov chain Monte Carlo settings |
| ☒ | ☐ | For hierarchical and complex designs, identification of the appropriate level for tests and full reporting of outcomes |
| ☒ | ☐ | Estimates of effect sizes (e.g. Cohen's *d*, Pearson's *r*), indicating how they were calculated |

*Our web collection on statistics for biologists contains articles on many of the points above.*

## Software and code

Policy information about availability of computer code

| Data collection | Cytek SpectroFlo software (Version 2.0.2), BD FACSChorus (v2) |
|---|---|
| Data analysis | FlowJo (V10.8), Graphpad Prism 9 for macOS (v9.5.0), R 4.1.2, CellRanger 7.1, Cell Ranger Arc 1.0.1, Seurat 4.1.0, ggplot2 3.4.0, Matrix 1.4-0, rhdf5 2.38.0, H5weaver 1.2.0, dplyr 1.0.8, viridis 0.6.2, harmony 0.1.0, Nebulosa 1.4.0, GenomicRanges 1.46.1, SummarizedExperiment 1.24.0, ArchR 1.0.2, ggrepel 0.9.1, scales 1.2.1, tidyverse 1.3.1, tidyr 1.2.0, plyranges 1.14.0 TxDb.Hsapiens.USCS.hg38.refGene 3.13.0, org.Hs.eg.db 3.14.0, OrganismDbi 1.36.0, ggbio 1.42.0, caret 6.0-93, ggdist 3.2.0, gghalves 0.1.4, ggraph 2.1.0, clustree 0.5.0, ggpmisc 0.5.2, UpSetR 1.4.0, doParallel 1.0.17, stringr 1.4.0. Code used for analysis and figure generation in this manuscript is available on Github (https://github.com/aifimmunology/Aging_Tcell_TEA-seq). |

For manuscripts utilizing custom algorithms or software that are central to the research but not yet described in published literature, software must be made available to editors and reviewers. We strongly encourage code deposition in a community repository (e.g. GitHub). See the Nature Portfolio guidelines for submitting code & software for further information.

## Data

Policy information about **availability of data**

All manuscripts must include a **data availability statement**. This statement should provide the following information, where applicable:
- Accession codes, unique identifiers, or web links for publicly available datasets
- A description of any restrictions on data availability
- For clinical datasets or third party data, please ensure that the statement adheres to our **policy**

Raw data is deposited in the NCBI Database of Genotypes and Phenotypes (dbGaP, study ID: phs003400.v1) for controlled access. Processed data is deposited in the NCBI Gene Expression Omnibus database (GEO, Series Accession ID: GSE214546). External cord blood (ID: GSE157007) and pediatric MIS-C (ID: GSE166489) datasets are from the GEO database. Thymus dataset is from Array Express (accession #E-MTAB-8581). A custom collection of genesets that included the Hallmark v7.2 genesets, KEGG v7.2, and Reactome v7.2 from the Molecular Signatures Database (MSigDB, v4.0) was used as the pathway database in GSEA analyses.

## Human research participants

Policy information about **studies involving human research participants and Sex and Gender in Research.**

| | |
|---|---|
| Reporting on sex and gender | Our initial TEA-seq of 16 people studied only women, in order to reduce data variation in our cohort of 8 pediatric and 8 adult donors. However, in order to expand these data, we selected a followup cohort for scRNA-seq of 48 individuals with 24 female and 24 male subjects, with each age group equally distributed by sex. Fixed scRNAseq for stimulation was performed on 4 female pediatric donors from the TEA-seq cohort. The cohort demographics are provided in Supplemental Table 3. |
| Population characteristics | Healthy pediatric (11-13yr olds), young adults (25-35 yrs) and older adults (55-65 yrs) were recruited for these studies. Relevant covariates include age, sex and CMV infection status and are provided in Figure 1b and Supplemental Table 3. |
| Recruitment | For sequencing studies: Adult: Healthy 25-35 year old and 55-65 year old adult subjects were recruited from the greater Seattle area as part of the Sound Life project at Benaroya Research Institute (BRI). Patients were excluded from enrollment if they had a history of chronic disease, autoimmune disease, severe allergy, or chronic infection. Subjects enrolled by BRI were compensated for their time, effort and incidental expenses related to the research visits with $50 per research visit that involved a blood draw. Pediatric: Healthy 11-13 year old pediatric subjects were recruited from the greater Philadelphia area. Patients were excluded from enrollment if they had a history of immune deficiency, fever or antibiotic usage within the month prior to sample collection, chronic medication usage or BMI more than 2 standard deviations above or below the mean for their age. Pediatric subjects enrolled were compensated for their time and effort. Payments were structured to increase as the longitudinal visits progressed and ranged from $25 to $100 per research visit that involved a blood draw. In addition, small thank-you token gifts were available for the pediatric subjects to choose from after each visit. |
| Ethics oversight | All studies were approved by Institutional Review Boards at Benaroya Research Institute (adult cohorts), University of Pennsylvania (pediatric cohort) and/or Allen Institute (all sample usage). All adult participants gave informed consent prior to participation in these studies. Informed consent for participation of minors was obtained from a legally authorized representative of the child. If capable, the participating child also provided assent to participate in the study. Cord and peripheral blood samples for follow-up studies were purchased from Bloodworks Northwest (Seattle, WA) and BioIVT (Hicksville, NY) obtained with written informed consent and use approved by Allen Institute IRB. |

Note that full information on the approval of the study protocol must also be provided in the manuscript.

# Field-specific reporting

Please select the one below that is the best fit for your research. If you are not sure, read the appropriate sections before making your selection.

☒ Life sciences        ☐ Behavioural & social sciences        ☐ Ecological, evolutionary & environmental sciences

For a reference copy of the document with all sections, see **nature.com/documents/nr-reporting-summary-flat.pdf**

# Life sciences study design

All studies must disclose on these points even when the disclosure is negative.

| | |
|---|---|
| Sample size | TEAseq experiments included 16 donors (8 per age group), cohort scRNASeq experiments included 48 donors (16 per age group) and stimulation scRNAseq included 4 donors (pediatric only). Sample size power calculation was performed for our larger validation cohort. The minimum sample size required to identify a 1% change while controlling for Type I and Type II errors at =0.05, = 0.2, respectively, and applying an estimated frequencies standard deviation of =0.45, is n=5 per group for a two-sample t-test. Applying a sample size correction based on the Asymptotic Relative Efficiency (ARE) of the Mann Whitney U test (i.e., 15.7%) results in a minimum required sample size of n=6 per group to identify 1% differences to attain 80% power and control for Type I and II error rates at =0.05, =0.2, respectively. Thus we exceed the minimum required n=6 per age group. |
| Data exclusions | No data were excluded from analyses. |

| | |
|---|---|
| Replication | All experiments have been biologically replicated in at least three donors and results were successfully reproduced. All sequencing was performed using a common PBMC batch control for technical replication confirmation and normalization. |
| Randomization | For the TEA-seq dataset, randomization of the groups was not possible since the study design was to compare donors based on specific clinical parameters; age group, and when appropriate, CMV infection status. However, sample were randomly distributed between batches of TEA-seq runs to mitigate assay variability. Samples used in scRNA-seq experiments were randomized across batches. Stimulation scRNA -seq experiments were performed as a single batch. |
| Blinding | Experiments and analyses were not performed blinded as the same investigator(s) oversaw the sample processing, data generation and data analyses. |

# Reporting for specific materials, systems and methods

We require information from authors about some types of materials, experimental systems and methods used in many studies. Here, indicate whether each material, system or method listed is relevant to your study. If you are not sure if a list item applies to your research, read the appropriate section before selecting a response.

## Materials & experimental systems

| n/a | Involved in the study |
|---|---|
| ☐ | ☒ Antibodies |
| ☒ | ☐ Eukaryotic cell lines |
| ☒ | ☐ Palaeontology and archaeology |
| ☒ | ☐ Animals and other organisms |
| ☒ | ☐ Clinical data |
| ☒ | ☐ Dual use research of concern |

## Methods

| n/a | Involved in the study |
|---|---|
| ☒ | ☐ ChIP-seq |
| ☐ | ☒ Flow cytometry |
| ☒ | ☐ MRI-based neuroimaging |

## Antibodies

| | |
|---|---|
| Antibodies used | TEA-seq panel
Antibody TotalSeq™-A0151 anti-human CD152 (CTLA-4) Antibody BNI3 BioLegend Cat# 369619, RRID:AB_2734423 (0.175 μg per million cells)
Antibody TotalSeq™-A0071 anti-human CD194 (CCR4) Antibody L291H4 BioLegend Cat# 359423, RRID:AB_2749979 (0.175 μg per million cells)
Antibody TotalSeq™-A0143 anti-human CD196 (CCR6) Antibody G034E3 BioLegend Cat# 353437, RRID:AB_2750534 (0.175 μg per million cells)
Antibody TotalSeq™-A0189 anti-human CD244 (2B4) Antibody C1.7 BioLegend Cat# 329527, RRID:AB_2750007 (0.175 μg per million cells)
Antibody TotalSeq™-A0396 anti-human CD26 Antibody BA5b BioLegend Cat# 302720, RRID:AB_2734261 (0.175 μg per million cells)
Antibody TotalSeq™-A0102 anti-human CD294 (CRTH2) Antibody BM16 BioLegend Cat# 350127, RRID:AB_2734360 (0.2 μg per million cells)
Antibody TotalSeq™-A0576 anti-human CD49d Antibody 9F10 BioLegend Cat# 304337, RRID:AB_2783166 (0.175 μg per million cells)
Antibody TotalSeq™-A0171 anti-human/mouse/rat CD278 (ICOS) Antibody C398.4A BioLegend Cat# 313555, RRID:AB_2800824 (0.05 μg per million cells)
Antibody TotalSeq™-A0161 anti-human CD11b Antibody ICRF44 BioLegend Cat# 301353, RRID:AB_2734249 (0.05 μg per million cells)
Antibody TotalSeq™-A0053 anti-human CD11c Antibody S-HCL-3 BioLegend Cat# 371519, RRID:AB_2749971 (0.025 μg per million cells)
Antibody TotalSeq™-A0390 anti-human CD127 (IL-7Rα) Antibody A019D5 BioLegend Cat# 351352, RRID:AB_2734366 (0.075 μg per million cells)
Antibody TotalSeq™-A0083 anti-human CD16 Antibody 3G8 BioLegend Cat# 302061, RRID:AB_2734255 (0.05 μg per million cells)
Antibody TotalSeq™-A0408 anti-human CD172a (SIRPα) Antibody 15-414 BioLegend Cat# 372109, RRID:AB_2783285 (0.25 μg per million cells)
Antibody TotalSeq™-A0144 anti-human CD185 (CXCR5) Antibody J252D4 BioLegend Cat# 356937, RRID:AB_2750356 (0.125 μg per million cells)
Antibody TotalSeq™-A0181 anti-human CD21 Antibody Bu32 BioLegend Cat# 354915, RRID:AB_2750006 (0.05 μg per million cells)
Antibody TotalSeq™-A0085 anti-human CD25 Antibody BC96 BioLegend Cat# 302643, RRID:AB_2734258 (0.08 μg per million cells)
Antibody TotalSeq™-A0154 anti-human CD27 Antibody O323 BioLegend Cat# 302847, RRID:AB_2750000 (0.05 μg per million cells)
Antibody TotalSeq™-A0088 anti-human CD279 (PD-1) Antibody EH12.2H7 BioLegend Cat# 329955, RRID:AB_2734322 (0.2 μg per million cells)
Antibody TotalSeq™-A0406 anti-human CD304 (Neuropilin-1) Antibody 12C2 BioLegend Cat# 354525, RRID:AB_2783261 (0.05 μg per million cells)
Antibody TotalSeq™-A0410 anti-human CD38 Antibody HB-7 BioLegend Cat# 356635, RRID:AB_2800967 (0.05 μg per million cells)
Antibody TotalSeq™-A0176 anti-human CD39 Antibody A1 BioLegend Cat# 328233, RRID:AB_2750005 (0.075 μg per million cells)
Antibody TotalSeq™-A0072 anti-human CD4 Antibody RPA-T4 BioLegend Cat# 300563, RRID:AB_2734247 (0.1 μg per million cells)
Antibody TotalSeq™-A0047 anti-human CD56 (NCAM) Antibody 5.1H11 BioLegend Cat# 362557, RRID:AB_2749970 (0.1 μg per million cells)
Antibody TotalSeq™-A0394 anti-human CD71 Antibody CY1G4 BioLegend Cat# 334123, RRID:AB_2800884 (0.05 μg per million cells)
Antibody TotalSeq™-A0080 anti-human CD8a Antibody RPA-T8 BioLegend Cat# 301067, RRID:AB_2734248 (0.2 μg per million cells)
Antibody TotalSeq™-A0006 anti-human CD86 Antibody IT2.2 BioLegend Cat# 305443, RRID:AB_2734273 (0.05 μg per million cells) |

Antibody TotalSeq™-A0581 anti-human TCR Vα7.2 Antibody 3C10 BioLegend Cat# 351733, RRID:AB_2783246 (0.0625 µg per million cells)

Antibody TotalSeq™-A0145 anti-human CD103 (Integrin αE) Antibody Ber-ACT8 BioLegend Cat# 350231, RRID:AB_2749996 (0.2 µg per million cells)

Antibody TotalSeq™-A0168 anti-human CD57 Recombinant Antibody QA17A04 BioLegend Cat# 393319, RRID:AB_2810588 (0.2 µg per million cells)

Antibody TotalSeq™-A0146 anti-human CD69 Antibody FN50 BioLegend Cat# 310947, RRID:AB_2749997 (0.2 µg per million cells)

Antibody TotalSeq™-A0242 anti-human CD192 (CCR2) Antibody K036C2 BioLegend Cat# 357229, RRID:AB_2750501 (0.25 µg per million cells)

Antibody TotalSeq™-A0063 anti-human CD45RA Antibody HI100 BioLegend Cat# 304157, RRID:AB_2734267 (0.25 µg per million cells)

Antibody TotalSeq™-A0156 anti-human CD95 (Fas) Antibody DX2 BioLegend Cat# 305649, RRID:AB_2750368 (0.25 µg per million cells)

Antibody TotalSeq™-A0159 anti-human HLA-DR Antibody L243 BioLegend Cat# 307659, RRID:AB_2750001 (0.05 µg per million cells)

Antibody TotalSeq™-A0153 anti-human KLRG1 (MAFA) Antibody SA231A2 BioLegend Cat# 367721, RRID:AB_2750373 (0.25 µg per million cells)

Antibody TotalSeq™-A0355 anti-human CD137 (4-1BB) Antibody 4B4-1 BioLegend Cat# 309835, RRID:AB_2783173 (0.25 µg per million cells)

Antibody TotalSeq™-A0149 anti-human CD161 Antibody HP-3G10 BioLegend Cat# 339945, RRID:AB_2749998 (0.1 µg per million cells)

Antibody TotalSeq™-A0140 anti-human CD183 (CXCR3) Antibody G025H7 BioLegend Cat# 353745, RRID:AB_2749993 (0.25 µg per million cells)

Antibody TotalSeq™-A0896 anti-human CD85j (ILT2) Antibody GHI/75 BioLegend Cat# 333723, RRID:AB_2814225 (0.1 µg per million cells)

Antibody TotalSeq™-A0179 anti-human CX3CR1 Antibody K0124E1 BioLegend Cat# 355709, RRID:AB_2832698 (0.1 µg per million cells)

Antibody TotalSeq™-A0169 anti-human CD366 (Tim-3) Antibody F38-2E2 BioLegend Cat# 345047, RRID:AB_2800924 (0.2 µg per million cells)

Antibody TotalSeq™-A0005 anti-human CD80 Antibody 2D10 BioLegend Cat# 305239, RRID:AB_2749958 (0.25 µg per million cells)

Antibody TotalSeq™-A0148 anti-human CD197 (CCR7) Antibody G043H7 BioLegend Cat# 353247, RRID:AB_2750357 (0.5 µg per million cells)

Antibody TotalSeq™-A0386 anti-human CD28 Antibody CD28.2 BioLegend Cat# 302955, RRID:AB_2783159 (0.5 µg per million cells)

Antibody TotalSeq™-A0031 anti-human CD40 Antibody 5C3 BioLegend Cat# 334346, RRID:AB_2749968 (0.375 µg per million cells)

Antibody TotalSeq™-A0087 anti-human CD45RO Antibody UCHL1 BioLegend Cat# 304255, RRID:AB_2734268 (0.5 µg per million cells)

Antibody TotalSeq™-A0224 anti-human TCR α/β Antibody IP26 BioLegend Cat# 306737, RRID:AB_2783167 (0.375 µg per million cells)

Antibody TotalSeq™-A0139 anti-human TCR γ/δ Antibody B1 BioLegend Cat# 331229, RRID:AB_2734325 (0.25 µg per million cells)

Antibody TotalSeq™-A0089 anti-human TIGIT (VSTM3) Antibody A15153G BioLegend Cat# 372725, RRID:AB_2734426 (0.5 µg per million cells)

Antibody TotalSeq™-A0158 anti-human CD134 (OX40) Antibody Ber-ACT35 BioLegend Cat# 350033, RRID:AB_2783245 (0.5 µg per million cells)

Antibody TotalSeq™-A0032 anti-human CD154 Antibody 24-31 BioLegend Cat# 310843, RRID:AB_2734283 (0.5 µg per million cells)

Antibody TotalSeq™-A0584 anti-human TCR Vα24-Jα18 (iNKT cell) Antibody 6B11 BioLegend Cat# 342923, RRID:AB_2783227 (0.5 µg per million cells)

Antibody TotalSeq™-A0180 anti-human CD24 Antibody ML5 BioLegend Cat# 311137, RRID:AB_2750374 (0.5 µg per million cells)

Antibody TotalSeq™-A0830 anti-human CD319 (CRACC) Antibody 162.1 BioLegend Cat# 331821, RRID:AB_2800872 (0.5 µg per million cells)

Antibody TotalSeq™-A0090 Mouse IgG1, κ isotype Ctrl Antibody MOPC-21 BioLegend Cat# 400199, RRID:AB_2868412 (0.5 µg per million cells)

Flow PBMC phenotyping panel

Antibody Mouse anti-human CD3/BUV395 UCHT1 BD Bioscience Cat# 563546, RRID:AB_2744387 2uL per sample

Antibody Mouse anti-human CD45/BUV496 HI30 BD Bioscience Cat# 624283 2uL per sample

Antibody Mouse anti-human CD8/BUV737 RPA-T8 BD Bioscience Cat# 624286 0.5uL per sample

Antibody Mouse anti-human CD127/BV711 A019D5 BioLegend Cat# 351328, RRID:AB_2562908 2uL per sample

Antibody Mouse anti-human CD197/PE-Cy7 G043H7 BioLegend Cat# 353226, RRID:AB_11126145 3uL per sample

Antibody Mouse anti-human CD14/BB660 MφP9 BD Bioscience Cat# 624295 0.5uL per sample

Antibody Mouse anti-human CD56/BUV563 NCAM16.2 BD Bioscience Cat# 612928 0.5uL per sample

Antibody Mouse anti-human CD19/BUV615 HIB19 BD Bioscience Cat# 624297 1uL per sample

Antibody Mouse anti-human CD27/BUV661 L128 BD Bioscience Cat# 624285 0.5uL per sample

Antibody Mouse anti-human CD39/BUV805 Tu66 BD Bioscience Cat# 624287 1uL per sample

Antibody Mouse anti-human CD103/BV421 Ber-ACT8 BioLegend Cat# 350214, RRID:AB_2563514 1uL per sample

Antibody Mouse anti-human abTCR/BV480 IP26 BD Bioscience Cat# 624278 3uL per sample

Antibody Mouse anti-human CD223/BV605 11C3C65 BioLegend Cat# 369324, RRID:AB_2721541 2uL per sample

Antibody Mouse anti-human CD95/BV650 DX2 BioLegend Cat# 305642, RRID:AB_2632622 1uL per sample

Antibody Mouse anti-human CD278/BV750 DX29 BD Bioscience Cat# 624380 2uL per sample

Antibody Mouse anti-human CD45RA/BV786 HI100 BioLegend Cat# 304140, RRID:AB_2563816 1uL per sample

Antibody Mouse anti-human CD185/BB515 RF8B2 BD Bioscience Cat# 564624, RRID:AB_2738871 2uL per sample

Antibody Mouse anti-human CD4/BB700 SK3 BD Bioscience Cat# 566392, RRID:AB_2744421 2uL per sample

Antibody Mouse anti-human HLA-DR/BB790 G46-6 BD Bioscience Cat# 624296 1uL per sample

Antibody Mouse anti-human CD279/PE EH12.2H7 BioLegend Cat# 329906, RRID:AB_940483 2uL per sample

Antibody Mouse anti-human TIGIT/PE-Dazzle594 A15153G BioLegend Cat# 372716, RRID:AB_2632931 4uL per sample

Antibody Mouse anti-human CD38/PE-Cy5 HIT2 BD Bioscience Cat# 555461, RRID:AB_395854 5uL per sample

Antibody Mouse anti-human CD69/APC FN50 BioLegend Cat# 310910, RRID:AB_314845 2uL per sample

Antibody Mouse anti-human CD25/APC-R700 2A3 BD Bioscience Cat# 565106, RRID:AB_2744339 2uL per sample

Antibody Mouse anti-human KLRG1/APC-Fire750 SA231A2 BioLegend Cat# 367718, RRID:AB_2687392 1uL per sample

Flow sorting panel (Naïve CD4 T cells)

Antibody Brilliant Violet 421™ anti-human CD95 (Fas) Antibody DX2 BioLegend Cat# 305624, RRID:AB_2561830 2.5uL per sample

Antibody FITC anti-human CD3 Antibody UCHT1 BioLegend Cat# 300406, RRID:AB_314060 0.5uL per sample

Antibody PerCP/Cyanine5.5 anti-human CD27 Antibody O323 BioLegend Cat# 302820, RRID:AB_2073318 2uL per sample
Antibody PE anti-human CD197 (CCR7) Antibody G043H7 BioLegend Cat# 353204 2uL per sample
Antibody PE-Cy™7 Mouse Anti-Human CD4 Antibody SK3 BD Bioscience Cat# 557852, RRID:AB_396897 2uL per sample
Antibody APC anti-human CD45RA Antibody HI100 BioLegend Cat# 304112, RRID:AB_314416 2uL per sample
Antibody APC/Cyanine7 anti-human CD8a Antibody RPA-T8 BioLegend Cat# 301016, RRID:AB_314134 2uL per sample
Flow sorting panel (Total T cells for TEA-seq)
Antibody PE anti-human CD3 Antibody UCHT1 BioLegend Cat# 300441, RRID:AB_2562047 1uL per sample
Antibody FITC anti-human CD45 Antibody HI30 BioLegend Cat# 304038, RRID:AB_2562050 1uL per sample
Flow sorting panel (MNP-2 population)
Antibody FITC anti-human CD3 Antibody UCHT1 BioLegend Cat# 300406, RRID:AB_2562047 0.5uL per sample
Antibody PE-Cy™7 Mouse Anti-Human CD4 Antibody SK3 BD Bioscience Cat# 557852, RRID:AB_396897 2uL per sample
Antibody APC/Cyanine7 anti-human CD8a Antibody RPA-T8 BioLegend Cat# 301016, RRID:AB_314134 2uL per sample
Antibody APC anti-human CD45RA Antibody HI100 BioLegend Cat# 304112, RRID:AB_314416 2uL per sample
Antibody BV786 anti-human abTCR Antibody IP26 BD Bioscience Cat# 306742, RRID:AB_2783171 2uL per sample
Antibody BV421 anti-human CD244 (2B4) Antibody C1.7 BioLegend Cat# 329532, RRID:AB_2814194 2.5uL per sample
Antibody PE anti-human CD11b Antibody ICRF44 BD Bioscience Cat# 555388, RRID:AB_395789 5uL per sample
Flow MNP-2 phenotyping panel
Antibody AF488 anti-human CD45RA HI100 BioLegend Cat# 304114, RRID:AB_528816 1.25uL per sample
Antibody Spark Blue 550 anti-human CD8a SK1 BioLegend Cat# 344760, RRID:AB_2819983 0.6uL per sample
Antibody BV650 anti-human CD4 SK3 BD Bioscience Cat# 563875, RRID:AB_2744425 0.6uL per sample
Antibody BV750 anti-human TCR alpha-beta IP26 BioLegend Cat# 306746, RRID:AB_2810463 2.5uL per sample
Antibody BV480 anti-human CD19 HIB19 BD Bioscience Cat# 746457, RRID:AB_2743759 0.6uL per sample
Antibody PE-Cy7 anti-human CD197 (CCR7) G043H7 BioLegend Cat# 353226, RRID:AB_11126145 2.5uL per sample
Antibody BV421 anti-human CD95 (Fas) DX2 BioLegend Cat# 305624, RRID:AB_2561830 2.5uL per sample
Antibody BUV395 anti-human CD27 L128 BD Bioscience Cat# 563815, RRID:AB_2744349 1.25uL per sample
Antibody BUV805 anti-human CD3 UCHT1 BD Bioscience Cat# 612895, RRID:AB_2870183 2.5uL per sample
Antibody BUV496 anti-human CD45 HI30 BD Bioscience Cat# 750179, RRID:AB_2868405 0.6uL per sample
Antibody PE-Dazzle594 anti-human CD244 (2B4) C1.7 BioLegend Cat# 329521, RRID:AB_2572018 2.5uL per sample
Antibody BUV661 anti-human CD11b ICRF44 BD Bioscience Cat# 741601, RRID:AB_2916939 5uL per sample
Antibody BV480 anti-human CD14 MOP9 BD Bioscience Cat# 566141, RRID:AB_2739539 0.6uL per sample
Antibody PE anti-human CD360 (IL-21R 17A12 BioLegend Cat# 359505, RRID:AB_2562368 5uL per sample
Antibody APC anti-human CD8b QA20A40 BioLegend Cat# 376705, RRID:AB_2910430 5uL per sample
Flow CD8 activation panel
Antibody BUV395 anti-human CD71 M-A712 BD Bioscience Cat# 568523, RRID:AB_2937039 2uL per sample
Antibody BUV496 anti-human CD69 FN50 BD Bioscience Cat# 750214, RRID:AB_2874415 2uL per sample
Antibody BUV805 anti-human CD3 UCHT1 BD Bioscience Cat# 612895, RRID:AB_2870184 1uL per sample
Antibody BV421 anti-human CD25 BC96 Biolegend Cat# 302630, RRID:AB_11126749 2uL per sample
Antibody BV480 anti-human CD19 HIB19 BD Bioscience Cat# 746457, RRID:AB_2743759 1uL per sample
Antibody BV480 anti-human CD14 MOP9 BD Bioscience Cat# 566141, RRID:AB_2739539 1uL per sample
Antibody BV650 anti-human CD4 SK3 BD Bioscience Cat# 563875, RRID:AB_2744425 1uL per sample
Antibody BV750 anti-human TCR αß IP26 Biolegend Cat# 306746, RRID:AB_2810463 1uL per sample
Antibody AF488 anti-human CD45RA HI100 Biolegend Cat# 304114, RRID:AB_528816 1uL per sample
Antibody PerCP-Cy5.5 anti-human CD11b M1/70 Biolegend Cat# 101228, RRID:AB_893232 5uL per sample
Antibody PE anti-human CD360 (IL-21R) 17A12 Biolegend Cat# 359506, RRID:AB_2562369 5uL per sample
Antibody PE-Dazzle594 anti-human CD244 (2B4) C1.7 Biolegend Cat# 329521, RRID:AB_2572018 2uL per sample
Antibody PE-Cy7 anti-human CD197 (CCR7) G043H7 Biolegend Cat# 353226, RRID:AB_11126145 2uL per sample
Antibody APC anti-human CD8b QA20A40 Biolegend Cat# 376706, RRID:AB_2937040 1uL per sample
Antibody APC-Cy7 anti-human CD8a RPA-T8 Biolegend Cat# 301016, RRID:AB_314134 1uL per sample

| | |
|---|---|
| Validation | All antibodies were purchased from established vendors with strict quality control assurances and validation statements can be found on the manufacturers' websites using the catalogue number or in the Antibody Registry database (https://antibodyregistry.org) via the provided RRID. TotalSeq antibodies were additionally titrated for optimal performance, with optimal concentrations listed above and in Supp Table 4. |

# Flow Cytometry

## Plots

Confirm that:

☒ The axis labels state the marker and fluorochrome used (e.g. CD4-FITC).

☒ The axis scales are clearly visible. Include numbers along axes only for bottom left plot of group (a 'group' is an analysis of identical markers).

☒ All plots are contour plots with outliers or pseudocolor plots.

☒ A numerical value for number of cells or percentage (with statistics) is provided.

## Methodology

| | |
|---|---|
| Sample preparation | All blood samples were collected, processed to PBMCs using a Ficoll-based approach and frozen in FBS with 10% DMSO within 4 hours of blood draw. For follow-up studies, T cells were directly isolated from whole blood using RosettaSep Human T-cell Enrichment Cocktail then immediately cryopreserved as described above. |
| Instrument | Cytek Aurora (5 laser), BD Melody |

| Software | FlowJo v10.8, BD FACSChorus (v2), Cytek SpectroFlo software (Version 2.0.2) |
| --- | --- |
| Cell population abundance | The purity of sorted cell populations (true naive CD4 T cells, MNP-2, naive CD8 T cells) was greater than 88%. |
| Gating strategy | FSC-A/SSC-A was used to identify lymphocytes. FSC-H/FCS-W and SSC-H/SSC-W were used to remove doublets. Viability was determined using live/Dead stain, gating on negative (i.e., live) cells. T cells were determined by the presence of CD3 and the absence of CD19 and CD14. CD4, CD8, CD27, CD45RA, CCR7, CD127 and CD25 were used to determine T cell subsets. For naive CD4 follow-up studies, CD95 was additionally used to separate true naive T cells (CD95-negative). MNP-2 cells were determined by CD244 and CD11b co-expression within CD8+ TCRab+ T cells. |

☒ Tick this box to confirm that a figure exemplifying the gating strategy is provided in the Supplementary Information.

