## [Peer Review File · Nature Immunology]

Peer Review Information

Journal: Nature Immunology

Manuscript Title: Tri-modal single cell profiling reveals a distinct pediatric CD8 $\alpha\alpha$ T cell subset and broad age-related molecular reprogramming across the T cell compartment

Corresponding author name(s): Dr. Claire Gustafson, Peter Skene

Reviewer Comments & Decisions:

Decision Letter, initial version:
--

17th Feb 2023

Dear Claire,

Thank you for providing a point-by-point response to the referees' comments on your Resource manuscript entitled, "Tri-modal single cell profiling reveals a distinct pediatric CD8 $\alpha\alpha$ T cell subset and broad age-related molecular reprogramming across the T cell compartment." As noted previously, while they find your work of considerable potential interest, they have raised a number of substantial concerns that must be addressed. In light of these comments, we cannot accept the current manuscript for publication, but would be very interested in considering a revised version that addresses these concerns along the lines proposed in your response.

Thus, we invite you to submit a substantially revised manuscript, however please bear in mind that we will be reluctant to approach the referees again in the absence of major revisions.

Specifically, the revision should include new experiments to address:

- (1) Increase validation cohort size/analysis and provide power calculation analysis. Include sex as a parameter for any differences observed between the groups.
- (2) Provide a functional analysis of the MNP-2 T cell subset identified in this study.
- (3) Provide a deeper scRNA-seq analysis on the MNP-2 cell population from cord blood and pediatric samples to address any heterogeneity in the cell population.
- (4) Assess Vd1/Vd2 TCR sequence usage of the MNP-2 cell subset and compare to other innate-like T cells.
- (5) Assess CD8b expression on the naive CD8 T cell population over time with TCR stimulation

Please include the additional textual clarifications as indicated in your response letter.

When you revise your manuscript, please take into account all reviewer and editor comments, please highlight all changes in the manuscript text file in Microsoft Word format.

* If you have not done so already please begin to revise your manuscript so that it conforms to our Resource format instructions at <http://www.nature.com/ni/authors/index.html>. Refer also to any guidelines provided in this letter.

The Reporting Summary can be found here:

When submitting the revised version of your manuscript, please pay close attention to our [href="https://www.nature.com/nature-portfolio/editorial-policies/image-integrity">Digital Image Integrity Guidelines.](https://www.nature.com/nature-portfolio/editorial-policies/image-integrity) and to the following points below:

[REDACTED]

If you wish to submit a suitably revised manuscript we would hope to receive it within 6 months. If you cannot send it within this time, please let us know. We will be happy to consider your revision so long as nothing similar has been accepted for publication at Nature Immunology or published elsewhere.

Nature Immunology is committed to improving transparency in authorship. As part of our efforts in this direction, we are now requesting that all authors identified as 'corresponding author' on published papers create and link their Open Researcher and Contributor Identifier (ORCID) with their account on the Manuscript Tracking System (MTS), prior to acceptance. ORCID helps the scientific community achieve unambiguous attribution of all scholarly contributions. You can create and link your ORCID from the home page of the MTS by clicking on 'Modify my Springer Nature account'. For more information please visit www.springernature.com/orcid.

Thank you for the opportunity to review your work.

Kind regards,

Laurie

Laurie A. Dempsey, Ph.D.
Senior Editor
Nature Immunology
l.dempsey@us.nature.com
ORCID: 0000-0002-3304-796X

Referee expertise:

Referee #1: T cell biology

Referee #2: Innate T cells

Referee #3: Human adaptive immunity

Reviewers' Comments:

Reviewer #1:

Remarks to the Author:

This paper describes a comparative multi-omic analysis of the peripheral blood lymphocyte compartment of a cohort of paediatric 1-13 years and older individuals aged 55-65. The conclusions from the work are that age drives molecular reprogramming across all T cell subsets toward a more activated basal state. The work also describes a naïve-like CD8a T cell subset in the paediatric cohort that was lost with age. The work is well done but I would question the power of this study given that the discovery cohort was 8 and 8 and the validation cohort 16 plus 16. The authors say there is no sex bias but is the data set sufficiently powered? What is the power calculation? What about ethnicity?

Re this population In Fig 4d the population labelled as MNP-2 seems to represent from <1% to 8% of the naïve T cells but there was heterogeneity and this population was not really visible in some of the paediatric cohort. What is the basis for this heterogeneity? How does it compare say with heterogeneity of other subsets>

Re the age decline the authors show this population to represent approximately >1% to approximately 3% of the paediatric cohort but to be very low levels in young adults and older people. However, is it valid to show the an apparent rate of decline as in Fig 4J when one only has a start and one end point? Given the heterogeneity in the paediatric cohort this could be quite misleading. Is it possible to address the heterogeneity question of the MNP-2 using cord blood samples?

Did the authors carry out any analysis of the functional impact of the changes shown in either the CD4 cells or the CD8 cells. How do the changes seen impact the functional capacity of the cells to proliferate or differentiate? Eg re the novel CD8 T cell subset there are flow plots of IL21 receptor for three MNP-2 populations from cord blood sample but these show heterogeneity. what about the protein expression in a paediatric cohort? On this point of IL-21 receptor expression? Are these IL21 receptors functional? Could IL -21 signal to these cells and how did the signal compare to signalling by other receptors using the common gamma chain like the IL7 receptor which are known to control CD8 T cell survival? Are there any experiments that address the functional capacity of these cells ? Can they proliferate or survive in response to IL21?

Reviewer #2:

Remarks to the Author:

This is a well performed study of the profile of PBMCs – focusing on T cells – with age. The newer tools available have been well utilised with a reasonably large dataset generated – it lacks TCR data which would have helped, but that is potentially a technical issue with the TEA-seq approach.

The conclusions about changes in naïve cells with age are quite interesting, although it would have been improved with some data on the relationship with function to give a bit of mcontext to the transcriptional/phenotypic findings. The CMV findings fit well with the literature.

For such a large resource dataset and broad descriptive paper it is quite well displayed but my concern is it is difficult to make that much sense of the MNP-2 subset which is the main novel conclusion drawn out.

1. Looking at Extended Figure 5, the cells are high in CD69 (similar to MAIT cells, which express maybe around 10% CD69 in healthy blood, likely due to recent microbial or cytokine activation). They also express PD-1 and CD71 which are also described as activation markers, together with CD11b (and CD244 also used). Even IL21R has been described as an activation marker. So from that heat map they look like an activated population. This doesn't mean that they are not also distinct (like MAIT cells), but those features alone don't seem very secure as a population marker.
2. They are described as innate-like in the abstract, but they don't seem to have any of the regular innate-like features you might expect at a transcriptional level (eg ZBTB16, IL18R). I could not really see how that conclusion was reached as the defining feature of the cells.
3. The MNP2 cells cluster closely with a subset of GD cells in Fig 5d, but there isn't very much information about this. Even whether these are Vd1 or Vd2 (some of latter can co-cluster with MAIT cells) isn't clear. I think this would warrant some more data – ideally using FACS/spectral analysis to

get a better idea of the cell populations directly in comparison.

4. Although it isn't really possible to perform a functional study on all the cell populations, it would be very much worth looking at the function of these MNP2 cells and try and answer the question of what they do. Firstly the authors could use some innate stimuli and test directly if they are innate-like functionally. Importantly, looking at whether they make/upregulate anything distinctive in response to TCR triggering would be helpful.

5. Although going back and getting TCRs on all of the T cells would be too much to expect, some TCR analysis of the MNP2 population (+/- a reference) would be helpful. The population is also at a certain point described as unconventional – those populations can have quite distinctive TCR usage associated with non-peptide ligands/restriction elements.

6. There are other human CD4a populations described as well as in chronic viral conditions as referenced, including one that links to CD161 expression amongst both non-MAIT and MAIT CD161+ cells seen in blood with age (Walker L Blood 2012). It would be interesting to see how that population relates to this one and where it sits on the UMAPs.

7. Interestingly in that same paper the authors report rapid downregulation of CD8b (in both CD161+ and – cells) upon stimulation/culture which would fit with the idea of recent activation affecting CD8 profile. The stability of CD8b could be readily addressed to explore this issue and help clarify the status of the MNP2 subset in children and adults.

7. I did not quite understand the violin plots in Fig 5d. They would suggest there was neither CD8a or b expression in MAIT cells. Some of that population can be DN (or CD4+) but it is only a fraction, so some clarification would be helpful.

8. Could the authors display what the frequency of MNP2 cells is in cord blood? I also did not understand the labelling on Fig 5k - it did not match with the legend.

Reviewer #3:

Remarks to the Author:

The study by Thomson, et al, presents a high dimensional analysis of peripheral blood T cells across different human ages—from 11-65yrs of age—using a tri-modal single cell profiling called TEA-seq, which includes protein (CITE-seq), gene expression (by scRNAseq) and chromatin accessibility (by scATAC-seq). The protein profiling uses a limited set of markers to define naïve and different memory T cell subsets that cannot be fully distinguished based on transcriptome alone, and they then did an analysis of differential gene expression and chromatin accessibility over age. They found differences in gene expression of naïve T cells between children and adults, including differential transcription factor activity in pediatric compared to adult T cells. They also identified some low frequency populations of naïve-like memory or memory-like naïve T cells, a subset which expressed CD8alpha-alpha subunits in children that is diminished in adults. While the analysis of naïve T cells is interesting and reveals the dynamics of naïve T cell aging, the resultant study is not comprehensive in the analysis of their dataset, or in investigating the functional role of the pediatric naïve-like subset they identified. Specific comments are enumerated below.

1. The analysis of naïve CD4 T cells with age and the differential expression analysis in Fig. 3 was clearly presented. However, the analysis and description of the results with naïve CD8 T cells was confusing with the introduction of both "Memory-like naïve precursors" (MNP-1 and MNP-2) and "naïve-like memory T cells". It was not clear how they distinguished these two types of T cells or whether they were using these designations interchangeably. Based on CD49d, it is possible that one or both of the MNP subsets could be so-called "Virtual Memory" T cells (see work by Steve Jameson and

Ross Kedl), but this subset is not mentioned or cited here. Also, neither of these MNP's expressed genes for effector molecules (Fig. 4h), so the designation of memory is not clear. Did they express CCL5?—this gene is a good marker for memory CD8 T cells in scRNAseq data.

2. The identification of a the CD8alpha-alpha subset enriched in pediatric blood is also designated a naïve-like memory subset, but again, the degree of memory-ness for this subset is not clear. Comprehensive functional analysis of this subset and the other naïve or memory-like subsets would certainly provide important information regarding their potential in immune responses, and should be feasible, given their ability to identify the MNP based on surface CD244 and CD11b expression.

3. The analysis of the dataset is incomplete in this manuscript, which is focused on naïve T cells, but they have information on all the major T cell subsets in blood. It would be interesting to determine if memory T cells (TEM) or TEMRA cells also undergo similar or distinct transcriptional changes in kids versus adults, as well as Tregs, if possible. The data are there and the pipeline could be readily applied to the other subsets. Without the full analysis of this dataset, the study seems incomplete.

Author Rebuttal to Initial comments

Editor and Reviewer Comment Responses

Manuscript < NI-RS35082A >

Tri-modal single cell profiling reveals a novel pediatric CD8aa T cell subset and broad age-related molecular reprogramming across the T cell compartment.

We thank the Reviewers and Editor for their valuable comments and recommendations, and we believe the revised manuscript is much stronger as a result of their feedback.

We have provided a point-by-point response as follows (the Editor/Reviewers' comments are in blue text and our responses are in black text).

Editor

(1) Increase validation cohort size/analysis and provide power calculation analysis.

Include sex as a parameter for any differences observed between the groups.

Response: We performed a robust power calculation on our validation cohort and provided these details in the methods section. The validation cohort size determined was $n=6$ per age group and our current cohort size was $n=16$, thus sufficiently powered for our analysis. Our validation analysis was not powered for specifically addressing sex related questions (as our initial cohort was all female), thus we have removed observations on sex-related differences. We have also added in a cohort demographics table for better clarity of the population studied (Supplemental Table 3).

(2) Provide a functional analysis of the MNP-2 T cell subset identified in this study.

Response: We performed new functional analyses on the MNP-2 subset using TCR, PMA/ionomycin and IL-21 stimulation, in conjunction with analyses of naïve, memory, MAIT and gdT cell subsets from 4 pediatric donors. These data are now provided in Fig 6, Extended Fig 9 and Extended Fig 10, as well as the results section. From these analyses, we determined that MNP-2 subset is a memory-like subset epigenetically poised for rapid effector responses.

(3) Provide a deeper scRNA-seq analysis on the MNP-2 cell population from cord blood and pediatric samples to address any heterogeneity in the cell population.

Response: To gain better insight into MNP-2 heterogeneity, we performed a deep dive analysis on pediatric MNP-2 cells from our TEA-seq dataset. This allowed us the ability to better delineate cell subsets from cell states based on epigenetic information in

addition to transcriptional and surface proteome information. We find multiple subsets within the MNP-2 population, but similar distribution of subsets across individual pediatric donors. These data are provided in Fig 5 and Extended Fig 8.

(4) Assess Vd1/Vd2 TCR sequence usage of the MNP-2 cell subset and compare to other innate-like T cells.

Response: We re-analyzed our TEA-seq dataset to identify Vd1 and Vd2 gdT cells based on their expression of TRDV1 and TRDV2 respectively. We then compared these subsets with MNP-2 and MAIT cells, observing that MNP-2 cells specifically cluster with a sub-population of Vd1 gdT cells. These data are provided in Fig 5 and Extended Fig 5.

(5) Assess CD8b expression on the naive CD8 T cell population over time with TCR stimulation

Response: We performed new analyses looking at the expression of MNP-2 specific cell markers (CD8b, CD244, CD11b, IL-21R) in addition to other classical activation markers (CD69, CD25, CD71) on cord blood naïve CD8 T cells in TCR stimulation experiments (**Extended Fig 7**). We find MNP-2 cells do not develop over the course of 7-days nor is the pattern of activation marker expression consistent with the MNP-2 phenotype.

Reviewer #1

1. This paper describes a comparative multi-omic analysis of the peripheral blood lymphocyte compartment of a cohort of paediatric 1-13 years and older individuals aged 55-65 . The conclusions from the work are that age drives molecular reprogramming across all T cell subsets toward a more activated basal state. The work also describes a naïve-like CD8aa T cell subset in the paediatric cohort that was lost with age. The work is well done but I would question the power of this study given that the discovery cohort was 8 and 8 and the validation cohort 16 plus 16. The authors say there is no sex bias but is the data set sufficiently powered? What is the power calculation? What about ethnicity?

Response: We consider our trimodal TEA-seq dataset a hypothesis-generating discovery resource designed with the intent of a dive deep into T cell subset heterogeneity on a unique combination of pediatric and older adult donors. Our validation cohort of 48 additional donors was to confirm and expand our MNP-2 finding. The minimum sample size required to identify a 1% change while controlling for Type I and Type II errors at $\alpha=0.05$, $\beta=0.2$, respectively, and applying an estimated frequencies standard deviation of $\sigma=0.45$, is $n=5$ per group for a two-sample t-test. Applying a sample size correction based on the Asymptotic Relative Efficiency (ARE) of the Mann Whitney U test (i.e., 15.7%) results in a minimum required sample size of $n=6$ per group to identify 1% differences to attain 80% power and control for Type I and II error rates at

$\alpha=0.05$, $\beta=0.2$, respectively. Thus, we are sufficiently powered in our cohorts. We have added our power analysis into the Methods. (Line 469-477) Our sample-size and power calculation do not cover

additional hypotheses beyond the pediatric-older adult cohort comparison, thus we have adjusted the text and conclusions about sex differences accordingly. There is some racial diversity within our cohort. For better transparency, we added a cohort demographics table, including age, sex, CMV infection status, race and ethnicity, into the supplement. (**Supplemental Table 3**)

2. Re this population In Fig 4d the population labelled as MNP-2 seems to represent from <1% to 8% of the naïve T cells but there was heterogeneity and this population was not really visible in some of the paediatric cohort. What is the basis for this heterogeneity? How does it compare say with heterogeneity of other subsets>

Re the age decline the authors show this population to represent approximately >1% to approximately 3% of the paediatric cohort but to be very low levels in young adults and older people. However, is it valid to show the an apparent rate of decline as in Fig 4J when one only has a start and one end point? Given the heterogeneity in the paediatric

cohort this could be quite misleading. Is it possible to address the heterogeneity question of the MNP-2 using cord blood samples?

Response: To further understand if there is heterogeneity within the MNP-2 population that contributes to variance, we performed a deeper dive specifically on the MNP-2 cells from the 8 pediatric donors in the TEA-seq data. (**Results Line 217-237**) From these analyses, we were able to determine that the MNP-2 population is composed of multiple subsets, but we found no major variation in subset distribution by donor. (**Fig 5h-k, Extended Fig 8**).

CMV infection correlated with some minor decrease in major subset (cluster 1) however did not significantly impact overall subset distribution (**Extended Fig 7**).

We additionally analyzed 3 cord blood scRNAseq datasets for MNP-2 frequencies and found these to be similar to that of children (**Figure 4j**).

Thus, although we cannot determine the underlying cause of the wide range of overall MNP-2 frequency within children, we do see that the general distribution of subsets with MNP-2 cells is relatively consistent.

3. Did the authors carry out any analysis of the functional impact of the changes shown in either the CD4 cells or the CD8 cells. How do the changes seen impact the functional

capacity of the cells to proliferate or differentiate? Eg re the novel CD8 T cell subset there are flow plots of IL21 receptor for three MNP-2 populations from cord blood sample but these show heterogeneity. what about the protein expression in a paediatric cohort? On this point of IL-21 receptor expression? Are these IL21 receptors functional? Could IL -21 signal to these cells and how did the signal compare to signalling by other receptors using the common gamma chain like the IL7 receptor which are known to control CD8 T cell survival? Are there any experiments that address the functional capacity of these cells ? Can they proliferate or survive in response to IL21?

Response: The function of MNP-2 was of great interest to us. To address the functionality of these cells, we performed IL-21 stimulation of CD8 T cells from 4 different children to compare early transcriptional responses in naïve, MNP-2 and memory T cell populations (**Results Line 275-282, Extended Figure 10**), using a new single cell CITE-seq method from 10x that allows fixation for rapid time course analyses. All populations responded to IL-21, however MNP-2 displayed some gene expression reflective of memory and MAIT populations, including the upregulation of perforin (PRF1). Moreover, MNP-2 demonstrated significantly higher expression of specific genes BCL6 and IL2RB and corresponding increases in CD122-ADT (gene name: IL2RB).

To better understand the effector capacity of MNP-2 cells, we additionally assessed responses to TCR and PMA/ionomycin stimulation (**Results Line 240-274, Fig 6, Extended Fig 9**), determining that MNP-2 are poised for rapid effector responses but display limited polyfunctional capacity.

Reviewer #2

This is a well performed study of the profile of PBMCs – focusing on T cells – with age. The newer tools available have been well utilised with a reasonably large dataset generated – it lacks TCR data which would have helped, but that is potentially a

technical issue with the TEA-seq approach. The conclusions about changes in naïve cells with age are quite interesting, although it would have been improved with some data on the relationship with function to give a bit of context to the transcriptional/phenotypic findings. The CMV findings fit well with the literature.

For such a large resource dataset and broad descriptive paper it is quite well displayed but my concern is it is difficult to make that much sense of the MNP-2 subset which is the main novel conclusion drawn out.

1. Looking at Extended Figure 5, the cells are high in CD69 (similar to MAIT cells, which express maybe around 10% CD69 in healthy blood, likely due to recent microbial or cytokine activation). They also express PD-1 and CD71 which are also described as activation markers, together with CD11b (and CD244 also used). Even IL21R has been described as an activation marker. So from that heat map they look like an activated population. This doesn't mean that they are not also distinct (like MAIT cells), but those features alone don't seem very secure as a population marker.

Response: We thank the Reviewer for raising this interesting point. To delineate MNP-2 cells from an activated population of CD8 T cells, we took a multi-pronged approach. Firstly, we in vitro stimulated naïve CD8 T cells from cord blood with aCD3/aCD28 beads to determine whether MNP-2 surface receptor markers are upregulated post-antigen exposure. From these data, we find that although CD244 is upregulated over

time (7-day time series), CD11b was not. Additionally, the kinetics of other activation markers, including CD8b, IL-21R, CD69 and CD71 did not coordinate into a MNP-2-like phenotype (**Results 213-215, Extended Fig 7**).

In our deep dive into MNP-2 heterogeneity (**Results Line 217-237**), we were also able to determine that there is a subset of MNP-2 cells that demonstrated high PD-1 (CD279-ADT) expression (cluster 4), implicating that a small portion of MNP-2 cells may indeed be a more activated population but not all (**Fig.5, Extended Figure 8**).

Finally, we find little expression of the cell cycle–related gene in MNP-2 cells pre- and post- cytokine activation (**Extended Fig 9f**), collectively demonstrating that MNP-2 cells are not acutely activated or proliferating naive CD8 T cells but a distinct, population of resting CD8 T cells.

2. They are described as innate-like in the abstract, but they don't seem to have any of the regular innate-like features you might expect at a transcriptional level (eg ZBTB16, IL18R). I could not really see how that conclusion was reached as the defining feature of the cells.

Response: Our usage of “innate-like” was a hypothesis based mainly on the expression of KLRC receptors. The MNP-2 population does not express ZBTB16 and little IL18R (**Extended Figure 8**) nor do they closely cluster with MAIT cells or a majority of gdT cells (**Figure 5b**). Based on new functional analyses, MNP-2 rapidly express IFNG similar to classic memory CD8 T cells

(see below, **Figure 6, Extended Fig 9**), thus we agree with the reviewer and have adjusted our terminology to describe these cells as a memory-like population that is poised for rapid effector functions, but not a specific innate-like subset.

3. The MNP2 cells cluster closely with a subset of GD cells in Fig 5d, but there isn't very much information about this. Even whether these are Vd1 or Vd2 (some of latter can co-cluster with MAIT cells) isn't clear. I think this would warrant some more data – ideally using FACS/spectral analysis to get a better idea of the cell populations directly in comparison.

Response: The close clustering of MNP-2 cells with a gdT cell population is interesting as MNP-2 cells are a TCRab+ subset (**Figure 5c, Extended Figure 6c**). Based on Pizzolato et.al., (PNAS, 2019), we further divided the gdT cell population into Vd1/Vd2 subsets based on their distinct RNA expression of TRDV1 and TRDV2. (**Results Line 192-201, Figure 5a-c**)

The small population of gdT subset that the MNP-2 population co-clusters with is a TRDV1-expressing population. Notably, this population was LEF1-high and equally distributed between CMV+ and CMV- donors, unlike the major Vd1-expressing population that was LEF1-negative, CD57-ADT positive and primarily from CMV+ donors, collectively indicating the co-clustering gdT population is more naïve or stem-like (**Extended Figure 5a-b**).

This population also was largely absent in older adults (**Extended Figure 5c**), suggesting a close link between age, thymic involution and the loss of these two naïve-like populations (MNP-2 and gdT subsets).

4. Although it isn't really possible to perform a functional study on all the cell populations, it would be very much worth looking at the function of these MNP2 cells and try and answer the question of what they do. Firstly the authors could use some innate stimuli and test directly if they are innate-like functionally. Importantly, looking at

whether they make/upregulate anything distinctive in response to TCR triggering would be helpful.

Response: We agree with the reviewer that the function of MNP-2 is of great interest. Thus, we performed CITE-seq on CD8 T cells pre- and post-stimulation with TCR (aCD3/aCD28) and PMA/ionomycin, to compare the early responses of MNP-2 cells with naïve and memory CD8 T cell subset, in addition to MAIT and gdT cell subsets from 4 pediatric donors (**Results Line 240-274, Fig 6., Extended Figure 9**).

From these analyses, we find that MNP-2 can rapidly make IFNG reflective of the memory compartment but display limited polyfunctional capacity compared to memory CD8 T cells. (**Figure 6f, 6g**) Thus, these cells have distinct responses from both naïve and memory compartments.

Additionally, based on high expression of IL-21R on MNP-2 cells (**Figure 4h, 5f, Extended Figure 6**), we further interrogated cellular responses to IL-21 stimulation (**Results Line 275-282, Extended Figure 10**), revealing both common and MNP-2 specific responses to this cytokine stimulation.

5. Although going back and getting TCRs on all of the T cells would be too much to expect, some TCR analysis of the MNP2 population (+/- a reference) would be helpful. The population is also at a certain point described as unconventional – those populations can have quite distinctive TCR usage associated with non-peptide ligands/restriction elements.

Response: Although of interest, none of the single cell omics technologies used in these studies allow the analysis of TCR sequences. Additionally, cord blood has

restrictions in genomic data sharing and pediatric samples are of limited volumes, collectively excluding in-depth TCR analysis of the MNP-2 population in these data.

6. There are other human CDaa populations described as well as in chronic viral conditions as referenced, including one that links to CD161 expression amongst both non-MAIT and MAIT CD161+ cells seen in blood with age (Walker L Blood 2012). It would be interesting to see how that population relates to this one and where it sits on the UMAPs.

Response: We have added CD161-ADT expression on the innate-like subset analysis into **Extended Figure 5**. We observed that MAIT and Vd2+ gdT cells expressed CD161 but not Vd1+ gdT or MNP-2 populations. (see the response to comment #3 above)

7. Interestingly in that same paper the authors report rapid downregulation of CD8b (in both CD161+ and – cells) upon stimulation/culture which would fit with the idea of recent activation affecting CD8 profile. The stability of CD8b could be readily addressed to explore this issue and help clarify the status of the MNP2 subset in children and adults.

Response: As part of our functional studies, we will assess CD8b expression on the naive CD8 T cell population over time with TCR stimulation to determine if activation contributes to loss of this surface marker. (**Results Line 213-215, Extended Figure 7**)

We did not find significant downregulation of CD8-beta in conjunction with upregulation of other MNP-2 specific surface markers (CD244, CD11b).

8. I did not quite understand the violin plots in Fig 5d. They would suggest there was neither CD8a or b expression in MAIT cells. Some of that population can be DN (or CD4+) but it is only a fraction, so some clarification would be helpful.

Response: To make the violin plots in 5d more clear, we added single cell expression points, showing that a majority of the MAIT cells do indeed express CD8A RNA. To further clarify this, we added CD8A protein expression into **Extended Figure 5**. (see response to comment #3 above)

9. Could the authors display what the frequency of MNP2 cells is in cord blood? I also

did not understand the labelling on Fig 5k - it did not match with the legend.

Response: Yes, we added cord blood data into **Figure 4j**. We have also updated the legend for clarification.

Reviewer #3

The study by Thomson, et al, presents a high dimensional analysis of peripheral blood T cells across different human ages—from 11-65yrs of age—using a tri-modal single cell profiling called TEA-seq, which includes protein (CITE-seq), gene expression (by scRNAseq) and chromatin accessibility (by scATAC-seq). The protein profiling uses a limited set of markers to define naïve and different memory T cell subsets that cannot be fully distinguished based on transcriptome alone, and they then did an analysis of differential gene expression and chromatin accessibility over age. They found differences in gene expression of naïve T cells between children and adults, including differential transcription factor activity in pediatric compared to adult T cells. They also identified some low frequency populations of naïve-like memory or memory-like naïve T

cells, a subset which expressed CD8alpha-alpha subunits in children that is diminished in adults. While the analysis of naïve T cells is interesting and reveals the dynamics of naïve T cell aging, the resultant study is not comprehensive in the analysis of their dataset, or in investigating the functional role of the pediatric naïve-like subset they identified. Specific comments are enumerated below.

1. The analysis of naïve CD4 T cells with age and the differential expression analysis in Fig. 3 was clearly presented. However, the analysis and description of the results with naïve CD8 T cells was confusing with the introduction of both “Memory-like naïve precursors” (MNP-1 and MNP-2) and “naïve-like memory T cells”. It was not clear how they distinguished these two types of T cells or whether they were using these designations interchangeably. Based on CD49d, it is possible that one or both of the MNP subsets could be so-called “Virtual Memory” T cells (see work by Steve Jameson and Ross Kedl), but this subset is not mentioned or cited here. Also, neither of these MNP’s expressed genes for effector molecules (Fig. 4h), so the designation of memory is not clear. Did they express CCL5?—this gene is a good marker for memory CD8 T cells in scRNAseq data.

Response: The term naïve-like memory T cells includes SCM and MNP cell subsets. The designation of SCM and MNP were called based on literature definitions in conjunction with unsupervised clustering (i.e., two “MNP” populations were found

instead of the predicted one). MNP-1 population is consistent with the previously identified populations in immune aging literature, however MNP-2 has, to the best of our knowledge, never been characterized. We have clarified the text throughout the manuscript to help make this nomenclature more clear.

We additionally investigated markers of TVM cells. Although MNP-2 cells showed an increase in CD122 expression in line with a TVM phenotype, they did not show other main markers of this population pre or post- IL-21 stimulation, such as high expression of EOMES and KLRC1. **(Discussion Line 329-331, Extended Figure 10e)**

We also interrogated CCL5 expression in pediatric CD8 T cell subset pre- and post-PMA/ionomycin stimulation. Pre-stimulation MNP-2 expressed a lower level of CCL5 compared with memory CD8 populations, however rapidly upregulated expression, along with CCL3 and CCL4, post-stimulation. **(Extended Figure 9)** Moreover, consistent with their openness of IFNG locus, which is a feature of memory but not naïve CD8 T cells, MNP-2 cells rapidly upregulated IFNG expression. **(Fig 6, Extended Figure 9)**.

2. The identification of a the CD8alpha-alpha subset enriched in pediatric blood is also designated a naïve-like memory subset, but again, the degree of memory-ness for this subset is not clear. Comprehensive functional analysis of this subset and the other naïve or memory-like subsets would certainly provide important information regarding their potential in immune responses, and should be feasible, given their ability to identify the MNP based on surface CD244 and CD11b expression.

Response: We agree with the reviewer that the function of MNP-2 is of great interest and warranted to be able to term this population “memory-like”. Thus, we performed CITE-seq on CD8 T cells pre- and post-stimulation with TCR (aCD3/aCD28) and PMA/ionomycin, to compare the early responses of MNP-2 cells with naïve and memory CD8 T cell subset, in addition to MAIT and gdT cell subsets in children (**Results Line 240-274, Fig 6., Extended Figure 9**).

From these analyses, we find that MNP-2 cells display a small number of subset-specific genes, including SPRY2 that is known to inhibit poly-functionally. Consistently, they rapidly make IFNG reflective of the memory compartment, albeit to a lesser level, but display limited polyfunctional capacity (co-expression of IFNG, IL2 and TNF)

compared to memory CD8 T cells (see response to comment #1 above). Thus, MNP-2 cells have features most reflective of memory T cells.

3. The analysis of the dataset is incomplete in this manuscript, which is focused on naïve T cells, but they have information on all the major T cell subsets in blood. It would be interesting to determine if memory T cells (TEM) or TEMRA cells also undergo similar or distinct transcriptional changes in kids versus adults, as well as Tregs, if possible. The data are there and the pipeline could be readily applied to the other subsets. Without the full analysis of this dataset, the study seems incomplete.

Response: In our global analysis of T cell subsets, we examined age-specific changes in memory T cell subsets and Tregs at a high level. (**Figure 2**) However, we specifically focused on a deeper dive into the naïve compartment in this manuscript as changes in this compartment are hallmarks of aging. We are doing further analysis to investigate other subsets but it is out of the scope of this naïve T cell focused paper. To facilitate researchers' specific questions about other T cell subsets using this dataset, we provide a data visualization tool as well as access to the raw and processed data.

Decision Letter, first revision:

15th Jul 2023

Dear Claire

Thank you for providing your-point-by-point response to the referees' comments on your revised Resource manuscript entitled, "Tri-modal single cell profiling reveals a distinct pediatric CD8 $\alpha\alpha$ T cell subset and broad age-related molecular reprogramming across the T cell compartment." We are very interested in the possibility of publishing your study in Nature Immunology. As noted previously, the referee did consider the revised manuscript to be improved, but there were still questions about the functionality of the MNP-2 T cell subset in the pediatric donor cohort. However, I think we can go forward with the manuscript that includes the retrospective analysis of the COVID MIS-C donor cohort mentioned in your rebuttal response, which shows this cell population is present in the health cohort of pediatric donors, but substantially decreased in MIS-C patients but rebounds upon recovery.

We therefore invite you to revise your manuscript taking into account all reviewer and editor comments. Please highlight all changes in the manuscript text file in Microsoft Word format.

- * Include a "Response to referees" document detailing, point-by-point, how you addressed each referee comment. If no action was taken to address a point, you must provide a compelling argument. This response will be sent back to the referees along with the revised manuscript.
- * If you have not done so already please begin to revise your manuscript so that it conforms to our Resource format instructions at <http://www.nature.com/ni/authors/index.html>. Refer also to any guidelines provided in this letter.
- * Please include a revised version of any required reporting checklist. It will be available to referees to aid in their evaluation of the manuscript goes back for peer review. They are available here:

Reporting summary:

When submitting the revised version of your manuscript, please pay close attention to our [href="https://www.nature.com/nature-portfolio/editorial-policies/image-integrity">Digital Image Integrity Guidelines.](https://www.nature.com/nature-portfolio/editorial-policies/image-integrity) and to the following points below:

[REDACTED]

We hope to receive your revised manuscript within two weeks. If you cannot send it within this time, please let us know. We will be happy to consider your revision so long as nothing similar has been accepted for publication at Nature Immunology or published elsewhere.

Nature Immunology is committed to improving transparency in authorship. As part of our efforts in this direction, we are now requesting that all authors identified as 'corresponding author' on published papers create and link their Open Researcher and Contributor Identifier (ORCID) with their account on the Manuscript Tracking System (MTS), prior to acceptance. ORCID helps the scientific community achieve unambiguous attribution of all scholarly contributions. You can create and link your ORCID from the home page of the MTS by clicking on 'Modify my Springer Nature account'. For more information please visit www.springernature.com/orcid.

Kind regards,

Laurie

Laurie A. Dempsey, Ph.D.
Senior Editor
Nature Immunology
l.dempsey@us.nature.com
ORCID: 0000-0002-3304-796X

Referee #1: T cell biology

Referee #2: Innate T cells

Referee #3: Human adaptive immunity

Reviewers' Comments:

Reviewer #1:

Remarks to the Author:

This work describes an expansive multi-omic analysis of the peripheral blood lymphocyte compartment of a paediatric cohort versus older individuals. The conclusions are that age drives molecular reprogramming of T cells. I find this conclusion overstated as we do not know if it is age or infection history that causes these changes. One would really want to see data from a longitudinal cohort. The work also describes a CD8 $\alpha\alpha$ IL21R T cell subset (MNP-2 cells) in the paediatric cohort that was lost with age. This population represents a small percentage of cells <5% and was heterogeneous but appeared poised for rapid responses. There is a huge body of data in the paper but the weakness is that the work is very descriptive. There is no evidence that this population of MNP-2 cells is important for paediatric immune responses or that their loss impacts immune responses in older people. The lack of diversity in the cohorts in terms of sex, ethnicity and environment is also an issue. Would this population be seen in paediatric cohorts across the globe. Would it be seen in 'non healthy' paediatric cohorts. Are there any clues of how infection impacts this population that gives some clue as to its importance? Maybe from single cell RNA seq data from COVID cohorts?

Reviewer #2:

Remarks to the Author:

This is a revised version of the manuscript. The focus of the revisions was on the MNP2 subset - these issues have been well addressed. It is still not totally clear what they are there for, but the subset is better clarified, including new functional analyses.

The Figure 4j is useful but I wondered whether it really was % rather than proportion, judging by the text and other data. Putting the actual % of the cells in paediatric/cord blood somewhere in the abstract and/or introduction would be helpful in orienting the reader and giving context.

Reviewer #3:

Remarks to the Author:

The authors have provided new functional data and CITE-seq analyses showing additional features of the paediatric MNP-2 population including its intrinsic functional capacity for IFN- γ production and responses to IL-21. Since there are age-related changes across multiple subsets as shown in Fig. 2, I suggest that the authors provide a rationale in the results for focusing on naive T cells in this study-- perhaps due to the magnitude of the changes by chromatin analysis. Also, I suggest that the authors include extended data figure 10 as the last main figure 7, because they have the space and it rounds out the functional analysis. Can the authors speculate on the role of this MNP-2 subset in paediatric immunity-- perhaps to augment lymphoid responses--interact with T-follicular helper cells?

Author Rebuttal, first revision:

Point-by-point response to editor (on reviewers comments):

Reviewer #1

(Remarks to the Author)

This work describes an expansive multi-omic analysis of the peripheral blood lymphocyte compartment of a paediatric cohort versus older individuals. The conclusions are that age drives molecular reprogramming of T cells. I find this conclusion overstated as we do not know if it is age or infection history that causes these changes. one would really want to see data from a longitudinal cohort.

RESPONSE: We agree with the reviewer that disentangling the effects of age from life-time exposure is very difficult, as infection history, diet, environmental exposures, etc can all be linked with time. However, the aging process is subtle and a multi-decade longitudinal study is out of scope for this study. Thus, we have modified our conclusions throughout the manuscript about age 'driving' molecular programming, acknowledging this potential limitation.

The work also describes a CD8aa IL21R T cell subset (MNP-2 cells) in the paediatric cohort that was lost with age. This population represent a small percentage of cells <5% and was heterogenous but appeared poised for rapid responses. There is a huge body of data in the paper but the weakness is that the work is very descriptive. There is no evidence that this population of MNP-2 cells is important for paediatric immune responses or that their loss impacts immune responses in older people. The lack of diversity in the cohorts in terms of sex, ethnicity and environment is also an issue. Would this population be seen in paediatric cohorts across the globe. Would it be seen in 'non healthy' paediatric cohorts. Are there any clues of how infection impacts this population that gives some clue as to its importance? Maybe from single cell RNA seq data from COVID cohorts?

RESPONSE: With respect to population diversity, there are significant challenges in recruiting diverse cohorts for pediatric immune health studies. We have already acknowledged this limitation in our cohort and provided the specific cohort demographics with this last round of revisions. Expansion of our current cohort is out of the scope of this study. In the discussion, we highlight the need to expand cohort diversity to better understand immune heterogeneity.

The reviewer comments raise a good question about the extendibility of our MNP-2 finding across other pediatric cohorts, and whether this population may play a role in infection. Pediatric studies with large enough cell numbers to identify this population are limited due to the typically low number of cells interrogated by scRNA-seq per sample. However, from an analysis of an external COVID MIS-C scRNA-seq study in a primarily Hispanic/Latino pediatric cohort (PMID:

33891889), we were able to easily detect MNP-2 in PBMCs based on our defined cell signature (Fig 7f). Moreover, comparison of healthy children to those with MIS-C showed a striking loss in the frequencies of MNP-2 with increasing disease severity, with a robust rebound post-recovery. (Fig 7g and Extended Data Fig 10i) We have added these data into the results to support a potential role of this cell population in immune responses to infection and inflammation. (Lines 288-300) Moreover, this dataset also included T cell receptor sequencing, which we have now added into to Extended Data Fig 10j. These new data collectively demonstrate that, although initially 'descriptive' in nature, our dataset and findings can be translated across other studies and provide new insight into the human immune system and its functionality.

Reviewer #2

(Remarks to the Author)

This is a revised version of the manuscript. The focus of the revisions was on the MNP2 subset - these issues have been

well addressed. It is still not totally clear what they are there for, but the subset is better clarified, including new functional analyses.

RESPONSE: Thank you for the positive feedback and consideration of our edits. We have added in further discussion (Lines 343-370) about the potential roles of MNP-2 cells in infection and inflammation. Moreover, we included analysis of pediatric MIS-C dataset, which highlights the loss of MNP-2 cells in active disease but its rebound post-resolution, supporting the potential role for MNP-2 cells in inflammatory responses in children. (Result Line 288-300, Fig 7f, 7g, Extended Data Fig 10i, 10j)

The Figure 4j is useful but I wondered whether it really was % rather than proportion, judging by the text and other data. Putting the actual % of the cells in pediatric/cord blood somewhere in the abstract and/or introduction would be helpful in orienting the reader and giving context.

RESPONSE: Yes, the reviewer is correct that this graph is proportion and not %. We apologize for the oversight and have revised the axis labeling. We have also add the % of MNP-2 into the introduction (**Line 42**) and results (**Line 184**) for better clarity.

Reviewer #3

(Remarks to the Author)

The authors have provided new functional data and CITE-seq analyses showing additional features of the pediatric MNP-2 population including its intrinsic functional capacity for IFN-gamma production and responses to IL-21. Since there are age-related changes across multiple subsets as shown in Fig. 2, I suggest that the authors provide a rationale in the results for focusing on naive T cells in this study-- perhaps due to the magnitude of the changes by chromatin analysis.

RESPONSE: The loss of naïve CD8 T cells is a hallmark of aging. However, there is controversy in the immune aging field on whether these cells are or are not intrinsically changing with age. Our observation that naive CD4 T cells had a greater number of transcriptional and epigenetic changes between children and adults than naïve CD8 T cells was thus of significant interest. We have emphasized this point further in the results. (**Lines 90-94**)

Also, I suggest that the authors include extended data figure 10 as the last main figure 7, because they have the space and it rounds out the functional analysis.

RESPONSE: We agree with this suggestion and have moved extended figure 10 to the main figures (**Fig 6**).

Can the authors speculate on the role of this MNP-2 subset in pediatric immunity-- perhaps to augment lymphoid responses--interact with T-follicular helper cells?

RESPONSE: We have added in further discussion about the potential roles of MNP-2 cells in infection and inflammation. (**Lines 343-370**) Additionally, we included analysis of pediatric MIS-C dataset, which highlights the loss of MNP-2 cells in active disease and recovery post-resolution, supporting the potential role for MNP-2 cells in immune responses in children. (**Result Line 288-300, Fig 7f, 7g, Extended Data Fig 10i, 10j**)

Decision Letter, second revision:

31st Jul 2023

Dear Claire,

Thank you for submitting your revised manuscript "Tri-modal single cell profiling reveals a distinct pediatric CD8 $\alpha\alpha$ T cell subset and broad age-related molecular reprogramming across the T cell compartment" (NI-RS35082C). I have looked over your responses to the referees' comments posed in the previous round of review, and I think the addition of the new cohort datasets will satisfy the comment posed by referee #1 regarding where similar findings are likewise present in more diverse patient cohorts. Thus, we'll be happy in principle to publish it in Nature Immunology, pending minor revisions to comply with our editorial and formatting guidelines.

We will now perform detailed checks on your paper and will send you a checklist detailing our editorial and formatting requirements in about a week. Please do not upload the final materials and make any revisions until you receive this additional information from us.

If you had not uploaded a Word file for the current version of the manuscript, we will need one before beginning the editing process; please email that to immunology@us.nature.com at your earliest convenience.

Thank you again for your interest in Nature Immunology Please do not hesitate to contact me if you have any questions.

Kind regards,

Laurie

Laurie A. Dempsey, Ph.D.
Senior Editor
Nature Immunology
l.dempsey@us.nature.com
ORCID: 0000-0002-3304-796X

Final Decision Letter:

Dear Claire,

I am delighted to accept your manuscript entitled "Tri-modal single cell profiling reveals a distinct pediatric CD8 $\alpha\alpha$ T cell subset and broad age-related molecular reprogramming across the T cell compartment" for publication in an upcoming issue of Nature Immunology.

Over the next few weeks, your paper will be copyedited to ensure that it conforms to Nature Immunology style. Once your paper is typeset, you will receive an email with a link to choose the appropriate publishing options for your paper and our Author Services team will be in touch regarding any additional information that may be required.

Please note that *Nature Immunology* is a Transformative Journal (TJ). Authors may publish their research with us through the traditional subscription access route or make their paper immediately open access through payment of an article-processing charge (APC). Authors will not be required to make a final decision about access to their article until it has been accepted. [Find out more about Transformative Journals](https://www.springernature.com/gp/open-research/transformative-journals).

Your paper will be published online soon after we receive your corrections and will appear in print in the next available issue. Content is published online weekly on Mondays and Thursdays, and the embargo is set at 16:00 London time (GMT)/11:00 am US Eastern time (EST) on the day of publication. Now is the time to inform your Public Relations or Press Office about your paper, as they might be interested in promoting its publication. This will allow them time to prepare an accurate and satisfactory press release. Include your manuscript tracking number (NI-RS35082D) and the name of the journal, which they will need when they contact our office.

About one week before your paper is published online, we shall be distributing a press release to news organizations worldwide, which may very well include details of your work. We are happy for your institution or funding agency to prepare its own press release, but it must mention the embargo date and Nature Immunology. Our Press Office will contact you closer to the time of publication, but if you or your Press Office have any enquiries in the meantime, please contact press@nature.com.

Also, if you have any spectacular or outstanding figures or graphics associated with your manuscript - though not necessarily included with your submission - we'd be delighted to consider them as candidates for our cover. Simply send an electronic version (accompanied by a hard copy) to us with a possible cover caption enclosed.

Please note that we encourage the authors to self-archive their manuscript (the accepted version before copy editing) in their institutional repository, and in their funders' archives, six months after publication. Nature Portfolio recognizes the efforts of funding bodies to increase access of the research they fund, and strongly encourages authors to participate in such efforts. For information about our editorial policy, including license agreement and author copyright, please visit

www.nature.com/ni/about/ed_policies/index.html

Kind regards,

Laurie

Laurie A. Dempsey, Ph.D.
Senior Editor
Nature Immunology
l.dempsey@us.nature.com
ORCID: 0000-0002-3304-796X